



# Improving temperature reconstructions from ice-core water-isotope records

Bradley R. Markle[1, 2, 3] and Eric J. Steig[3]

[1]Institute of Arctic and Alpine Research, University of Colorado, Boulder
[2]Department of Geological Sciences, University of Colorado, Boulder
[3]Department of Earth and Space Sciences, University of Washington

**Correspondence:** Bradley R. Markle (bradley.markle@colorado.edu)

**Abstract.** Oxygen and hydrogen isotope ratios in polar precipitation are widely used as proxies for local temperature. Used in combination, oxygen and hydrogen isotope ratios also provide information on sea surface temperature at the oceanic moisture source locations where polar precipitation originates. Temperature reconstructions obtained from ice core records generally rely on linear approximations of the relationships among local temperature, source temperature and water-isotope values. However,

there are important nonlinearities that significantly affect such reconstructions, particularly for source-region temperatures. Here, we describe a temperature reconstruction method that accounts for these nonlinearities. We provide new reconstructions of absolute surface temperature, condensation temperature, and source-region evaporation temperature for all long Antarctic ice-core records for which the necessary data are available. We also provide thorough uncertainty estimates on all temperature histories. Our reconstructions constrain the pattern and magnitude of polar amplification in the past and reveal asymmetries in

the temperature histories of East and West Antarctica.

## 1   Introduction

Stable-isotope ratios of water have been the foundational proxy of polar paleoclimate research for more than a half-century (Langway, 1958; Gonfiantini, 1959; Dansgaard, 1964). Primarily used as a temperature proxy, stratigraphic records of water-isotope ratios in ice sheets provide detailed histories of Earth's climate over hundreds of thousands of years (Dansgaard et al.,

1969; Petit et al., 1999), providing insight into the past magnitudes, spatial patterns, and phasing of climate change across the globe (Masson-Delmotte et al., 2006; Barbante et al., 2006; WAIS Divide Project Members et al., 2013, 2015). Both oxygen and hydrogen have stable isotopes whose ratios ($^{18}O/^{16}O$ and $^{2}H/^{1}H$) are commonly expressed as deviations, $\delta^{18}$O and $\delta$D, from Vienna Standard Mean Ocean Water (VSMOW):

$$\delta = \frac{R_x - R_{std}}{R_{std}} \tag{1}$$

where $R_x$ is the ratio in the sample and $R_{std}$ is the ratio in VSMOW.

Poleward transport of moisture by the climate system, the progressive removal of moisture from the atmosphere by condensation and precipitation, and the fractionation of water isotope ratios during phase changes are all processes inherently linked





to temperature, and together underpin the use of water isotope ratios in polar precipitation as a temperature proxy (Craig, 1961; Epstein et al., 1963; Dansgaard, 1964; Gonfiantini, 1965). The strong empirical correlation between the water-isotope ratios in precipitation and surface temperature supports this interpretation (Petit et al., 1999; Jouzel et al., 1997; Masson-Delmotte et al., 2008). Air temperatures during condensation (Petit et al., 1999; Jouzel et al., 1997) and during initial moisture evapora-

tion (Vimeux et al., 2002) can be reconstructed from ice-core water-isotope records, if the relevant scaling relationships can be determined from theory, models, or observations (Vimeux et al., 2002; Kavanaugh and Cuffey, 2002; Stenni et al., 2010). Here, we examine the widely-used assumption of linearity in the scaling relationships between water-isotope ratios and temperature.

### 1.1  Temperature reconstructions

Any interpretation of water isotope ratios as a proxy for temperature requires a model, whether conceptual, statistical, or

numerical. A conceptual model of progressive distillation and integrated fractionation (e.g. Dansgaard (1964)) is sufficient to qualitatively interpret variations in water isotope ratios as variations in temperature in the high latitudes. The simplest quantitative interpretation of ice-core water-isotope records relies on the empirical correlation between observed water isotope ratios of precipitation and surface temperature at the precipitation site (Petit et al., 1999; Jouzel et al., 1997). A limitation of this approach is the possibility to conflate the "spatial slope" between water-isotopes and temperature, that is the relationship

observed across a range of modern sites, and the "temporal slope", the relationship at a single point through time (Jouzel et al., 1997). This approach also does not account for simultaneous and independent changes in evaporation conditions, which can impact high latitude water isotopes ratios in several ways. Initial evaporation temperature, together with the condensation temperature, determines the total temperature gradient through which moisture must be distilled to reach a given site. Further, evaporative conditions set the initial isotopic values of the vapor before distillation. The isotope ratios of vapor above the ocean

depend on the temperature during evaporation, the isotopic values of the seawater, and the occurrence of kinetic fractionation during evaporation, which is driven by sub-equilibrium relative humidity and influenced by sea surface temperature and wind speed (Merlivat and Jouzel, 1979; Jouzel et al., 1982).

A more complete approach to reconstructing temperature from water-isotope records is to employ numerical models that account for the combined influence of variability in both evaporation and condensation temperatures, as well as other fac-

tors. Reconstructing two unknowns (i.e. both evaporation-source and condensation-site temperatures) requires two constraints, which are provided by the oxygen and hydrogen isotope ratios and the relationship between them. While the oxygen and hydrogen isotope systems have similar behavior in the atmosphere, there are differences in their response to the same environmental conditions and to processes such as kinetic fractionation. The deuterium excess is the weighted difference between $\delta^{18}O$ and $\delta D$, $d_{xs} = \delta D - 8 \times \delta^{18}O$, and is commonly used to quantify these differences (Dansgaard, 1964; Merlivat and Jouzel, 1979).

Changes in water-isotope parameters measured in precipitation at an ice core site, $\Delta\delta^{18}O$ and $\Delta d_{xs}$, can be conceptualized as driven by changes in site and evaporation source temperature, $\Delta T_{site}$ and $\Delta T_{source}$:

$$\Delta\delta^{18}O = \gamma_1 \Delta T_{site} + \gamma_2 \Delta T_{source} \qquad (2)$$



$$\Delta d_{xs} = \beta_1 \Delta T_{site} + \beta_2 \Delta T_{source} \tag{3}$$

where $\beta$ and $\gamma$ are the partial derivatives of $\delta^{18}O$ and $d_{xs}$ with site and source temperature, respectively. The magnitudes of $\beta$ and $\gamma$ can be diagnosed from water-isotope distillation models for the ice-core site in question (Vimeux et al., 2002; Kavanaugh and Cuffey, 2002; Stenni et al., 2010; Uemura et al., 2012). Once these slopes are established, the equations may be solved for

$\Delta T_{site}$ and $\Delta T_{source}$ using records of $\delta^{18}O$ and $d_{xs}$ (Vimeux et al., 2002; Stenni et al., 2010; Uemura et al., 2012).

## 1.2 Nonlinearities in isotope fractionation and the deuterium excess definition

The temperature reconstruction approach described above depends on the assumption that the parameters, $\beta$ and $\gamma$, are fixed in time and independent of temperature. However, the $\beta$ and $\gamma$ parameters, as diagnosed from model simulations, are found to be different for different ice-core sites with differing modern surface conditions (e.g. Stenni et al. (2010) and Uemura et al.

(2012)). This means that $\beta$ and $\gamma$ depend on the site conditions, which obviously change over time.

Another issue with the linear reconstruction approach is the definition of the deuterium excess parameter (Uemura et al., 2012; Markle et al., 2017). The origin of the slope in the definition of deuterium excess is an empirical fit to global precipitation measurements (Dansgaard, 1964). However, a linear relationship between $\delta^{18}O$ and $\delta D$ is not fundamental (Craig, 1961); equilibrium fractionation alone drives a nonlinear relationship between $\delta^{18}O$ and $\delta D$ (Markle et al., 2017). While the effects

of source-region conditions on deuterium excess of vapor are nearly linear during initial evaporation (Merlivat and Jouzel, 1979; Uemura et al., 2008), the signal is not uniformly preserved as moisture is transported toward the deposition site. Kinetic fractionation that occurs during transport (Jouzel et al., 1982) alters the deuterium excess of the vapor, as does equilibrium fractionation during condensation, owing to biases in the linear definition (Markle et al., 2017). Thus the sensitivity of $d_{xs}$ in precipitation to evaporation and condensation temperatures must vary as a function of the total condensation and fractionation

experienced during transport to any deposition site, and is thus a function of $T_{site}$.

Some of these issues have been addressed by redefining the deuterium excess parameter (Uemura et al., 2012; Markle et al., 2017). Uemura et al. (2012) fit a second-order polynomial to a compilation of $\delta'^{18}O$ and $\delta'D$ data, where $\delta'_x = ln(1 + \delta_x)$, and defined a phenomenological, non-linear deuterium excess parameter:

$$d_{ln} = \delta'D - \left(A \times (\delta'^{18}O)^2 + B \times \delta'^{18}O\right) \tag{4}$$

with coefficients $A = -28.5$ and $B = 8.47$ (note that the coefficients and $\delta'$ values are unitless; for example $\delta'^{18}O = 0.040$ not 40‰.

This definition of deuterium excess reduces the influence of kinetic fractionation during transport and the biases inherent to the linear definition, making it a more faithful qualitative proxy for source-region conditions (Uemura et al., 2012; Markle et al., 2017), and is particularly important at the coldest Antarctic sites where nonlinear effects overwhelm the $d_{xs}$ definition.

However, the same distillation processes that lead to biases in the linear definition of the deuterium excess parameter will also bias the results of temperature reconstructions if fixed sensitivities (Equations 2 and 3) are assumed.



Here we examine these issues in water isotope-based temperature reconstructions and suggest an improved technique.

## 2 A (relatively) simple water isotope model

The quantitative reconstruction of temperatures from water isotope ratios rests on the encapsulation of fractionation processes in models. Any investigation into nonlinearity in those relationships will depend on the representation of those physics. To
assess the importance of those nonlinearities, we construct a model that is relatively simple while still faithfully representing the observed relationships between the hydrogen and oxygen isotope ratios in polar precipitation. We describe the construction of the Simple Water Isotope Model (SWIM) in detail in the Appendix. Here we describe the conceptual framework of the model.

The underpinning of SWIM is shared by many water isotope models: the transport and distillation of moisture down clima-
tological temperature gradients. Moisture is evaporated from the oceans in the low and mid latitudes and transported toward the poles. As air cools, the saturated vapor pressure decreases nonlinearly, and moisture above saturation is removed by precipitation. During these phase changes, water fractionates; the vapor and precipitation falling from it become increasingly depleted in the heavier isotope. The total fractionation at any point is a consequence of the temperature gradient through which the water is distilled, as well as the mean temperature of that gradient, owing to nonlinearity in the Clausius-Clapeyron relationship. A
change in the average condensation temperature at a site thus results in a change in the isotope ratios of precipitation at that site. This is the essential (though not sole) reason that high-latitude water-isotope ratios are a useful temperature proxy. It is driven by two basic nonlinear processes, the Clausius-Clapeyron relationship and Rayleigh distillation (see A2.2).

Other processes can be important as well. The temperature dependence of fractionation factors, for example, generally amplifies the temperature relationship. While any single precipitation event at a site may be subject to a variety of additional
factors and processes, the long term mean is strongly influenced by climatological moisture distillation.[1]

Our model distills moisture down thermodynamic pathways defined by temperature and pressure. Changes in water-isotope ratios are driven neither by changes in space nor time but by changes in the thermodynamic variables that cause the water to change phase. The temperature gradient of the pathway is prescribed from an initial evaporation temperature, $T_0$, to a final condensation temperature, $T_c$. The pathways are pseudo-adiabatic, consistent with isentropic moisture transport to the
Antarctic (Bailey et al., 2019) and the basic assumption of Raleigh distillation, that moisture is removed after precipitation. A superposition of many thermodynamic pathways is required to represent a single Antarctic precipitation site, reflecting both the range of precipitation conditions experienced at a site as well as moisture transport from sources with a distribution of evaporative conditions (Markle et al. (2017), Figure A8). An example of a set of these pathways is shown in Figure 1. We use climatological correlations to relate initial evaporation air temperature, $T_0$, to other initial conditions including sea surface
temperature, $SST_0$, and relative humidity, $RH_0$ (see A1.1).

---

[1]This process is not incidental but rather fundamental to the climate system itself. The Earth's surface absorbs shortwave radiation form the sun and transfers that heat to the atmosphere. The majority of that transferred heat is latent, in the form of evaporated moisture. The basic climatological function of the atmosphere and its motions is to transport heat from regions of net gain to regions of net loss, from low to high latitudes, and a large fraction of that transported heat is also moisture.



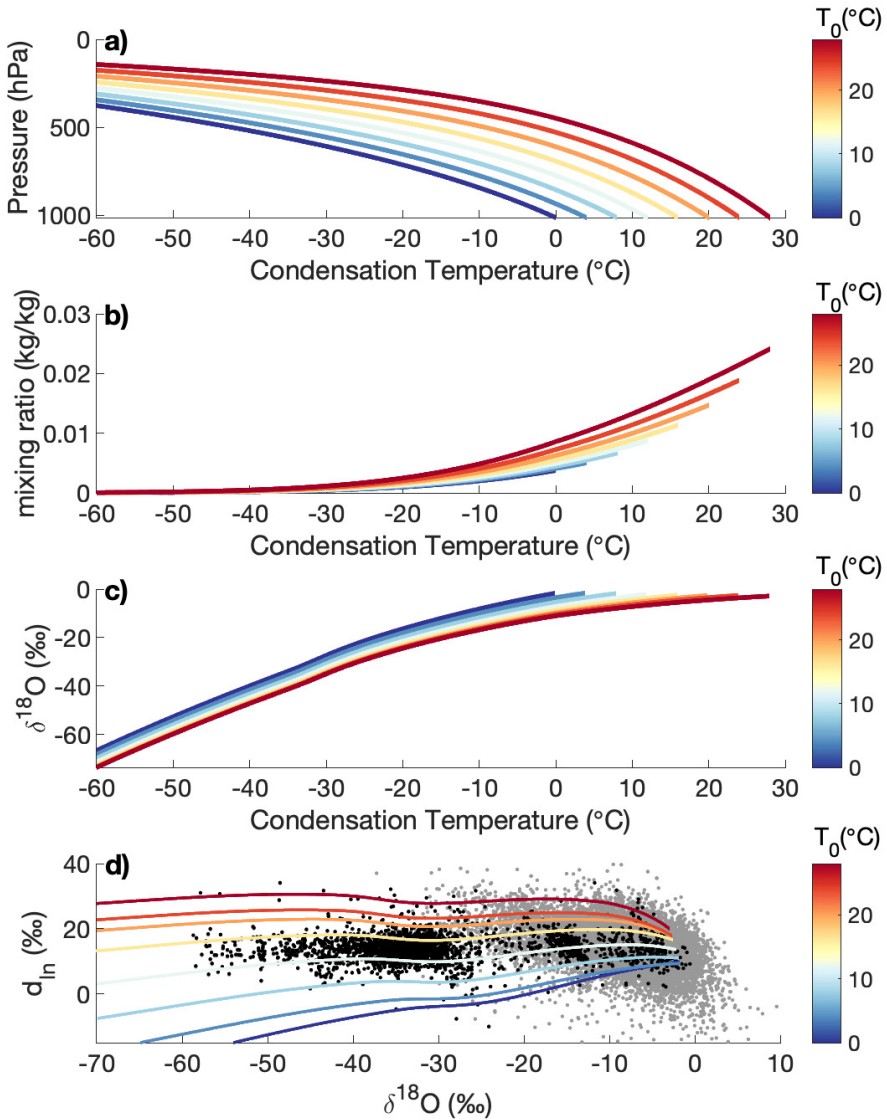

**Figure 1.** Simple Water Isotope Model. **a)** Condensation temperature-pressure pathways for pseudo-adiabatic distillation. Pathways are colored by initial evaporation air temperature in all subplots. **b)** Mixing ratios for all pathways. **c)** $\delta^{18}O$ of precipitation for all pathways. **d)** Relationship between $\delta^{18}O$ and $d_{ln}$ of precipitation for all pathways. Black dots show water isotope values of annual averaged precipitation from a compilation of global observations (see text), while grey dots show broader set of monthly observations from the GNIP data base (IAEA, 2001).





We consider the temperature dependence of kinetic and equilibrium fractionation during both evaporation and transport, as well as mixed liquid and ice phases of precipitation. The model incorporates supersaturation at very cold conditions, which is tuned to match the observed relationship between the oxygen and hydrogen isotope ratios in global precipitation (Figure 1.d) rather than the relationships between those parameters and climate variables such as temperature. We investigate the sensitivity

of the model and the resulting reconstructions to uncertainties and assumptions including fractionation factors, evaporative closure assumptions, precipitation schemes, supersaturation, the pseudo-adiabatic pathway, initial climatological correlations during evaporation, and non-uniqueness. We also investigate the influence of mixing both during evaporation and transport, the potential influence of seasonality (and intermittency) of precipitation and evaporation, as well as the relationship between surface temperature at a site and the vertically-integrated, precipitation-weighted condensation temperature above that site. We

make no attempt to model post-depositional processes.

## 3 Temperature-dependent slopes

To investigate the sensitivity of water-isotope ratios of Antarctic precipitation to site and source conditions, we use SWIM to model isotopic state spaces. We run SWIM through a large ensemble of temperature pathways defined by $T = T_0, T_0 - dT, ..., T_c$, with $dT = 0.1°C$. We run nearly 24,000 trajectories to fill the plausible parameter space of $0°C \leq T_0 \leq 28°C$ and

$-70°C \leq T_c \leq 10°C$. We first examine the model with base assumptions and parameterizations[2]. We next investigate the sensitivity to these choices.

The modeled isotopic state spaces are shown as maps whose $x$ and $y$ coordinates are the condensation and evaporation temperatures, $T_c$ and $T_0$ respectively, and whose $z$ (color) dimension is the isotopic value of final precipitation in Antarctica: $\delta^{18}O$ in Figure 2.a, $\delta D$ in Figure 2.b, $d_{xs}$ in Figure 2.c, and $d_{ln}$ in Figure 2.d. The partial derivatives of these isotopic parameter

spaces with respect to $T_c$ and $T_0$, as functions of both $T_c$ and $T_0$, are shown in Figures 3 and 4, respectively.

The gradients of both the $\delta^{18}O$ and $\delta D$ surfaces are predominantly in the direction of the condensation temperature (the $x$-axis in Figure 2), emphasizing the strong condensation-temperature dependence of these parameters. However, the slopes of both $\delta^{18}O$ and $\delta D$ are not strictly linear with condensation temperature $T_c$, clearly varying with its absolute value (and to a much lesser extent with the evaporation temperature, $T_0$, due to its influence on the total distillation gradient). Further, the

partial slopes of $\delta^{18}O$ and $\delta D$ with respect to the evaporation source temperature depend strongly on the absolute values of both the evaporation and condensation temperatures, evidenced by the changing angle of the contour lines in Figure 2. The partial derivatives of the isotopic surfaces with respect to $T_0$ and $T_c$ are shown in Figures 3 and 4. It is important to recognize that the partial derivatives with respect to $T_0$ are not for the initial vapor at the point of evaporation, but for the precipitation after the vapor has passed through the distillation pathway to the final precipitation site. The sensitivity of isotopic values of

precipitation to source region conditions is a function of the total distillation that the moisture experiences.

---

[2]The base model includes local closure during evaporation, values of $SST_0$ and $RH_0$ for a given $T_0$ determined by spline fit to NCEP/NCAR climatology, and a tuned supersaturation such that $S = 1 - 0.00525T$, where $T$ is in $°C$. See Appendix for details.



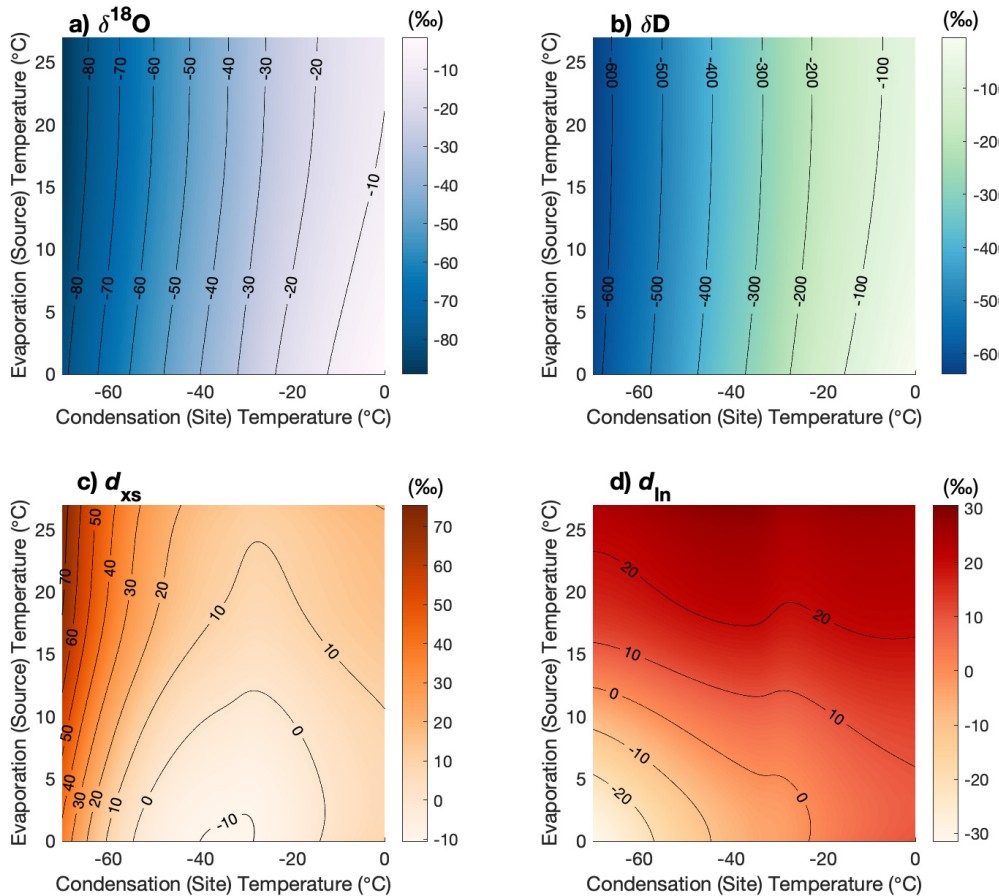

**Figure 2.** Water isotope state spaces as a function of the boundary conditions $T_c$, the condensation temperature, and $T_0$, the mean evaporation temperature. Surface shading and contour lines are the water isotope values of precipitation (in units of ‰) **a)** $\delta^{18}O$ **b)** $\delta D$, **c)** linear deuterium excess $d_{xs}$, and **d)** nonlinear deuterium excess $d_{ln}$.

The modeled $d_{xs}$ surface shows strong slopes along both the condensation temperature and evaporation temperature axes (Figure 2.c), as does modeled $d_{ln}$(Figure 2.d). The $d_{ln}$ depends more strongly on the evaporation temperature than the $d_{xs}$. In particular, at the coldest condensation temperatures, variability in $d_{xs}$ is dominated by the condensation temperature, reflecting the influence of kinetic fractionation during condensation and the nonlinear bias inherent to the historical linear definition (Uemura et al., 2012; Markle et al., 2017). These model results (Figures 2, 3, and 4) demonstrate that the logarithmic definition of the deuterium excess parameter ($d_{ln}$) is a more faithful qualitative proxy for source-region conditions than the linear definition ($d_{xs}$). Even at very low condensation temperatures, $d_{ln}$ still depends strongly on the initial evaporation temperature, whereas linear $d_{xs}$ becomes more dependent on condensation temperature.



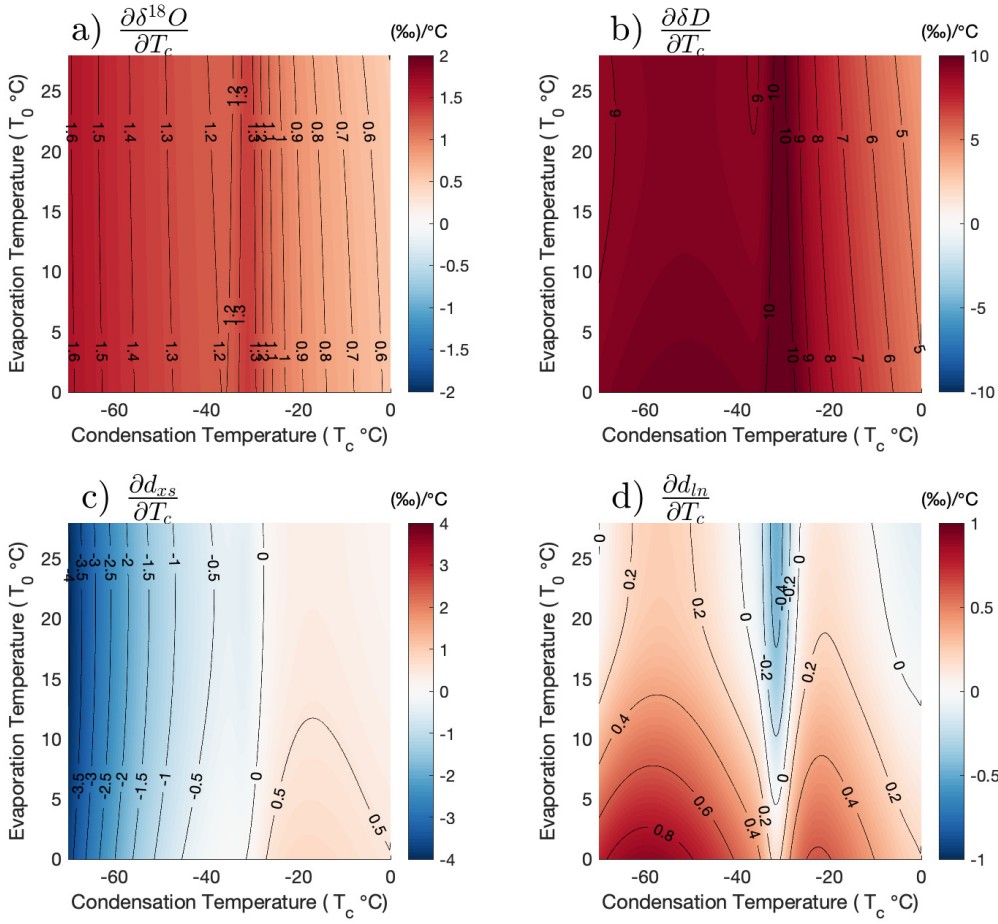

**Figure 3.** Partial derivatives of isotope state spaces with respect to condensation temperature: **a)** $\partial\delta^{18}O/\partial T_c$, **b)** $\partial\delta D/\partial T_c$, **c)** $\partial d_{xs}/\partial T_c$, **d)** $\partial d_{ln}/\partial T_c$. Shading and contours in all subplots is the slope in ‰/°C.

A "bump" in the partial derivatives of all isotope parameters with respect to the condensation temperature is seen around -30°C, arising primarily from the transition between liquid and ice condensate (Figure 3), whose relationship to temperature is prescribed in the model and based on satellite data (see Appendix A2.2). The slopes in this region also depend on the parameterization of supersaturation. This local change in partial slope is smoothed somewhat when atmospheric mixing is

5   incorporated into the model (see Appendix A5), and the changes in slopes across the parameter space are larger than these local changes.

The temperature reconstruction technique described in Section 1.1 is based on the linearization of the slopes between $\delta^{18}O$, $d_{xs}$, condensation temperature, and evaporation temperature, and assumes the $\beta$ and $\gamma$ parameters in Equations 2 and 3 are fixed over the range of reconstructed $\Delta T_{site}$ and $\Delta T_{source}$. Our results demonstrate that the assumption that $\beta$ and $\gamma$




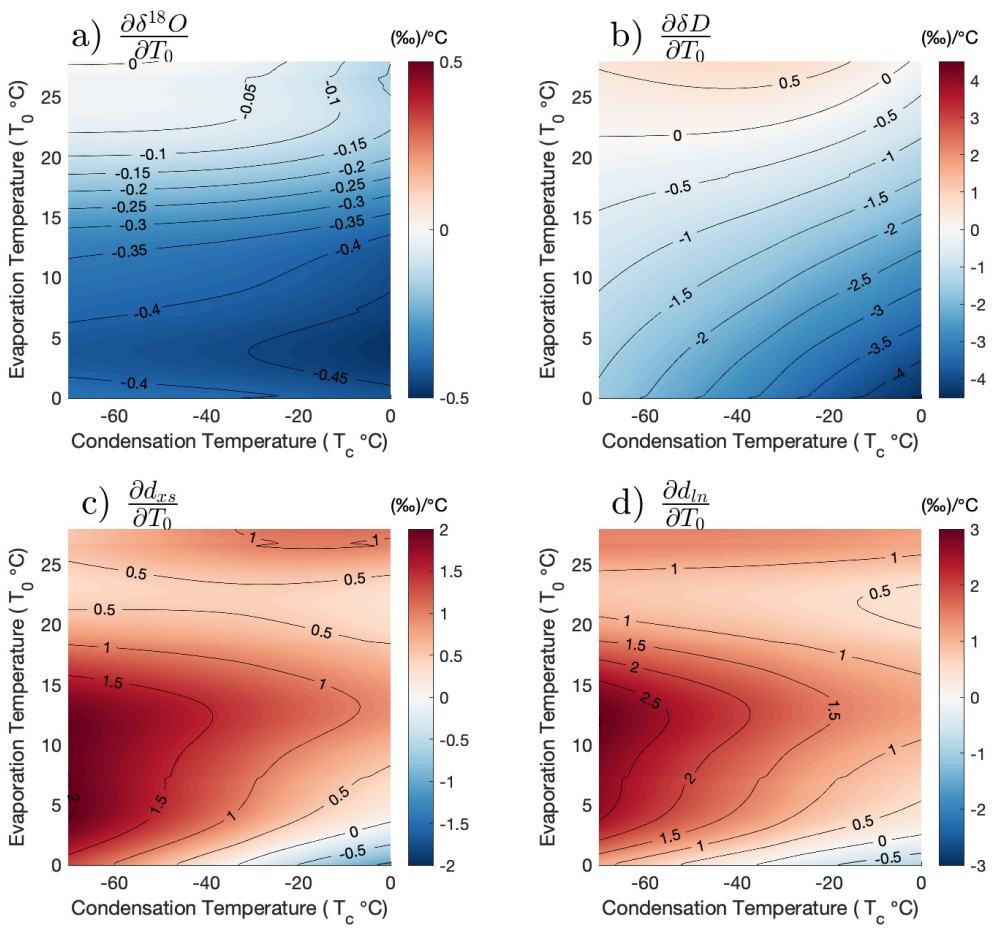

**Figure 4.** Partial derivatives of isotope state spaces with respect to evaporation temperature: **a)** $\partial\delta^{18}O/\partial T_0$, **b)** $\partial\delta D/\partial T_0$, **c)** $\partial d_{xs}/\partial T_0$, **d)** $\partial d_{ln}/\partial T_0$. Shading and contours in all subplots is the slope in $‰/°C$.

are independent of temperature (i.e. that the sensitivities are fixed and linear) is problematic. The parameters $\gamma_1$ and $\gamma_2$ in Equation 2 are comparable to the slopes $\partial\delta^{18}O/\partial T_c$ and $\partial\delta^{18}O/\partial T_0$ in Figures 3 and 4. Although the slope of $\delta^{18}O$ along the condensation temperature axis, $\partial\delta^{18}O/\partial T_c$, doesn't change dramatically, it is clearly variable, as is $\partial\delta^{18}O/\partial T_0$. The slopes $\partial d_{xs}/\partial T_c$ and $\partial d_{xs}/\partial T_0$ (comparable to $\beta_1$ and $\beta_2$ in Equation 3, respectively) are highly variable. Indeed the $d_{xs}$ surface in

5 Figure 2 has a saddle at moderate condensation temperatures, over which $\partial d_{xs}/\partial T_c$ changes sign (Figure 3.c). This shows that the assumption of constant $\beta$ and $\gamma$ parameters in isotope-based temperature reconstructions is valid only under narrow ranges of $\Delta T_{site}$ and $\Delta T_{source}$. For plausible changes in site temperatures, assuming a fixed $\gamma$, for example, may not only lead to errors in magnitude but even to errors in the sign of $\gamma$, and ultimately $\Delta T$. The use of a fixed $\gamma$ can introduce spurious variability into temperature reconstructions, particularly of $T_0$, the evaporation source temperature.



Evidence for the critical change in the sign of $\partial d_{xs}/\partial T_c$ (Figure 3.c) can actually be seen directly in Antarctic ice core records. Deuterium excess ($d_{xs}$) records from core sites whose average conditions lie on the same side of this change in slope are generally correlated over the last 60 thousand years, while sites whose average conditions lie on opposite sides of the change in slope are weakly or even anti-correlated with each other (Figure 5). Comparison of $d_{xs}$ and $d_{ln}$ between cores further demonstrates this fundamental change in slope (Figure A20).

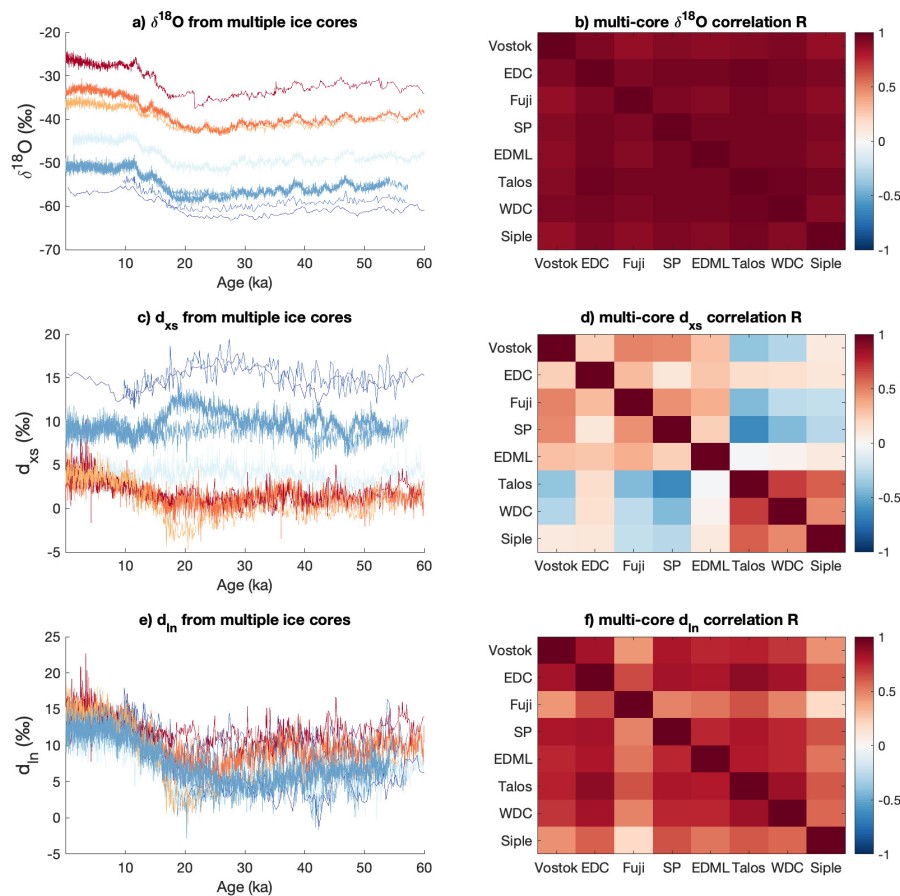

**Figure 5.** Time series and cross correlation matrices for eight different deep Antarctic ice core sites. **a, b)** $\delta^{18}$O; **c, d)** $d_{xs}$; and **e, f)** $d_{ln}$. In the time series plots, each record is colored by its Holocene average $\delta^{18}$O value. See Section 4.2 in the text for details and references for the ice core records. All original records are interpolated to even 50 yr time spacing on the Buizert et al. (2018) synchronized timescale where possible, else they are plotted on original published timescales. All records are ordered by their approximate modern surface temperature in the cross correlation matrices.



The issue of variable isotope-temperature scalings is implicit in previous work. Uemura et al. (2012) for example, following Stenni et al. (2010), used an isotope model to calculate the relevant $\beta$ and $\gamma$ parameters for several East Antarctic ice-core sites. Using the same isotope model, they calculated different scalings for each site. However, by assuming these slopes are constant for each site, they do not consider the possibility that one site's conditions may have been more like another's in the

past. Recognizing this as well as the inability of their model to simultaneously match observed site temperature and $\delta$ values, Uemura et al. (2012) create several reconstructions for the Dome Fuji site utilizing different linearizations of the model. They do not however attempt a reconstruction that accounts for the nonlinearities in the water isotope-temperature relationships.

The solution to this issue of slope nonlinearity, within the linear isotope temperature reconstruction framework (Equations 2 and 3), is not obvious. The nonlinearities in the slopes of the isotope surfaces depend on the absolute condensation and evapora-

tion temperatures, the target of the reconstruction, which are of course not known *a priori*. We next present a novel temperature reconstruction framework, which takes into account the inherent non-linearities in the water-isotope fractionation process.

## 4 Nonlinear temperature reconstructions

### 4.1 Reconstruction method

For every pair of $T_0$ and $T_c$ inputs to SWIM there is a corresponding modeled value of $\delta^{18}$O, $\delta$D, and $d_{ln}$ as shown in Figure 2.

We invert the modeled state space and project each independent temperature parameter onto a pair of dependent isotope values, e.g. $\delta^{18}$O and $d_{ln}$. This defines a set of maps, with $x$ and $y$ axes of $\delta^{18}$O and $d_{ln}$, and with $z$ axes $T_c$ and $T_0$, shown in Figure 6. To reconstruct $T_c$ and $T_0$, the inverted model results may be used as a lookup table: a pair of $\delta^{18}$O and $d_{ln}$ measurements determine a pair of $T_c$ and $T_0$ reconstructions (Figure A16). While previous reconstruction methods (e.g. Vimeux et al. (2002); Kavanaugh and Cuffey (2002); Stenni et al. (2010)) linearize the slopes calculated by a water-isotope model around the modern

climate state, this method accounts for the changes in the slopes that depend on the mean state. Further, there is no need to find analytical solutions to the model or fit families of high-order polynomials to the results.

The boundary conditions $T_c$ and $T_0$ may be projected onto axes defined by any two isotope parameters, which may then be used to reconstruct temperature. Since the only unique isotope information comes from the original $\delta^{18}$O, $\delta$D measurements ($d_{xs}$ and $d_{ln}$ being second-order parameters), any combination of those parameters may seem equally well-suited for the

purposes of temperature reconstruction. In practice, however, $\delta^{18}$O and $d_{ln}$ are the optimal pair of parameters to use for temperature reconstruction. This result is examined in more detail in the Appendix. The fundamental reason is that the logarithmic excess parameter, as a second constraint, provides an axis more orthogonal to the variability we are attempting to reconstruct. This is also the same reason $d_{ln}$ is a better qualitative proxy for source region temperature than $d_{xs}$. After proposing the $d_{ln}$ parameter, Uemura et al. (2012) suggest that there is no added value in the logarithmic parameter over the traditional linear

$d_{xs}$, in terms of the temperature reconstruction equations. While true for the linear temperature reconstruction equations, this is not the case when the nonlinearities of $\beta$ and $\gamma$ are accounted for.

Note that we could just as readily reconstruct $RH_0$ instead of $T_0$, since we assume climatological relationships between them. We reconstruct $T_0$ out of interest in the parameter from a climate dynamics perspective. In principle, our method can

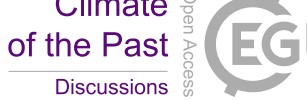

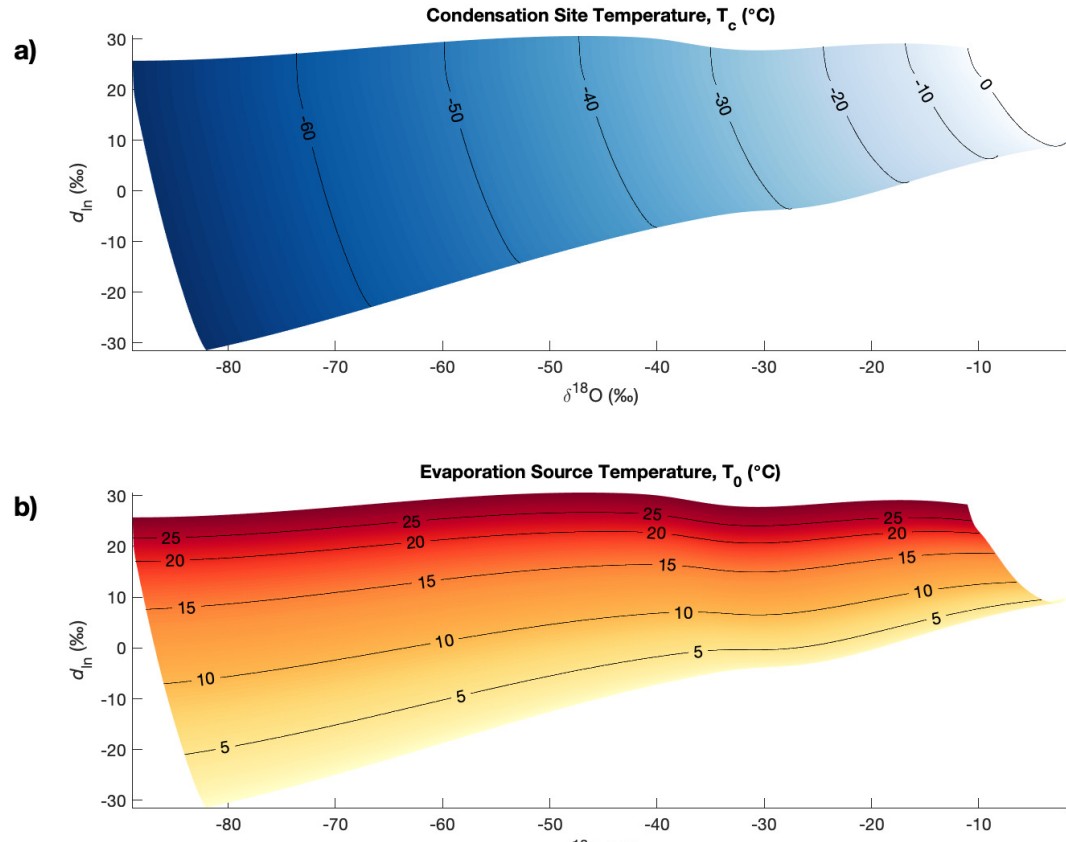

**Figure 6.** Inverted $T_0$ and $T_c$ surfaces as a function of modeled $d_{ln}$ and $\delta^{18}$O of precipitation. **a)** Surface shading and contours show condensation site temperature, $T_c$ in °C, as a function of $d_{ln}$ and $\delta^{18}$O of precipitation at a site. **b)** Surface shading and contours show evaporation source temperature, $T_0$ in °C, as a function of $d_{ln}$ and $\delta^{18}$O of precipitation at a site.

be extended to reconstruct more than two variables simultaneously, such as $T_c$, $T_0$, and $RH_0$, by modeling multi-dimensional parameter spaces. This of course requires additional measured constraints, which could include the $\delta^{17}O$ of precipitation, the accumulation rate, or other variables modeled in this framework, and is discussed in the Appendix.

### 4.2 Absolute temperature reconstructions

5   An advantage of the reconstruction technique presented here is that we are able to reconstruct absolute evaporation and condensation temperatures, not just relative variability, as in previous techniques.



There are several additional considerations important to making these reconstructions. First, we are often interested in the surface air temperature for paleoclimate studies, rather than the condensation temperature. In the Appendix A3.2 we review previous work on the relationship between surface and condensation temperature over Antarctica, and conduct novel analysis using the high resolution MERRA2 reanalysis data (Gelaro et al., 2017). We also examine the seasonality and vertical distri-

bution of Antarctic precipitation and reevaporation. Based on these analyses we use a simple, linear temperature dependent relationship to estimate weighted, annual-mean surface air temperature from our condensation temperature reconstructions and account for the uncertainty in this relationship. Similarly we examine the potential for bias from seasonality in evaporation from the ocean. We also examine the influence of non-uniqueness on our temperature reconstruction technique arising from below-freezing evaporation conditions (Appendix A8.4).

Finally, we conduct an extensive uncertainty analysis on our temperature reconstructions (Appendix A8). We calculate numerous isotope state spaces from the same temperature parameter space using multiple iterations of the model in which model assumptions are altered and parameters are varied over plausible ranges. We calculate the uncertainty in our reconstructions arising from the supersaturation parameterization, the evaporation fractionation factors, the evaporation closure assumption, the precipitation scheme, the influence of vapor mixing during transport, as well as other model choices. We use the ensemble

of isotope state spaces to estimate both the absolute and relative uncertainty in our temperature reconstructions. As an example, the central reconstructions and their uncertainties for the evaporation and condensation temperatures for the WAIS Divide ice core (WDC) are shown in Figure 7. Our reconstruction of relative temperature variability has much lower uncertainty than the the reconstruction of absolute temperature, and we find lower uncertainty in the reconstruction of condensation temperature than evaporation temperature.

We reconstruct condensation site and surface temperatures and evaporation source temperatures for eight different Antarctic deep ice-core sites for which there are $\delta^{18}O$ and $d_{ln}$ records (Figure 8). The records include WDC (Markle et al., 2017; WAIS Divide Project Members et al., 2013; Steig et al., 2013) and Siple Dome (Brook et al., 2005; Schilla, 2007) from West Antarctica, as well as the EDML (Stenni et al., 2010), EDC (Stenni et al., 2010), Vostok (Vimeux et al., 2002), Dome Fuji (Uemura et al., 2012), Talos Dome (Stenni et al., 2011), and South Pole (SP, Steig et al. (2021)) records from East Antarctica.

We correct all records for changes in the isotopic composition of seawater (Bintanja and Van de Wal, 2008), $\delta^{18}O_{sw}$, following the method outlined in Uemura et al. (2012) and Stenni et al. (2010).

### 4.3   Comparison of linear and nonlinear reconstruction techniques

#### 4.3.1   Linear temperature reconstruction using SWIM

We evaluate the significance of our approach by comparing our nonlinear reconstructions of condensation and evaporation

temperature with reconstructions following the traditional linear approach. We calculate the equivalent linear $\beta$ and $\gamma$ coefficients for each of eight ice core sites by regressing the $T_c$ and $T_0$ temperature fields to subsets of $\delta^{18}O$ and $d_{xs}$ SWIM results representative of different intervals of each core, including the Holocene (<10ka) and Last Glacial Period (20 to 30ka). We



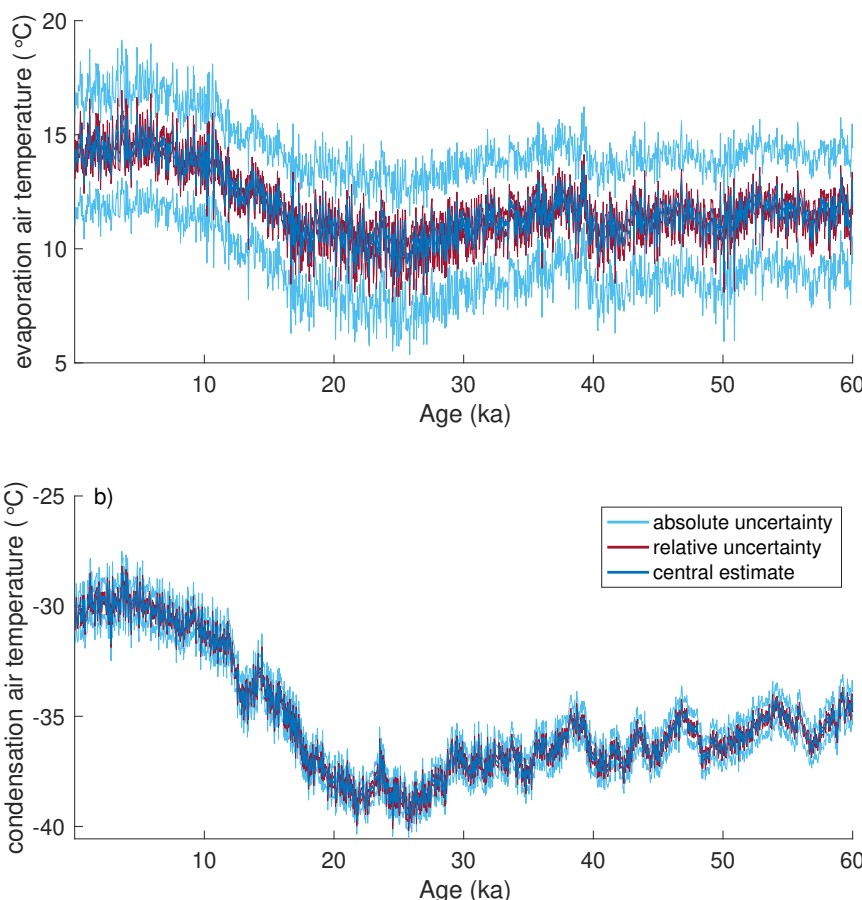

**Figure 7.** Temperature reconstructions and uncertainty estimates for the WAIS Divide ice core (WDC) site. **a)** moisture source mean evaporation temperature and **b)** ice core site condensation temperature. Dark blue lines are the central estimate, while red lines show the bounds of relative temperature uncertainty and cyan lines show the bounds of the absolute temperature uncertainty. Reconstructions resampled to even 40 year spacing.

find that the $\beta$ and $\gamma$ values significantly differ for each core and between the different time intervals, owing to the temperature dependence of the sensitivities. In particular, $\beta_2$ changes substantially between the Glacial and the Holocene.

We reconstruct relative changes in moisture-source and site surface temperature for each ice core location using the two sets of linear $\beta$ and $\gamma$ coefficients found above and the traditional linear method. We then compare these linear reconstructions to

5    our full nonlinear reconstruction and show the difference in reconstructed surface temperature and evaporation temperature in Figure 9. We also show the residuals between the two linear reconstructions. In general, the residuals are not constant offsets, but vary as a function reconstructed temperature demonstrating the temperature dependence of the slopes. They also differ,



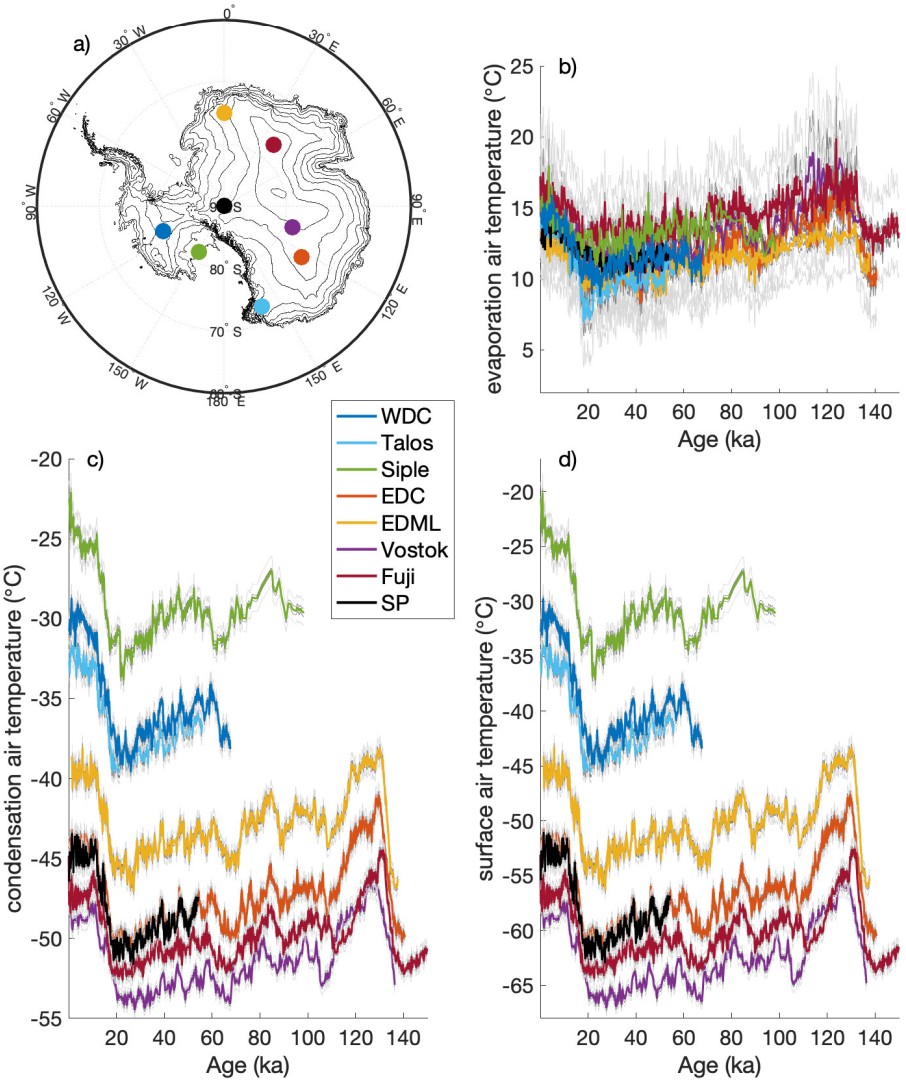

**Figure 8.** Temperature reconstructions for seven ice-core sites: WDC, EDC, EDML, Siple dome, Vostok, Dome Fuji, Talos Dome, and the South Pole ice core (SP). **a)** Ice-core site locations on the Antarctic continent. Reconstructions of **b)** moisture source evaporation air temperature, **c)** ice core site condensation air temperature, and **d)** ice core site surface air temperature. All records are resampled to even 200 year resolution for visual clarity. Thin, light grey lines in panels **b-d)** are the absolute temperature uncertainty, while thin dark grey lines show the relative temperature uncertainty.

sometimes in sign, between the ice core sites. Linearization can obscure true variability or introduce spurious variability into the reconstructions, depending on the actual conditions of the site over time.



In the case of the surface temperature reconstruction, the errors introduced by linearization can be up to $\pm 1^\circ$C depending on the core site, and are generally smaller for the colder East Antarctic sites. In the case of evaporation temperatures, the introduced errors are considerably larger, up to $\pm 2^\circ$C. Further, the total variability in reconstructed evaporation temperature is much smaller than that in ice core site surface temperature. The errors introduced into the reconstructed evaporation temperatures by ignoring the nonlinearities can be nearly as large as the total reconstructed variability. It is thus problematic to attempt reconstructing evaporation temperatures without accounting for nonlinearities.

In the Appendix A7 we show that using the nonlinear reconstruction technique yields greater correlation amongst all records of $T_s$ and especially of $T_0$ (with increases in shared variance up to 38%, Figure A18). These results suggests that linear reconstructions have obscured coherent underlying climate signals, especially in evaporation temperatures. This same reasoning supports the qualitative use of the $d_{ln}$ parameter over the linear $d_{xs}$ parameter (Figure A19; Markle et al. (2017)).

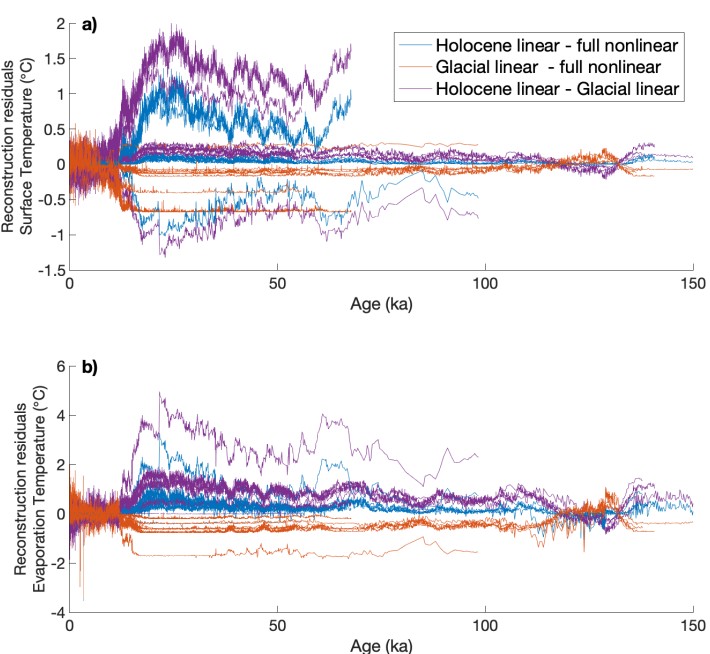

**Figure 9.** Differences between reconstructed **a)** $T_s$ and **b)** $T_0$ using different reconstruction techniques for multiple core sites. Blue lines show the difference between our full nonlinear reconstruction and a linear reconstruction using $\beta$ and $\gamma$ slopes linearized around Holocene conditions. Red lines lines show the difference between our full nonlinear reconstruction and a linear reconstruction using slopes linearized around Glacial conditions. Purple lines show the difference between the two linear reconstructions.

The relationship between $\delta^{18}$O and $T_c$ is largely linear across a wide range of values of $T_c$, regardless of evaporation temperature. The ice core site temperature reconstructions from the linear and nonlinear reconstruction techniques have relatively small differences. However, as seen in Figure 9, there are small artifacts arising from slight nonlinearity in the $\delta^{18}$O-to-$T_c$





relationship, particularly for relatively warm sites like those in West Antarctica. The primary source of this nonlinearity is the change in total fractionation factor as the air parcel transitions between liquid-only and ice-only condensate. The SWIM model retains liquid condensate at colder temperatures then previous models (e.g. Kavanaugh and Cuffey (2002)) in line with satellite measurements (Hu et al., 2010). The resulting transition of fractionation factors drive the nonlinearities in the $\delta^{18}$O-to-$T_c$

relationship at temperatures relevant to West Antarctica, and ultimately resulting in larger differences between the linear and non-linear reconstruction techniques at those sites compared to cores from East Antarctica. Because our model uses a consistent supersaturation parameterization in the model's isotope and precipitation schemes, the relationship between $\delta^{18}$O and $T_c$ is actually more linear in SWIM than in other comparable models.

In the Appendix A9 we compare our temperature reconstructions for several East Antarctic ice cores to previously published

reconstructions using the linear technique with coefficients estimated from different water isotope models (Stenni et al. (2004), Stenni et al. (2010), Uemura et al. (2012)). The differences between our reconstructions and previous reconstructions arise both from differences in the underlying isotope models as well as the reconstruction techniques. In general, the previously-published linear reconstructions overestimate changes in both site and source temperature compared to our nonlinear reconstructions. For example, we find smaller values of glacial to interglacial temperature change for most East Antarctic sites than previous

reconstructions.

## 5   Discussion

Using the self consistent, nonlinear temperature reconstruction technique for eight different ice core sites, we next investigate the patterns of Southern Hemisphere temperature change through time. In Figure 8 we show the nonlinear reconstructions of Antarctic surface temperature and moisture-source evaporation temperature for the eight ice-core records. At the WDC

site in West Antarctica there is an independent estimate for the magnitude of glacial-interglacial temperature change from the borehole temperature profile (Cuffey et al., 2016). Our results are in good agreement with those findings, both in the absolute value of reconstructed temperatures as well as the magnitude of glacial-interglacial change. Cuffey et al. (2016) find 11.3±1.8°C warming at WDC during the deglaciation; we reconstruct 11.2±0.5°C of warming (calculated as the difference between the average surface temperature 27-24 ka and 11-9 ka, for direct comparison to Cuffey et al. (2016)). Using an

independent temperature reconstruction technique for the South Pole ice core, Kahle et al. (in review) find and interglacial site temperature warming of 8.12±0.96 °C, between 19.5-22.5 ka and 0.5-2.5 ka. Our reconstruction yields a site temperature warming of 8.9±0.4 °C, for the same interval.

We create a stack of each reconstructed temperature variable (the evaporation temperature $T_0$, condensation temperature $T_c$, and surface temperature $T_s$) for all eight ice core records (Figure 10). We weight the records equally; we do not adjust for the

spatial distribution of the cores, nor weight by area, latitude, or elevation.

In Figure 10 one can see that the Antarctic-wide average surface temperature change during the last deglaciation was considerably larger than the concurrent temperature change in the mean moisture evaporation source. In fact, average deglacial change

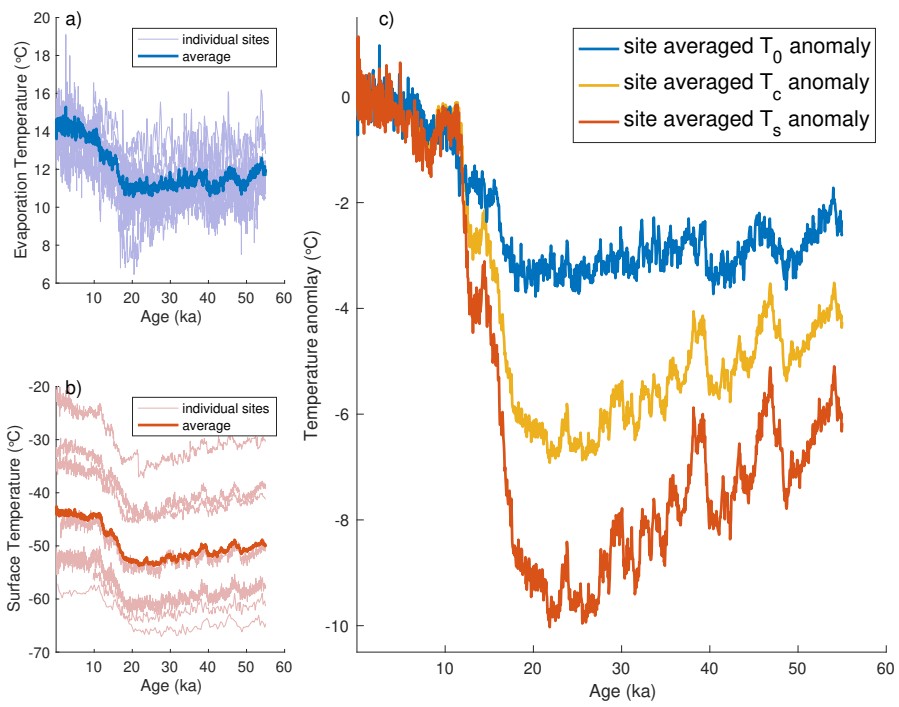

**Figure 10.** Antarctic-wide stacks of reconstructed temperature histories. **a)** Moisture source evaporation temperatures, $T_0$, of all ice core sites are shown in light blue while the average of all records is shown in dark blue. **b)** All reconstructions of ice core site surface temperatures, $T_s$, are shown in light red while the continent-wide average is shown in dark red. **c)** Anomalies of site averaged evaporation temperature (blue), condensation temperature (gold), and surface temperature (red) are shown with respect to the mean value of the most recent 2,000 years. All records are interpolated to even 50 year spacing. Where possible records are on the synchronized Buizert et al. (2018) timescale.

in Antarctic surface temperature was about three times as large as the changes in evaporation temperatures, while changes in condensation temperature were about twice as large as the evaporation temperature changes.

   In Figure 11 we show the pattern of glacial-interglacial temperature change across the Antarctic continent. The magnitude of warming since the Last Glacial Maximum is calculated as the temperature difference between the Late Holocene (LH, 0-4

5  ka) and Last Glacial Maximum (19-23 ka), for comparison with other proxy reconstructions. There may be some uncertainty in the relative magnitudes of these changes owing to offsets in the individual timescales of each record. While the relative magnitudes of interglacial change depend on the exact time periods used in the differencing, the pattern of changes in surface temperature across the continent is robust. We make no corrections to the records for elevation changes or ice flow, lacking sufficient constraints for all records. These effects are likely small in East Antarctica (Stenni et al., 2011), and smaller still in

10  West Antarctica (WAIS Divide Project Members et al., 2013).





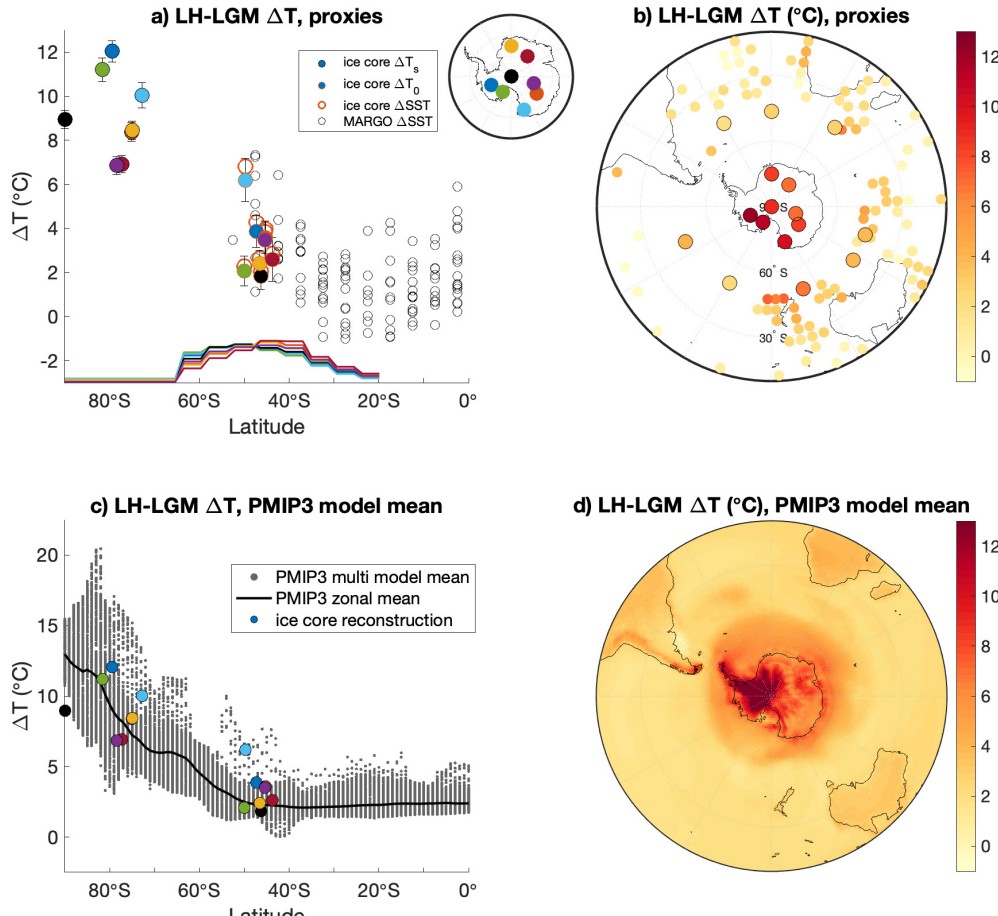

**Figure 11.** Spatial pattern of temperature change since the Last Glacial Maximum. **a)** Warming between 19-23 ka and 0-4 ka for ice core site surface temperatures (colored circles with black outline corresponding to different ice core sites as shown in map inset, uncertainty in the temperature change shown as error bars), moisture source evaporation air temperature (colored circles, red outline and uncertainty), and moisture source sea surface temperature (red circles). Moisture source warming is plotted at the mean latitude of the moisture source distributions for each ice core site, while the latitudinal extent of each moisture source is indicated by the relative histograms along the x-axis. Sea surface temperature warming form the MARGO compilation of SST estimates from ocean sediment cores are shown in open black circles. **b)** Ice core site surface temperature changes and moisture source sea surface temperature changes shown in large colored circles with black outline. MARGO compilation SST changes shown in small colored circles. **c & d)** Spatial pattern of temperature change from the multi-model mean PMIP3 simulations of the LGM and pre-industrial. **c)** Multi model mean for all grid points shown in grey dots with the zonal mean in black. Estimates of $\Delta T_s$ and $\Delta T_0$ from the ice core reconstructions are shown in colored circles as in **a)** for reference.





We find smaller glacial-interglacial temperature change for East Antarctic sites compared to previous reconstructions. Our results show that the surface temperatures of the lower, warmer areas of West Antarctica warmed significantly more than the higher, colder East Antarctic since the LGM. For example the average surface temperature warming between the LGM and LH for the two lowest sites, WDC and Siple Dome, is 11.6°C. The average warming at the two highest sites however, Dome Fuji

and Vostok, is significantly less, just 6.9°C or 59% of that at the lower sites.

In Figure 11 we plot the magnitude of warming since the LGM of Antarctic moisture source evaporation air temperatures for all ice-core records as a function of the mean latitude of the moisture source distribution for each site (based on water-tagged GCM simulations, see Appendix). Additionally we calculate the change in sea surface temperature (SST) during evaporation (red circles Figure 11.a), using the $T_0 - SST$ relationship from our model, for comparison to other SST proxy reconstructions.

While plotted as points, note that these changes in moisture source temperature reflect the integrated warming over the moisture source area. Moisture source distributions for each site are indicated by relative histograms along the latitude axis in Figure 11.a (see Figure A8 for further information), colored to correspond to the ice core site. The moisture source points are plotted at the longitude of the respective ice core sites in Figure 11.b, though in reality the MSDs have significant meridional extent (c.f. Extended Data Figure 8, Buizert et al. (2018)), often asymmetrically to the west owing to the westerly winds. Changes

in $T_0$ for Antarctic moisture sources may reflect both warming SSTs at fixed locations as well as potential changes in the mean latitude of the moisture source distributions, for example due to changes atmospheric circulation e.g. a meridional shift in the mean westerly winds (Markle et al., 2017). Disentangling these two influences requires additional constraints and is beyond the scope of this study. While the ice core $T_0$ reconstructions have low spatial resolution owing to broad moisture source distributions, they benefit from the temporal resolution and precision of the ice core age scales, compared to other proxy

records of temperature from the Southern Hemisphere mid latitudes.

It is clear from Figure 11 that the Antarctic continent warmed two to three times as much as the Southern Hemisphere mid-latitude moisture source areas since the Last Glacial Maximum. This result is in line with other paleoclimate reconstructions, as well as modeling of the pattern of polar amplification since the LGM (Masson-Delmotte et al., 2006; Otto-Bliesner et al., 2006). In particular our estimates of moisture source region changes agree with completely independent estimates from the MARGO

compilation of SST changes (Members et al. (2009), open circles Figure 11). There appears to be some zonal asymmetry in the warming of Southern Ocean surface temperatures in both our moisture source reconstructions and the MARGO compilation. The waters around New Zealand and Australia that comprise the moisture source of Talos Dome appear to show the most warming since the LGM.

These patterns of Southern Hemisphere warming are also in reasonable agreement with modeling expectations, e.g. from the

Paleoclimate Model Inter-comparison Project (PMIP3, Braconnot et al. (2012)). The multi-model mean pattern of Southern Hemisphere polar amplification from PIMP3 simulations is shown in Figure 11. There is broad similarity to our reconstructions, though there are important differences as well. The spread in temperature change about the zonal mean over both the Antarctic and ocean surface is similar between the model and the reconstructions. Our reconstructions show more warming in the ice core moisture source areas equator-ward of the polar front than the PMIP3 mean, and less warming over the surface of West

Antarctica. We note that the magnitude and pattern of modeled Antarctic surface warming is predominately a function of





imposed changes in ice sheet surface elevation to PMIP3 experiments. The extreme warming seen in parts of West Antarctica in the PMIP3 model mean (e.g. > 20 °C), outside the range found in our reconstructions, likely reflect unrealistically large ice sheet thickness changes prescribed in PIMP3. Understanding the full set of processes responsible for the reconstructed pattern of Southern Hemisphere polar amplification is a topic for future work.

## 6  Conclusions

Ice-core records of the stable isotopes of water provide detailed histories of Earth's climate. Both qualitative and quantitative interpretation of these records requires understanding the relationships between fractionation processes and environmental conditions.

Qualitatively, $\delta^{18}$O and $\delta$D are reliable indicators of relative change in condensation temperature, over a sufficiently long timescale. The assumption of a roughly-linear relationship is generally justified, as shown in this study and previously. However, the linear definition of deuterium excess, $d_{xs}$, is an unreliable indicator of relative evaporation-site temperature change, particularly at East Antarctic sites with very depleted $\delta^{18}$O and $\delta$D values. In these cases, the logarithmic definition of the parameter, $d_{ln}$, is a more faithful qualitative proxy of evaporation temperature.

We can use models to make quantitative interpretations of water-isotope variability and to disentangle the combined influences of the source and site temperatures. To date, most water-isotope temperature inversions have assumed linear relationships (Kavanaugh and Cuffey, 2003; Vimeux et al., 2002; Stenni et al., 2011; Uemura et al., 2012). However, as shown here, this assumption is flawed. Even in the simplified water-isotope models that underly most temperature reconstructions, there are inherent nonlinearities in the isotope-temperature relationships. Ignoring these nonlinearities distorts reconstructed temperature variability. In the case of evaporation source temperature changes, these distortions may be a significant fraction of the total reconstructed variability.

There is a long standing debate regarding the interpretation of "spatial" and "temporal" slopes in the water isotope-temperature relationship (e.g. Jouzel et al. (1997)). These discussions are conceptually useful. However, while space and time are obvious coordinates through which to understand climate, they are not the most relevant for water-isotope fractionation. Neither space nor time can independently cause water to change phase and fractionate.

The fundamental dimension through which to understand water isotope fractionation is temperature. In this study we use a relatively simple model to investigate the relationships of water isotopes in precipitation to temperature. While the distinction between temporal and spatial slopes is not considered in this context, we are able to resolve the core question: is the water isotope-temperature relationship fixed? It is not.

Our nonlinear reconstruction technique allows for the estimation of absolute temperatures in the past, in addition to their variability, and is corroborated by independent temperature constraints. By taking into account the inherent nonlinearities of water isotope fractionation we are better able to constrain evaporation source region changes. Our reconstructions reveal a spatial pattern of temperature change across the Antarctic continent in which West Antarctica warmed significantly more than

East Antarctica since the Last Glacial Maximum. Further our reconstructions provide insight into the spatial pattern of polar amplification, suggesting that the warming since the LGM in Antarctica was two to three times that in the mid latitudes.

*Code and data availability.* The temperature reconstructions, underlying data and Simple Water Isotope Model code will be made available through the United States Antarctic Research Program (USAP) Data Center upon publication. The ice core data are all already available
through links provided in the primary papers cited for each data set.

*Author contributions.* BRM conceived of the study and with EJS designed the model and analyses. BRM and EJS wrote the manuscript.

*Competing interests.* The authors declare no competing interests.

*Acknowledgements.* This work was supported in part by grant numbers 1043092, 1141839, and 1443105 from the United States National Science Foundation. BRM was supported in part by the Stanback Postdoctoral Fellowship. The authors wish to thank H.C. Steen-Larsen,
Spruce Schoenemann, Emma Kahle, Vasileios Gkinis, Marina Dütsch, and Peter Blossey for helpful discussions.

**Appendix A: Simple Water Isotope Model**

The Simple Water Isotope Model (SWIM) is based on existing numerical Rayleigh-type distillation models (Merlivat and Jouzel, 1979; Jouzel and Merlivat, 1984; Ciais and Jouzel, 1994; Criss, 1999; Kavanaugh and Cuffey, 2003), though we make several important improvements and updates. We first describe the physical environmental aspects of the model and then the
details of the fractionation scheme.

**A1   Environmental trajectory**

Our model considers moisture transported from evaporative sources down an atmospheric temperature gradient (i.e. from the midlatitudes toward the pole), driving condensation and fractionation. Our model operates in the dimension of temperature; we consider pseudo-adiabatic temperature pathways from an initial surface air temperature, $T_0$, to a final condensation temperature,
$T_c$, and discrete steps $dT$, and Euler numerics.

**A1.1   Source-region conditions**

The moisture-source surface air temperature ($T_a$), sea surface temperature ($SST$), and relative humidity ($RH$) influence the fractionation of vapor evaporating from the ocean. We use modern climatological correlations to find initial values of $SST_0$ and $RH_0$ given a specified initial air temperature, $T_0$, using the 1980-2010 annual mean climatological fields from the





NCEP/NCAR reanalysis project (Kalnay et al., 1996) and the ERA-Interim reanalysis (Dee et al., 2011). Correlations with surface air temperature are better defined than spatial correlations and give greater flexibility to the model for use in different climate states. Surface air temperature and $SST$ are extremely well-correlated in the reanalysis (Figure A1) with a well-defined near-linear relationship over most of the temperature range, except where the SSTs asymptote to the freezing point of seawater.

The relationships are nearly identical between the NCEP and ERA reanalysis products.

Relative humidity gradients in the modern climate are fairly weak, though surface $RH$ over the ocean is consistently higher at lower surface temperatures on climatological timescales. While variable on short timescales, $RH$ appears largely invariant on timescales longer than interannual (Dai, 2006; Vimeux et al., 2002). We find that the over-ocean, surface relative humidity is systematically about 5% higher in the NCEP reanalysis compared to the ERA reanalysis, though the relationship to surface

temperature is similar.

Given a specified initial air temperature, $T_0$, our model uses values of $SST_0$ and $RH_0$ based on fits to the modern climatology. We use three methods to calculate the climatological relationships over the interval -10°$\leq T_a \leq$ 28°C: a cubic spline with specified noise tolerance; the mean and median of $SST$ and $RH$ distributions within binned values of $T_a$; and high-order polynomial fits. All methods show effectively indistinguishable relationships in both reanalysis products. We calculate the un-

certainty in these fits and test the model's sensitivity. Our base model uses the cubic spline method which is least susceptible to edge effects, phase shifting, and maintains a smooth first derivative.

We find some differences in the $T_a$-to-$SST$ fit between the Northern and Southern Hemispheres for air temperatures between 5° and -15°C, as seen in Figure A1. In this study we will use the Southern Hemisphere fit for $T_a$-to-$SST$. We find no major hemispheric differences for the $T_a$-to-$RH$ fit, and find little impact of zonal asymmetry on either fit. We find relatively small

differences in the fit between $T_a$ and $SST_0$ for different seasons and somewhat larger seasonal changes in the $T_a$ and $RH_0$ relationship.We use the annual average fits hereafter and test the sensitivity of the water-isotope values of evaporation to these seasonal differences.

The normalized relative humidity, $RH_n$, is critical to kinetic fractionation of water isotopes during evaporation (Merlivat and Jouzel, 1979; Risi et al., 2010) and depends on the three variables above:

$$RH_n = \frac{RH \times e_s(T_a)}{e_s(SST)} \tag{A1}$$

where $e_s(T_a)$ and $e_s(SST)$ are the saturated vapor pressures of air at the surface air temperature and at the sea surface temperature, respectively.

### A1.2  Transport

After evaporation at initial air temperature, $T_0$, and specified surface pressure, $P_0$, moisture is transported toward the pole in

isolation, cooling and condensing along the way. The air parcel is cooled pseudoadiabatically defining a pressure trajectory, $P$ as a function of temperature, $T$ (Figure 1). As the air parcel cools, moisture above saturation is removed and the latent heat released during the phase change keeps the air parcel warmer than in an otherwise equivalent isobaric pathway. Following the





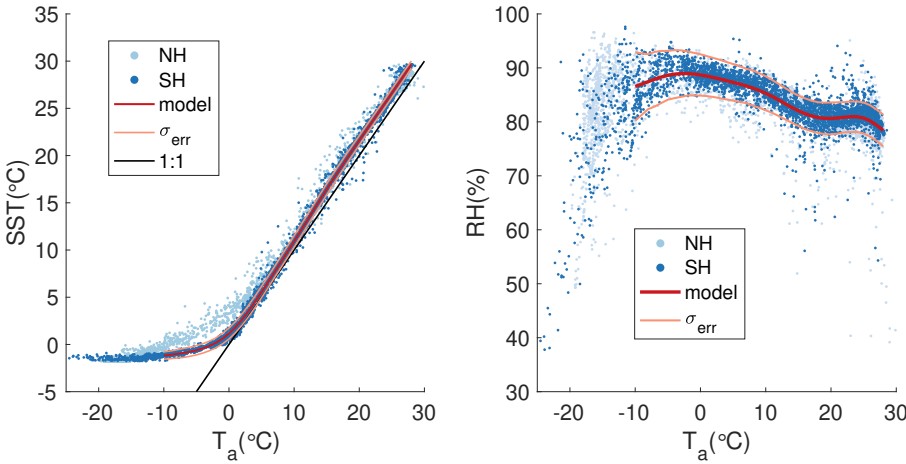

**Figure A1.** Climatological correlations between $T_a$, $SST$, and $RH$. Left: Annual mean climatological surface air temperature, $T_a$, and sea surface temperature, $SST$, from the NCEP/NCAR reanalysis (Kalnay et al., 1996). Light blue dots are Northern Hemisphere (NH) grid points, while dark blue dots are Southern Hemisphere (SH) grid points. The polynomial fit for the SH is in red. The error estimate on the fit, $\sigma_{err}$ in orange, is the standard deviation of the misfit in the model. The 1:1 line is shown in black. Right: same as on the left but for the surface air temperature over ocean, $T_a$, and surface relative humidity (RH) over oceans.

pseudoadiabatic assumption, we consider no other heat sources to the air parcel and moisture is removed immediately after condensation. Below we investigate in the influence of air-parcel mixing on our model framework, which is a relaxation of the adiabatic assumption.

We calculate a pseudoadiabat following the iterative routine described in Bakhshaii and Stull (2013) but taking into account the saturated vapor pressures of both ice and liquid water condensate. The temperature dependent saturated vapor pressures of ice and liquid water (Murray, 1966; Murphy and Koop, 2005), together with air pressure $P(T)$ define saturated mixing ratios for ice and liquid water,

$$r_s = \frac{R_d}{R_{wv}} \times \frac{e_s}{P - e_s} \tag{A2}$$

as functions of $T$, where $R_d/R_{wv}$ is the ratio of gas constants of dry air and water vapor.

We consider air parcels with mixed ice and liquid condensate (Ciais and Jouzel, 1994), in which the ice fraction smoothly increases as temperatures decreases below freezing. Many models, including isotope-enabled GCMs, approximate the temperature dependence of cloud ice-liquid fraction as piecewise linear functions (Hu et al., 2010), while others use smoothly varying error integrals (Ciais and Jouzel, 1994; Kavanaugh and Cuffey, 2003). We use temperature-dependent functions for the cloud ice fraction derived from satellite observations (Hu et al., 2010) over the Southern Ocean and over land snow and ice surfaces, which preserve significantly more liquid water at colder temperatures than previous parameterizations (e.g. Kavanaugh and Cuffey (2003)).





The specific heat at constant pressure, $c_p$, latent heat, $L$, saturated vapor pressure, $e_s$, and saturated mixing ratio, $r_s$, are temperature-dependent and calculated for the liquid and ice phases individually. Effective values for the parcel as a whole are calculated from the mixing fractions of each phase (Kavanaugh and Cuffey, 2003). For example, $r_{s(eff)} = r_{s(ice)}F_{(ice)} + r_{s(liq)}F_{(liq)}$, where $r_{s(ice)}$ and $r_{s(liq)}$, are the saturated mixing ratios of ice and liquid, respectively, and $F_{(ice)}$ and $F_{(liq)}$ are

the (temperature dependent) fractions of each phase of condensate.

Moisture is removed along the temperature pathway owing to the temperature-dependent changes in the saturated mixing ratio, $-dq/dT = dr_{s(eff)}/dT$. This is a simplified view of large-scale precipitation, commonly used in similar models (e.g. Markle et al. (2018)). We do not consider reevaporation of falling precipitation. There are several reasonable choices in the implementation of our simplified view of moisture removal, once air parcels are cooled from initial relative humidity to saturation. The

instantaneous moisture removal process may leave the air parcel at saturation, or at some specified level below saturation (e.g. RH = 90%), or at the air parcel's initial level below saturation, in which case relative humidity is constant along the path. We test the sensitivity of our model to these assumptions below, using constant relative humidity as our default.

At very cold temperatures moisture is removed not at saturation but at a specified level of supersaturation. The presence of both ice and liquid condensate in the cloud dictates a supersaturation of vapor over ice due to the difference in liquid and ice

vapor pressures (Jouzel and Merlivat, 1984). A paucity of condensation nuclei may lead to further supersaturation at very cold temperatures (Tegen and Fung, 1994). The total supersaturation is parameterized here to depend on temperature (discussed in detail in Section A2.3).

## A2   Isotope fractionation

In this section we outline the water-isotope fractionation scheme used in SWIM. We model equilibrium and kinetic fractiona-

tion of the $^2H/^1H$, $^{18}O/^{16}O$, and $^{17}O/^{16}O$ ratios in water. We use conventional notation in which $R$ is the number ratio of heavy to light isotopes of a species, for example $^DR = {}^2H/^1H$ and $^{18}R = {}^{18}O/^{16}O$.

The fractionation factor is the ratio of $R$ values between phases. For example, the fractionation factor between liquid and vapor phases for $\delta^{18}O$ is:

$$^{18}\alpha_{l-v} = \frac{\left(\frac{^{18}O}{^{16}O}\right)_{liquid}}{\left(\frac{^{18}O}{^{16}O}\right)_{vapor}} = \frac{^{18}R_l}{^{18}R_v} \tag{A3}$$

We use the empirically-determined, temperature-dependent equilibrium fractionation factors between liquid and vapor, $^{18}\alpha_{eq(l-v)}$ and $^D\alpha_{eq(l-v)}$, as well as those between vapor and ice, $^{18}\alpha_{eq(i-v)}$ and $^D\alpha_{eq(i-v)}$, (Majoube, 1970, 1971; Merlivat and Nief, 1967; Criss, 1999) with updates for the ice-vapor equilibrium fractionation factor found by Lamb et al. (2017).





### A2.1 Evaporation from the ocean

The isotopic values of vapor evaporating from the ocean are determined, in part, by the isotopic values of the seawater. By definition, globally averaged seawater has $\delta$ values near 0‰. However the $\delta^{18}O$ of seawater ($\delta^{18}O_{sw}$) in the regions of Antarctic moisture sources is more depleted than average, with a mean around -0.3‰, (Schmidt et al., 1999). We use the observed

correlation between $\delta^{18}O_{sw}$ and $\delta D_{sw}$ from a compilation of global seawater measurements (Schmidt et al., 1999) to find an initial $\delta D_{sw}$ given the specified initial $\delta^{18}O_{sw}$ (which changes with mean climate) and investigate the sensitivity of the model to these initial conditions. We assume a $^{17}O_{xs}$ of sea water equal to 0, where $^{17}O_{xs} = \delta'^{17}O - 0.528 \times \delta'^{18}O$, and is typically reported in per meg.

The atmosphere above the global oceans is not at saturation on average, with relative humidity typically around 80% (Hart-
mann, 2015). Because of this steady-state disequilibrium, significant kinetic fractionation occurs during evaporation from the ocean. Kinetic fractionation depends both on the relative humidity and the wind speed at the air-ocean interface during evaporation (Merlivat and Jouzel, 1979). The effective fractionation factor associated with diffusion and turbulence is,

$$\alpha_{diff} = \left(\frac{D}{D^*}\right)^n \tag{A4}$$

where $D$ and $D^*$ are the diffusivities of the light and heavy isotopes, respectively (Merlivat and Jouzel, 1979). The exponent
$n$ ranges from 0 to 1 and relates to the wind regime and speed, and the ratio of turbulent to molecular diffusion. For the diffusive fractionation between $H_2^{18}O$ and $H_2^{16}O$ during initial evaporation, the fractionation factor $^{18}\alpha_{diff}$ equals 1.0 for pure turbulence and 1.0028 for pure molecular diffusion (Merlivat and Jouzel, 1979; Barkan and Luz, 2007).

Following Kavanaugh and Cuffey (2003), we do not explicitly consider surface wind speeds. Instead we use the empirical results of Uemura et al. (2008) and Uemura et al. (2010) for $^{18}\alpha_{diff}$, who estimate the parameter based on measurements of
$\delta D$, $\delta^{18}O$, and $\delta^{17}O$ in vapor above the Southern Ocean. Uemura et al. (2010) find a value of $^{18}\alpha_{diff} = 1.007 \pm 0.0013$ and $1.008 \pm 0.0018$, when optimizing for observations of $d_{xs}$ and $^{17}O_{xs}$ of vapor, respectively. These results are within uncertainty of each other and of independent analysis by Pfahl and Wernli (2009) which found a value of 1.0076. Using a complication of vapor measurements (including Uemura et al. (2008), Uemura et al. (2010), Liu et al. (2014), Kurita et al. (2016), and Benetti et al. (2017)), we find that $^{18}\alpha_{diff} = 1.009$ leads to a good match between modeled and observed values of both $d_{xs}$ and
$^{17}O_{xs}$ of vapor when using the observed values of $T_0$, $SST$, and $RH$ at the time of the vapor measurements. We investigate the sensitivity of the model to this parameter below.

The diffusive fractionation factor between hydrogen and deuterium, $^D\alpha_{diff}$ may be determined experimentally by measuring the ratio of diffusive fractionation factors (Merlivat, 1978; Luz et al., 2009):

$$\phi_{diff} = \frac{^D\alpha_{diff} - 1}{^{18}\alpha_{diff} - 1} \tag{A5}$$

Merlivat (1978) found a mean value for $\phi_{diff}$ of 0.88 based on laboratory evaporation studies, and Luz et al. (2009) found that the value of $\phi_{diff}$ depends on the evaporation temperature, ranging between 0.73 and 1.06 for temperatures between 10°C and 69.5°C. We use a piecewise linear function based on the results of Luz et al. (2009) to relate $\phi_{diff}$ and evaporation temperature, and thus $^D\alpha_{diff}$ to $^{18}\alpha_{diff}$. For evaporation temperatures colder than the experimental range of Luz et al. (2009) ($< 10°$





C), we use the measured value at $10°C$ ($\phi_{diff} = 1.06$). The differences in model results for evaporation with a temperature dependent $\phi_{diff}$ and a constant $\phi_{diff} = 0.88$, are small ($< 1‰$ for initial $\delta D$ of vapor).

For the fractionation of $^{17}O/^{16}O$ we use the following relationships, which are backed both by theory and empirical observation (Barkan and Luz, 2005, 2007): $^{17}\alpha_{eq} = ^{18}\alpha_{eq}^{0.529}$ for vapor and liquid in equilibrium, and $^{17}\alpha_{diff} = ^{18}\alpha_{diff}^{0.518}$ for vapor
diffusion.

The relationship between the initial $R$ value of the vapor and the ocean due to kinetic fractionation depends on the normalized relative humidity during evaporation, $RH_n$, the equilibrium and diffusive fractionation factors, $\alpha_{eq(l-v)}$ and $\alpha_{diff}$. Following Criss (1999) and Luz et al. (2009),

$$\alpha_{evap} = \frac{R_o}{R_e} = \frac{\alpha_{eq}\alpha_{diff}(1 - RH_n)}{1 - \alpha_{eq}RH_n\left(R_v/R_o\right)} \tag{A6}$$

where $R_o$ and $R_v$ are the isotopic ratios of the ocean water and the water vapor in the atmospheric boundary layer, respectively. $R_e$ is the ratio of the evaporate, the net vapor lost to the atmosphere, a quantity that is not directly-measurable (Criss, 1999).

If we assume that the only source of vapor to the boundary layer is the local evaporate, we may equate $R_v$ and $R_e$ and solve Equation A6 for $R_v$ (Merlivat and Jouzel, 1979; Criss, 1999; Risi et al., 2010):

$$R_v = \frac{Ro}{\alpha_{eq} \times (\alpha_{diff} + RH_n(1 - \alpha_{diff}))} \tag{A7}$$

The modeled isotopic composition of vapor evaporated from the ocean is shown in Figure A2. This "local" closure assumption is within the range of observations of water isotopes in vapor over the Southern Ocean and elsewhere (Uemura et al., 2008, 2010; Liu et al., 2014; Kurita et al., 2016; Benetti et al., 2017). However, the validity of the local closure assumption under certain conditions is in question (Uemura et al., 2010; Risi et al., 2010). In addition to moisture from the ocean surface, the boundary layer may receive moisture from advection, convection, subsidence, and reevaporation of precipitation. Risi et al.
(2010) explored this issue extensively, using a model that takes into account these other sources of moisture. They show that the local closure assumption leads to vapor that is too enriched in both $\delta D$ and $\delta^{18}O$, and too low in $d_{xs}$, and that these offsets are a function of environmental conditions (Risi et al., 2010).

We investigate the influence of the closure assumption in a few ways. First, we examine closure globally rather than locally. The mean ocean has $\delta$ values of about $0‰$, and global average precipitation has $\delta^{18}O = -4.5‰$ and $\delta D = -26.7‰$, (Craig
and Gordon, 1965). Following Criss (1999) and considering the global average steady state in which the ocean is the ultimate vapor source for precipitation, the delta values of precipitation must reflect the net loss by evaporate from the ocean. Thus globally, the ratio $R_o/R_e$ is 1.0267 for D and 1.0045 for $^{18}O$. Instead of equating $R_e$ and $R_v$ locally, we define a global $\overline{\alpha}_{evap} = (R_o/R_e)_{global}$, and solve for $R_v$. Substitution into Equation A6 and rearranging leads to:

$$R_v = R_o\left(1 - \frac{\alpha_{eq}\alpha_{diff}(1 - RH_n)}{\overline{\alpha}_{evap}}\right)(\alpha_{eq}RH_n)^{-1} \tag{A8}$$

Though an obviously blunt approach for determining local evaporation, this "global closure" assumption is the limit for a globally-mixed atmosphere. Modeled evaporation using both closure assumptions is compared to isotopic measurements of Southern Ocean vapor in Figure A2.



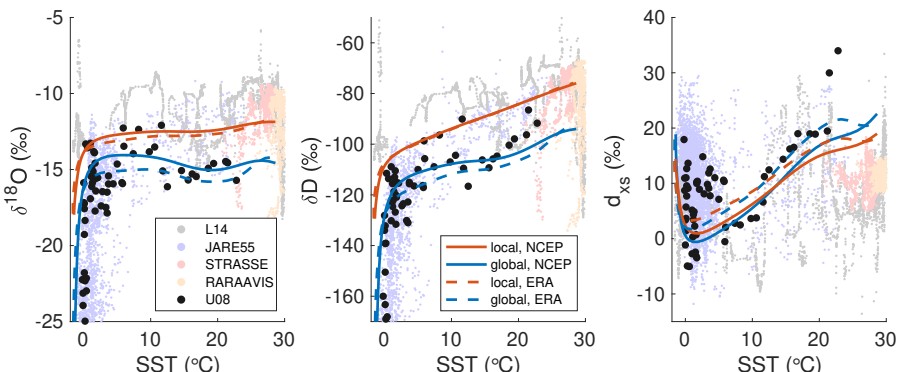

**Figure A2.** Modeled isotopic values of evaporation compared to Southern Ocean vapor measurements. a) Modeled $\delta^{18}O$ versus $SST$ using the "local" closure assumption (red lines) and "global" closure assumption (blue lines). We use two different reanalysis data sets for the $SST$ and $RH$ climatology, NCEP/NCAR (solid lines) and ERA Interim (dashed lines). Black dots are discrete vapor measurements from the Southern Ocean made by Uemura et al. (2008) (U08), while grey, blue, red and yellow dots are continuous ship-based Southern Ocean vapor measurements made by Liu et al. (2014) (L14), Kurita et al. (2016) (JARE55), and Benetti et al. (2017) (STRASSE, RARA AVIS), respectively. b) Same as a) but for modeled $\delta D$ versus $SST$. c) Same as a) but for modeled $d_{xs}$ versus $SST$. The vertical tails at low SST in panels a) and b) result from SSTs asymptoting to the freezing point of seawater while air temperatures may continue to decrease.

Next, we consider specific mixing of evaporative conditions instead of the generalized globally-mixed case above. Rather than the local ocean being the only source of vapor, we can consider a simple scenario in which the isotopic composition of the boundary layer, $R_v$, is comprised of both local evaporate, $R_e$, and vapor evaporated at some distal location and advected to the site, $R_v = (1-\theta)R_e + \theta R_{distal}$, where $\theta$ is the fraction of non-local vapor with composition $R_{distal}$. The local evaporative

conditions are defined by $T_0$, while the distal conditions may be either warmer or colder than $T_0$. Vapor is evaporated under those distal conditions (using a local closure assumption) and advected without fractionation before mixing with the local vapor of composition $R_e$. Figure A3 shows the isotopic composition of vapor over a range of $T_0$, with a full range of contributions from both a 5°C warmer moisture source ($T_0 + 5$°C, red colors), a 5°C colder moisture source ($T_0 - 5$°C, blue colors). This range of mixing leads to a spread of delta values of the initial vapor around the simple local closure assumption, though the

difference is generally less than that between the local and global closure assumptions.

The isotopic values of vapor produced by any of these closure assumptions ("local", "mixed", or "global"), are within the range of mean values of the observational data and show similar relationships to local environmental conditions like $SST$ and $RH$. These closure assumptions represent the bounds of a well-mixed and unmixed atmosphere, or something in between. The amount of mixing in the boundary layer could change with location and with climate mean state. Rather than tying our model

to one closure assumption, we view mixing at evaporation as an inherent uncertainty.

It is important to note that in Figure A2 we use climatological correlations between $T_0$, $SST$, and $RH$ while the observational data represent far shorter time intervals, mostly from one season. When using the observed values of $T_0$, $SST$, and $RH$ at the time of the observational measurements (Uemura et al., 2008; Liu et al., 2014), we are able to capture the complex



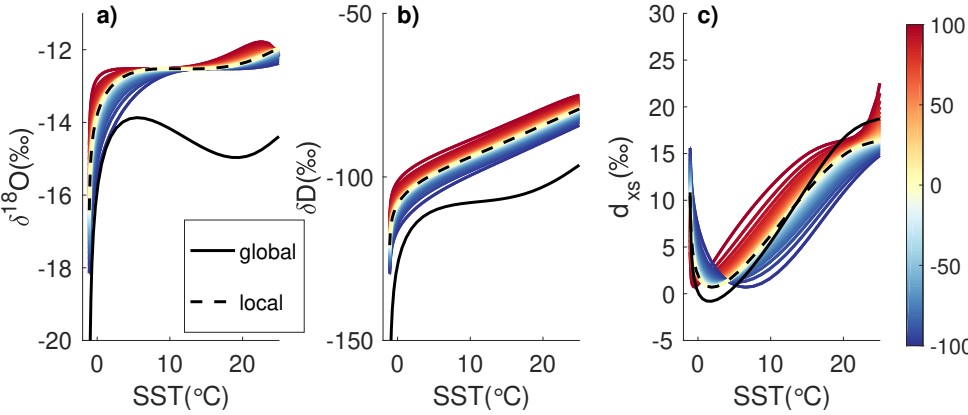

**Figure A3.** Relationship between isotopic composition of vapor and SST under different mixing scenarios. Positive values on the color scale reflects the fraction of moisture from a $5°C$ warmer moisture source mixed with the local moisture source (50= 50% moisture from local source + 50% moisture from $5°C$ warmer-than-local moisture source), while negative values reflect the fraction of moisture from a $5°C$ colder moisture source mixed with the local moisture source (-50 = 50% moisture from local source + 50% moisture from $5°C$ colder-than-local moisture source). Model results use NCEP/NCAR reanalysis for SST and RH climatology. Unmixed model results for a local closure assumption shown in black dashed line and global closure assumption in solid black line.

variability in the isotopic values of the vapor on those given days. For example, in Figure A4 we compare modeled $d_{xs}$ of vapor and modeled $^{17}O_{xs}$ of vapor to Southern Ocean vapor observations, using the observed environmental conditions at the time of the vapor measurements. The modeled relationships between $d_{xs}$ and $^{17}O_{xs}$ with $SST_0$ and $RH_0$ are in good agreement with observations.

We examine the sensitivity of initial evaporation to several model parameters discussed above in Figure A5. In all cases the modeled sensitivity to these parameterizations and uncertainties are relatively small compared to the natural variability in observations of isotopic vapor compositions. The choice of reanalysis product used to derive the climatological relationships between $T_0$, $SST_0$, and $RH_0$, and the uncertainty in those fits, has relatively small effects on the results of evaporation (Figure A2, Figure A5.a-f). We also show the influence of the initial $\delta^{18}O_{sw}$ of the ocean (Figure A5.g-i) as well as the value of

$\alpha_{diff}$ (Figure A5.j-l).

    The direct comparisons of observed and modeled vapor composition using observed $T_0$, $SST_0$, and $RH_0$ at the time of the vapor measurements (Figure A4) suggests that a single effective $\alpha_{diff}$ may not fully capture the kinetic effects across the range of surface conditions. While $\alpha_{diff} = 1.009$ leads to a good fit between observed and modeled deuterium excess for much of the range of surface conditions, there is a small persistent misfit for surface temperatures between $20°C$ and $27°C$; where a

smaller $\alpha_{diff}$ is suggested (Figure A6). While it is possible to implement a temperature dependent $\alpha_{diff}$ to reduce this misfit, we prefer a fixed $\alpha_{diff}$ to avoid over fitting a relatively small dataset in the absence of further evidence or physical reasoning. While we do not consider temperature-dependence of $\alpha_{diff}$, we do consider a range of $\alpha_{diff}$ as an inherent uncertainty in our model and account for this in the uncertainty analysis of our temperature reconstructions as discussed in Section A8.



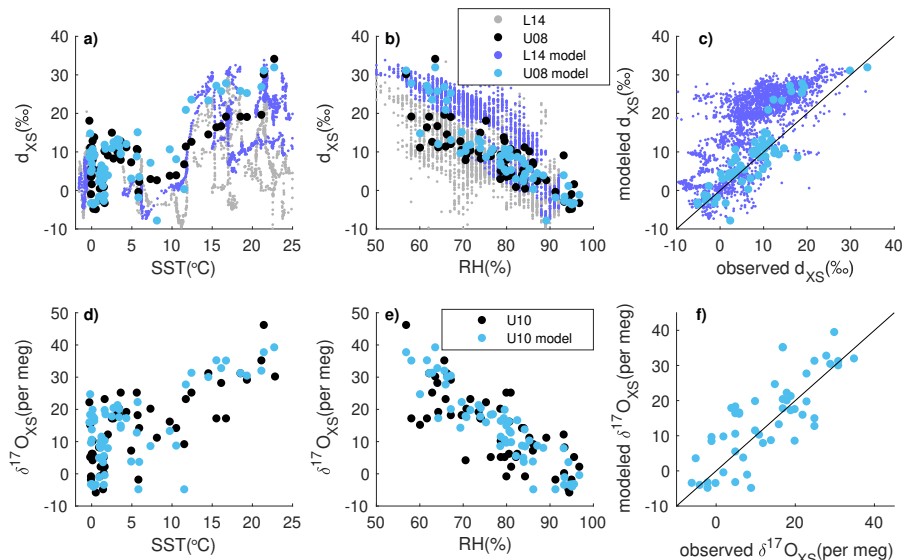

**Figure A4.** Comparison of modeled and observed isotope excess parameters and relationship to source-region conditions. a) Observed $d_{xs}$ and $SST$ relationship in Southern Ocean vapor from Uemura et al. (2008) (black dots, U08) and Liu et al. (2014) (grey dots, L14). SWIM model results for evaporation under $SST$ and $RH$ conditions observed coincident with vapor measurements of Uemura et al. (cyan dots, U08 model), and Liu et al. (purple dots, L14 model). b) Same as a) but for modeled and observed $d_{xs}$ to $RH$ relationship from observations of Uemura et al. The Liu et al. observations and model show a similar trend and are omitted for visual clarity. c) Observed $^{17}O_{xs}$ and $SST$ relationship in Southern Ocean vapor from Uemura et al. (2010) (black dots, U10), and SWIM model results run under observed sea surface conditions (cyan dots, U10 model). d) Same as c) but for observed and modeled $^{17}O_{xs}$ and $RH$ relationship in Southern Ocean vapor. Update for new figure...

We note that it is of course also possible that other factors, rather than temperature dependence of $\alpha_{diff}$, could account for the apparent misfit, such as difference between the ship measured $RH$ and $T_a$ and that felt at the water's surface, or specific mixing of non local moisture in the boundary layer on the days of the ship-based measurements. Given the magnitude of the misfit, it is also possible that spatial or temporal variability in $\delta^{18}O_{sw}$ and $\delta D_{sw}$, could account for the misfit (Schmidt et al., 5 1999).

### A2.2 Distillation

We next discuss the distillation of water isotopes during transport. As an air parcel cools, water condenses, fractionates, and is removed as precipitation. The essential differential equation for Rayleigh distillation (Rayleigh, 1902; Dansgaard, 1964; Criss, 1999) is:





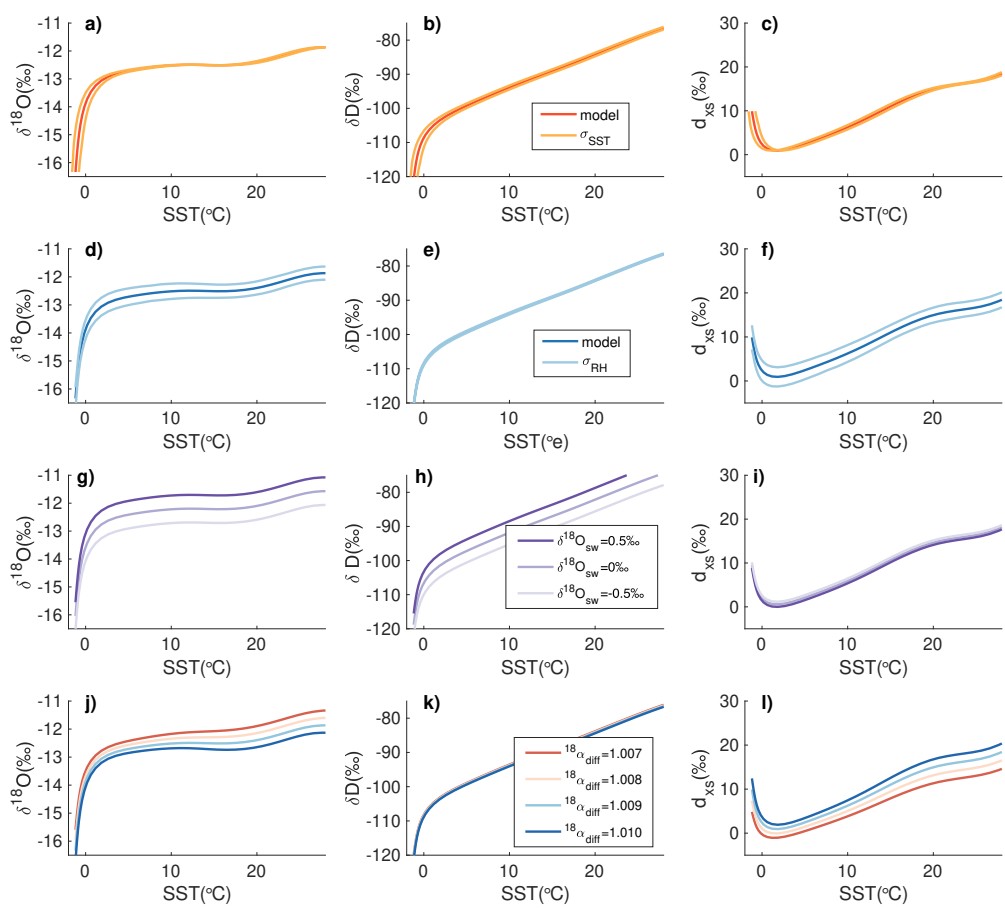

**Figure A5. a-f** Sensitivity of modeled $\delta^{18}O$, $\delta D$, and $d_{xs}$ of vapor to uncertainty in the reanalysis-based fits between climatological $T_a$, $SST$, and $RH$ in the NCEP/NCAR reanalysis. a) and b) show modeled isotope vapor relationship to uncertainty in the climatological $T_a$-to-$SST$ relationship. Red lines show model run using the central estimate of the fit, orange lines show the spread expected with $\pm\sigma_{err}$ of the fit as shown in Figure A1. c) and d) are the same as a) and b) but showing the central estimate (dark blue) and spread associated with $\pm\sigma_{err}$ (light blue) in the climatological $T_a$-to-$RH$ relationship shown in Figure A1. g-i) Sensitivity of modeled isotope values of vapor to the $\delta^{18}O_{sw}$ of sea water. Values of $\delta^{18}O_{sw}$ from -0.5‰ to 0.5‰ are specified, representing most of the global variance in $\delta^{18}O_{sw}$. Values of $\delta D_{sw}$ are determined based on correlations of $\delta^{18}O_{sw}$ and $\delta D_{sw}$ from observations (Schmidt et al., 1999). j-l) Sensitivity of modeled isotope values of vapor to a range of $^{18}\alpha_{diff}$ values from 1.007 to 1.010. Local closure assumed in all panels.

$$\frac{d\ln(R)}{d\ln(f)} = \alpha - 1 \tag{A9}$$





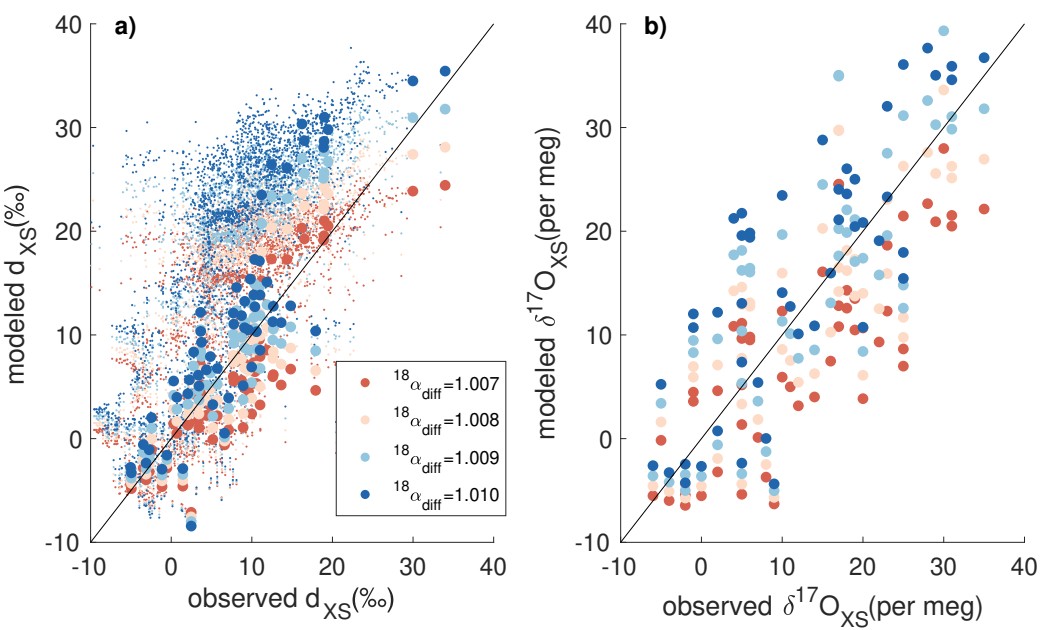

**Figure A6.** Relationship between observed and model $d_{xs}$ (a) and $^{17}O_{xs}$ (b) of vapor over the Southern Ocean. Observed vapor values are from Uemura et al. (2008, 2010) (large dots) and Liu et al. (2014) (small dots). Modeled values use the reported $T_0$, $SST_0$, and $RH_0$ from the observations and four different values of $^{18}\alpha_{diff}$ (shown in color of dots). 1:1 line shown in black.

where $f$ is the fraction of initial water vapor remaining in the air parcel. The amount of moisture at any temperature along the pathway is found by integrating the changes in the saturated mixing ratio $r_s$ owing to pseudoadiabatic cooling from the source (Dansgaard, 1964) . Thus,

$$f = \frac{q}{q_0} = \frac{r_s}{r_{s0}} \qquad (A10)$$

5    In general, condensation occurs in the model at saturation, and thus the temperature dependent equilibrium fractionation factor $\alpha_{eq}$ is used in Equation A9. However at cold conditions there may be supersaturation of vapor over ice leading to additional kinetic fractionation. Following previous models (Jouzel and Merlivat, 1984), the total fractionation, $\alpha_{tot}$ factor is $\alpha_{tot} = \alpha_{eq}\alpha_k$. Equation A9 thus becomes:

$$d\ln(R) = (\alpha_{tot} - 1)d\ln(f) \qquad (A11)$$

10    The kinetic modification factor, $\alpha_k$, depends on the supersaturation of vapor over ice, $S_i$:


$$\alpha_k = \frac{S_i}{\alpha_{eq} \times \frac{D}{D^*}(S_i - 1) + 1} \tag{A12}$$

Following Jouzel and Merlivat (1984), we use the ratio of diffusivities for oxygen isotopes $D^{16}/D^{18} = 1.0285$ during moisture transport, representative of pure molecular diffusion and ignoring the negligible ventilation effect. Likewise we use $D^1/D^2 = 1.0251$ for the ratio of diffusivities of hydrogen isotopes. These values imply a constant $\phi_{diff}$ during transport equal to 0.88

(Jouzel and Merlivat, 1984), rather than the temperature dependent $\phi_{diff}$ used in the evaporation scheme. We prefer this value for simplicity, consistency with earlier work, and for a lack of experimental measurements of $\phi_{diff}$ at the colder temperatures experienced during transport.

In the mixed-phase portion of the transport pathway, the effective fractionation factors are determined by the mixing fractions of ice and liquid condensate. Following Kavanaugh and Cuffey (2003),

$$\alpha_{eff} = \alpha_{tot(l-v)}F_{(liq)} + \alpha_{tot(i-v)}F_{(ice)} \tag{A13}$$

The temperature dependence of the ice and liquid fraction is shown in Figure A7.a and based on satellite observations (Hu et al., 2010).

### A2.3 Supersaturation

The supersaturation of vapor over ice is a critical parameterization in water-isotope distillation models. The true relationship
of supersaturation to environmental conditions is the result of complex cloud-microphysics (Hong et al., 2004). Because of its strong influence on water-isotope fractionation and the uncertainty in the underlying physics, the supersaturation is often parameterized to depend on temperature and tuned to fit water-isotope models to observations (Jouzel and Merlivat, 1984; Kavanaugh and Cuffey, 2003; Schoenemann et al., 2014). Jouzel and Merlivat (1984) parametrized the supersaturation as a function of temperature and note that available water-isotope data could not distinguish among possible functional forms of the
parameterization (e.g. linear, exponential, etc.). Their linear parametrization has been used extensively in water-isotope models since:

$$S_i = a - b \times T \tag{A14}$$

where $a$ and $b$ are tuned to fit observational data.

It is important to note here that the prescribed mixing of liquid and ice in the cloud imply a supersaturation of vapor over ice that follows the blue curve shown in Figure A7.b, which is inconsistent with the supersaturation driving kinetic fractionation as prescribed in Equation A14. The presence of both liquid and ice phases in a cloud is not the only potential source of supersaturation. The lack of condensation nuclei, for example, allows supersaturation to remain high in cold, ice-only conditions (Hong et al., 2004), rather than returning to unity as the cloud becomes entirely ice-phase. It is common for





water-isotope models, even those in some GCMs (e.g. Schoenemann et al. (2014)), to have multiple variables equivalent to supersaturation in different aspects of the same model, such as the isotope-fractionation and precipitation schemes, which may not be self-consistent. Because the environmental supersaturation experienced by the air parcel is related to the relationship between temperature and moisture removal (that is $d\ln(f)/dT$), and the supersaturation driving kinetic fractionation relates temperature to $\delta$ values (largely through Eq. A12), an inconsistency in the model's view of supersaturation can influence the modeled water isotope-temperature relationship in unphysical ways.

To resolve this physical inconsistency, precipitation only occurs in SWIM when the the parcel reaches the prescribed supersaturation by dictating an effective saturated mixing ratio of the air parcel, in which $r_{s(eff)} = r_{s(ice)}S_i$. Ensuring consistent supersaturation across the model leads to a smoother relationship between temperature and the $\delta$ values of precipitation. This is in contrast to rather complex curvature in the temperature-water isotope relationship that results if inconsistent relationships between saturation and temperature are used in the precipitation and fractionation schemes, which is generally incompatible with observations. In line with previous work (Jouzel and Merlivat, 1984), we find that using only the supersaturation implied by the mixing of ice and liquid, across all aspects of the model, results in a relationship between $\delta^{18}O$ and $\delta D$ irreconcilable with observations. Were the air parcel to return to non-supersaturated conditions in the ice-only portion of the cooling pathway, the simultaneous transition to equilibrium-only fractionation would drive a slope of $\partial \delta^{18}O/\partial \delta D$ that is not compatible with measured values in Antarctic precipitation. This gives additional credence to sustained supersaturation at cold temperatures.

## A3    Application to Antarctic ice core sites

### A3.1    Moisture source distributions

The atmospheric circulation transports moisture poleward of $\approx 30°$S (Figure A8a). The mean evaporation latitude of moisture that precipitates at any given site can be estimated from moisture-tagged GCM experiments (Markle et al., 2017). The difference between the mean latitudes of moisture evaporation and precipitation steadily increases between the subtropics and the pole (Figure A8a and b). The mean evaporative latitude of moisture that precipitates in Antarctica ranges between 44 and 50°S (Figure A8c). The surface elevation of the ice sheet is a strong predictor of the mean latitude of precipitation with higher elevation sites having more equator-ward moisture sources (Figure  A8d) due to topographic isolation (Sodemann and Stohl, 2009; Bailey et al., 2019). There are some notable asymmetries in this general pattern. The large embayments are areas of comparatively high-latitude mean moisture source, such as the Victoria Land coast in the Ross Sea region.

The mean moisture source latitude is, however, not the full story. The moisture reaching any Antarctic site does not originate from just a single mean source-latitude nor follow a single temperature pathway. The contribution of moisture evaporated from different latitudes to the final precipitation at a site defines a moisture source distribution (Markle et al., 2017), which reflects the combination of the spatial pattern of evaporation, cumulative rainout, and the influence of atmospheric circulation. Here we diagnose annual mean moisture-source distributions (MSDs) as a function of latitude from a moisture-tagged run of the Community Atmosphere Model (CAM) for East and West Antarctic sites (details are given in Markle et al. (2017)), shown in Figure A8e. Moisture source distributions derived from other methods like trajectory modeling are similar (e.g. Sodemann and





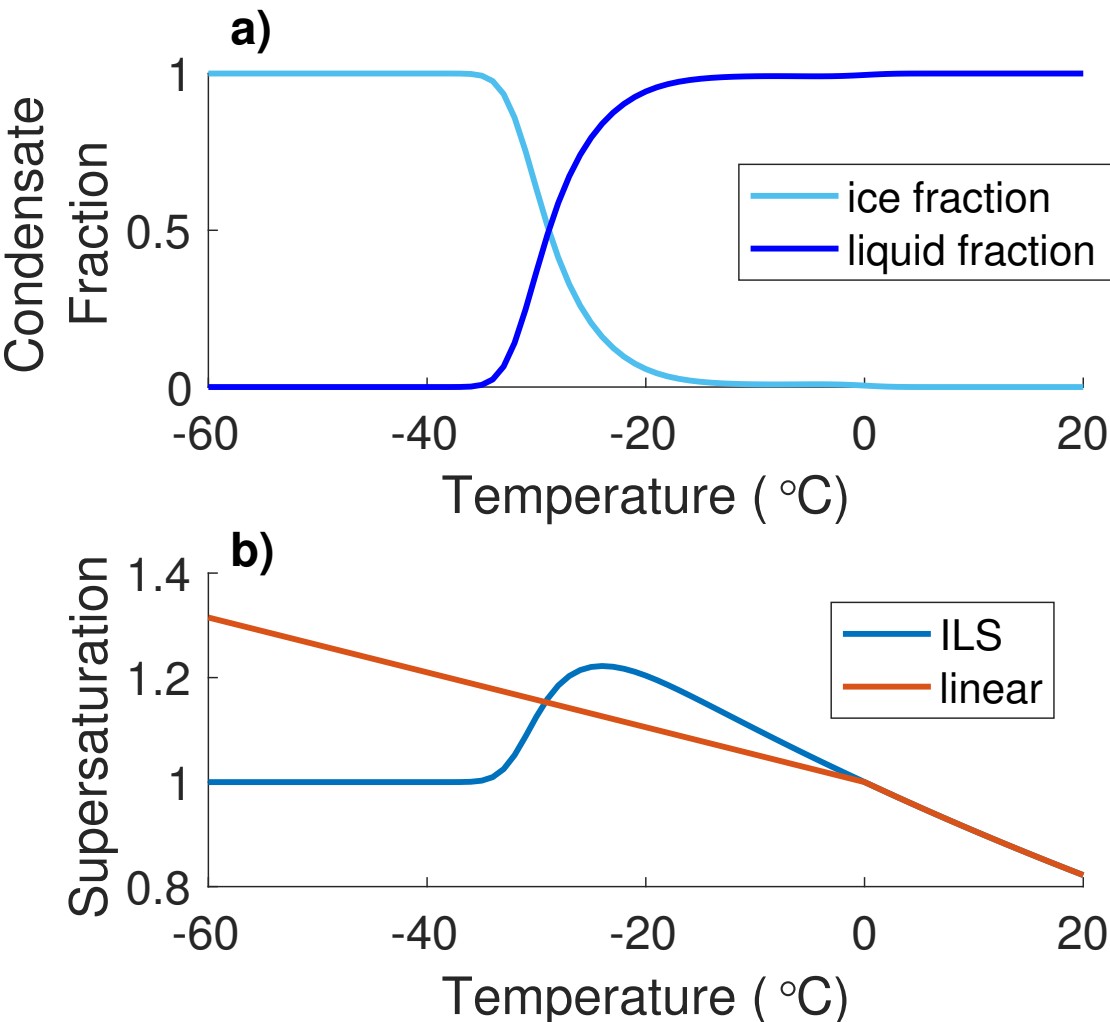

**Figure A7.** Supersaturation of vapor over ice. a) Fractions of ice (cyan) and liquid (blue) condensate as a function of temperature. Curves are derived from satellite based measurements (Hu et al., 2010). b) Supersaturation as a function of temperature. The blue curve shows supersaturation based solely on the saturated vapor pressures of ice and liquid, and the mixing fractions based on the curves shown in panel a), and the pseudoadiabatic assumption ("ILS"). The red curve shows the linear parameterization of supersaturation ("linear") used in the model, $S_i = 1 - 0.00525°C^{-1} \times T$.

Stohl (2009); Markle et al. (2012); Buizert et al. (2018)). These MSDs dictate the influence of evaporation source conditions ($T_a$, $RH$, $SST$) on moisture reaching any Antarctic site. There are some zonal asymmetries in surface conditions over the Southern Hemisphere oceans, but the strong latitudinal gradients are the largest source of spatial variance in these conditions at climatological time scales.



While the mean latitudes of moisture sources vary between Antarctic sites, largely as a function of site elevation, Antarctic MSDs are not fundamentally distinct in latitude, but rather span broadly overlapping swaths of the Southern Hemisphere, from the Antarctic coast to the tropics (Figure A8e). The difference in weighted mean moisture source latitude between Antarctic ice core sites is less than $10°$ of latitude, while the moisture source distribution for any one site spans over $40°$ of latitude. Local evaporation is a small contribution to the moisture precipitating at Antarctic sites. On average moisture is transported more than $20°$ of latitude to reach Antarctica.

Given the broad range of evaporative conditions that contribute to moisture precipitating at an ice core site, what is the meaning of the $T_{source}$ that can be reconstructed from water isotope records? It is the moisture-weighted evaporative temperature, determined by the convolution of the spatial pattern of the MSD and the underlying surface temperatures (Figure A24, (Markle et al., 2017)). Both surface temperatures at fixed locations and the pattern of the MSD can change independently. The water isotope records alone do not allow the disentanglement of these two patterns which may have different temporal evolution (Markle et al., 2017).

To understand the moisture transport and water isotope distillation to Antarctic sites it is important to consider evaporation from the range of conditions comprising the moisture source distribution. We thus use an ensemble of temperature pathways for Antarctic precipitation defined by a range of Antarctic condensation temperatures as well as the broad range of evaporation temperatures underlying the Antarctic moisture source distributions. The means of these distributions vary across the continent.

### A3.2 Condensation site conditions

During transport, moisture is cooled from initial surface air temperature at evaporation to subsequent condensation temperatures. The condensation temperature is not the same as the surface temperature where that precipitation falls. Indeed, there is a vertical and temporal distribution of condensation contributing to precipitation that falls at any point on the surface, analogous to the horizontal and temporal distribution of evaporation contributing the moisture ultimately transported to any precipitation site. What is the meaning of $T_c$ reconstructed from ice core records? It is the vertical profile of temperature weighted by the vertical profile of condensation that yields net accumulation to a site. The weighted condensation temperature has a distinct relationship to the surface temperature across the globe.

Antarctica has strong climatological inversions such that temperature aloft is often warmer than the surface (Connolley, 1996). Masson-Delmotte et al. (2008) review the relationship between the condensation temperature and the surface temperature ($T_s$) over Antarctica, and compare the surface temperature to the weighted annual-mean condensation temperature in both ERA-40 reanalysis (1980-2002) and MAR, a high-resolution mesoscale model forced by ERA-40 (c.f. Fig. 8, Masson-Delmotte et al. (2008)). In both models the upper bound of the Antarctic condensation temperature appears to be set by the peak inversion temperature, though condensation temperatures are on average colder than the peak inversion temperature (meaning simply that condensation occurs at aa range of temperature up to the peak inversion temperature). Masson-Delmotte et al. (2008) calculate a best fit of the surface to condensation temperature slope as $0.65°C/°C$ in the ERA-40 data, consistent with previous work that found a slope of $0.67°C/°C$ (Connolley, 1996; Jouzel and Merlivat, 1984). The spread of condensation temperatures in the higher-resolution MAR model suggests colder condensation temperatures than in the lower resolution



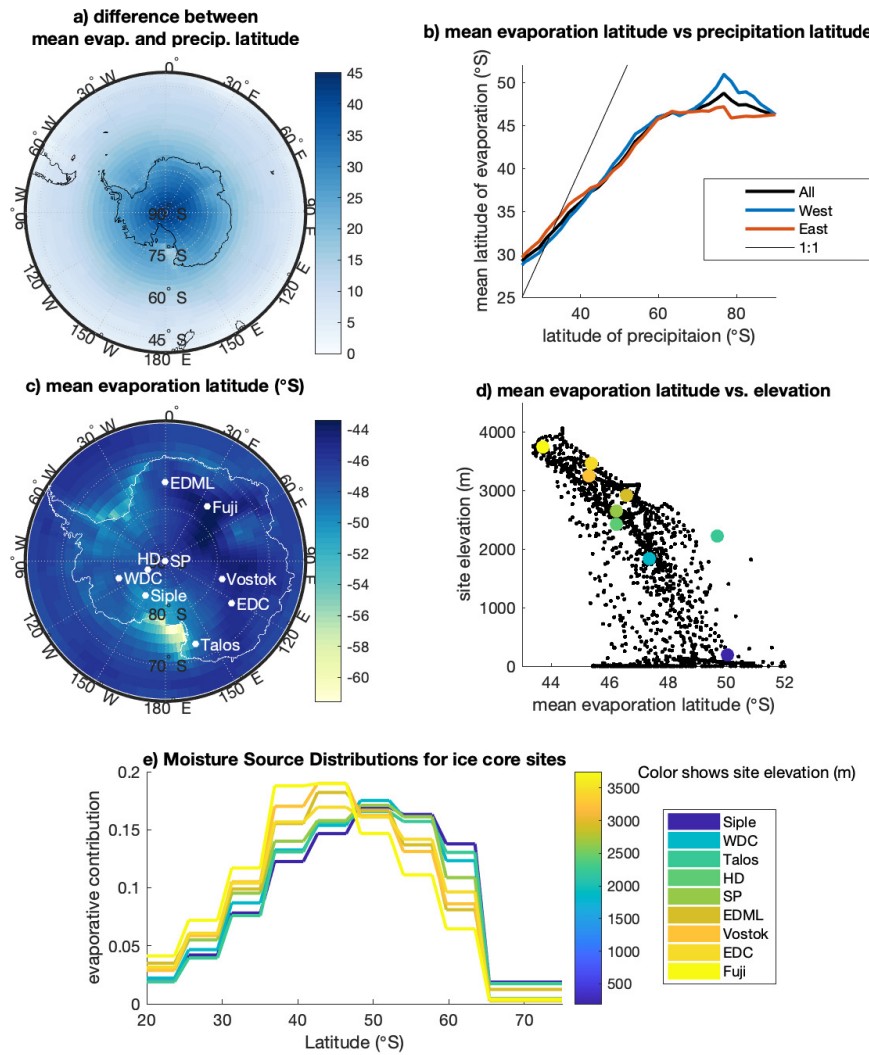

**Figure A8.** Moisture sources and transport to Antarctica from moisture tagged CAM experiment. a) Difference, in degrees latitude, between the latitude of precipitation and mean latitude of evaporation (effectively mean transport in degrees of latitude to any site). b) Mean latitude of evaporation vs latitude of precipitation. All longitudes are shown in the black line; longitudes encompassing West Antarctica is shown in the blue line while longitudes encompassing East Antarctica is shown in red. c) The mean evaporation latitude (in °S) of precipitation falling at all Antarctic grid points. Select ice core sites shown in white. d) The relationship between mean evaporation latitude and site elevation across Antarctica. Select ice core locations shown in color. e) Latitudinal moisture source distributions for select Antarctic Ice core sites, colored by site elevation.

reanalysis (Masson-Delmotte et al., 2008). The strength of the Antarctic inversion diminishes with increasing surface temperature (Connolley, 1996), and relatively warm Antarctic surface temperatures (e.g. $> -20°C$) are associated with condensation temperatures colder than the surface temperature (Masson-Delmotte et al., 2008).





We analyze the relationship between surface temperature and condensation temperature in monthly MERRA-2 reanalysis from 2008 through 2017 (Gelaro et al., 2017). We show the climatological, zonal mean, vertical profile of air temperature, the sum of the convective and large-scale precipitation source production rate, and the rate of reevaporation and sublimation of precipitation in Figure A9. The relationship between the climatological weighted condensation temperature and the surface

air temperature at every grid point is shown in Figure A10. Note that this calculation accounts for the seasonality of precipitation throughout the atmospheric column, as well as the reevaporation and sublimation of falling precipitation. Ignoring reevaporation and sublimation leads to qualitatively similar results.

The primary take-away is that the MERRA2 data show a generally linear relationship between condensation and surface temperature for typical Antarctic surface temperatures. That relationship, however, is not linear at warmer surface tempera-

tures. Indeed, even at surface temperatures below zero, the relationship is not strictly linear, but rather steepens with decreasing temperature. The relationship between the surface air temperature and the weighted condensation temperature (for surface temperatures below -10°C) has an average slope between 0.61 and $0.64^{\circ}C/^{\circ}C$ depending on whether one accounts for reevaporation and whether the comparison is between the surface or 2-meter air temperature. Note that this slope is weighted toward the surface temperature of regions comprising more model grid points. Further, the slope clearly steepens with decreasing

temperature, reaching $\approx 0.71 - 0.75^{\circ}C/^{\circ}C$ at the very coldest Antarctic surface temperatures. Given the uneven distribution of grid points in temperature space, it is difficult to estimate the robustness of this steepening of slope.

Using our non-linear temperature reconstruction method, we model the condensation temperature for every pair of $\delta^{18}$O and $\delta$D samples in the MD08 and GNIP data sets (Masson-Delmotte et al., 2008; IAEA, 2001) that also have a reported mean surface temperature. We compare the relationship between the modeled condensation temperature and the reported surface

temperature in Figure A10. The pattern of this reconstructed relationship is remarkably consistent with that found in the MERRA2 data set, even to some extent at warm surface temperatures. This is despite the potential for offsets between the reported surface temperature and that at the time of the precipitation for the MD08 and GNIP precipitation samples, differing time intervals, potential moisture biases in the column in MERRA2 (Bosilovich et al., 2017), as well as the lack of processes in our isotope distillation model that should be important, for example, in tropical convection, or for example that might alter

Antarctic precipitation after deposition.

Examining all modeled condensation temperatures for samples in the MD08 and GNIP data sets with reported surface temperatures below 15°C, we find slopes between 0.62 and 0.67 $^{\circ}C/^{\circ}C$. For just the Antarctic precipitation samples in the MD08 data set we find a best fit slope between the reported surface temperature and our modeled condensation temperature of 0.67-0.69$^{\circ}C/^{\circ}C$ (Figure A10), depending on the model assumptions (0.69$^{\circ}C/^{\circ}C$ under our base assumptions) and whether

below-freezing source evaporation is included (see below), in good agreement with previous Antarctic observational studies (Connolley, 1996; Jouzel and Merlivat, 1984). These slopes sit well within the range found in the MERRA2 data. We also reconstruct the condensation temperatures for the top-most samples from several deep ice cores and compare those to the reported annual average temperatures for those sites (Figure A10). We find a best fit slope between 0.68-0.70$^{\circ}C/^{\circ}C$, depending on whether we average samples from the last 50 or 100 years, though only five points describe these lines.



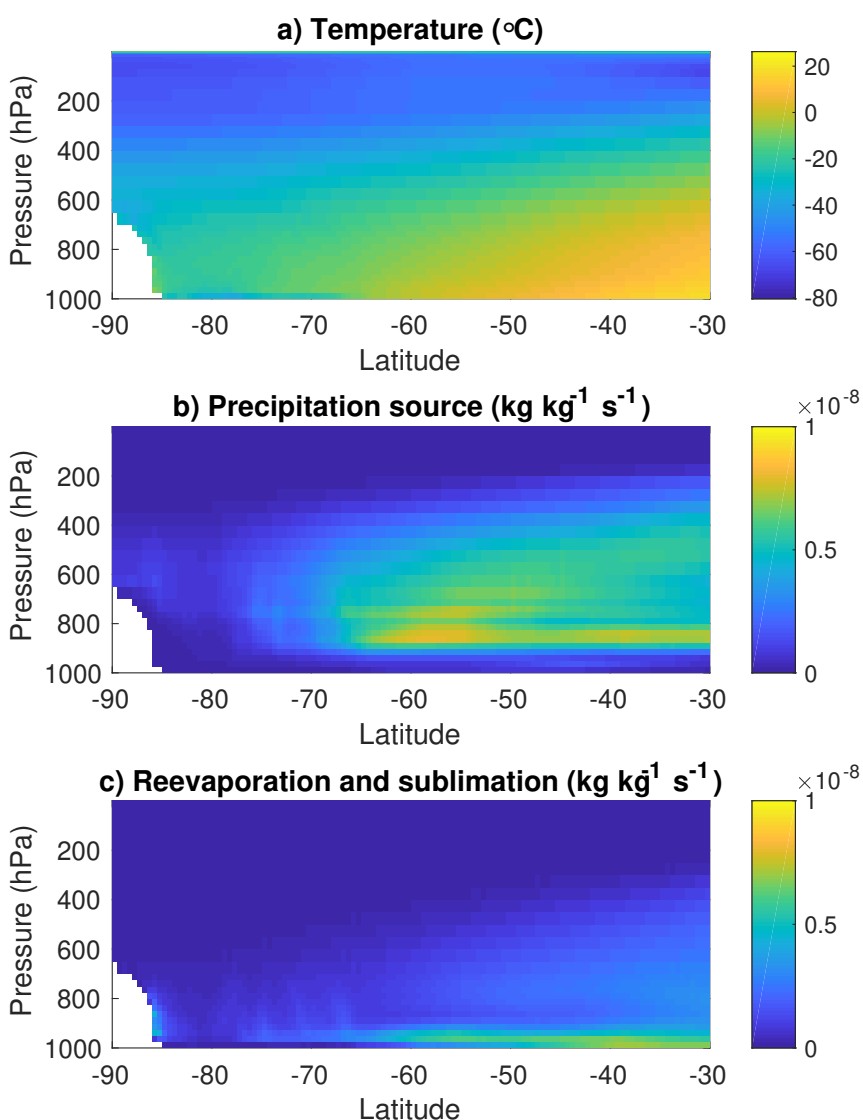

**Figure A9.** MERRA2 Reanalysis Data 2008-2017. A) Annual mean zonal mean air temperature as a function of pressure and latitude. B) Annual mean zonal mean precipitation source, kg of water per kg of air per second. C) Annual mean zonal mean reevaportion and sublimation of falling precipitation, in same units and color scale as panel B.

Based on the above results we use the equation $T_c = 0.69°C/°C\ T_s - 8.2°C$ as our base estimate to reconstruct Antarctic surface temperatures, however we consider an uncertainty of $\pm 0.02°C/°C$ in the slope. Our base estimate leads to good agreement with the observed relationship between global $\delta^{18}O$ and surface temperature (Figure A11).



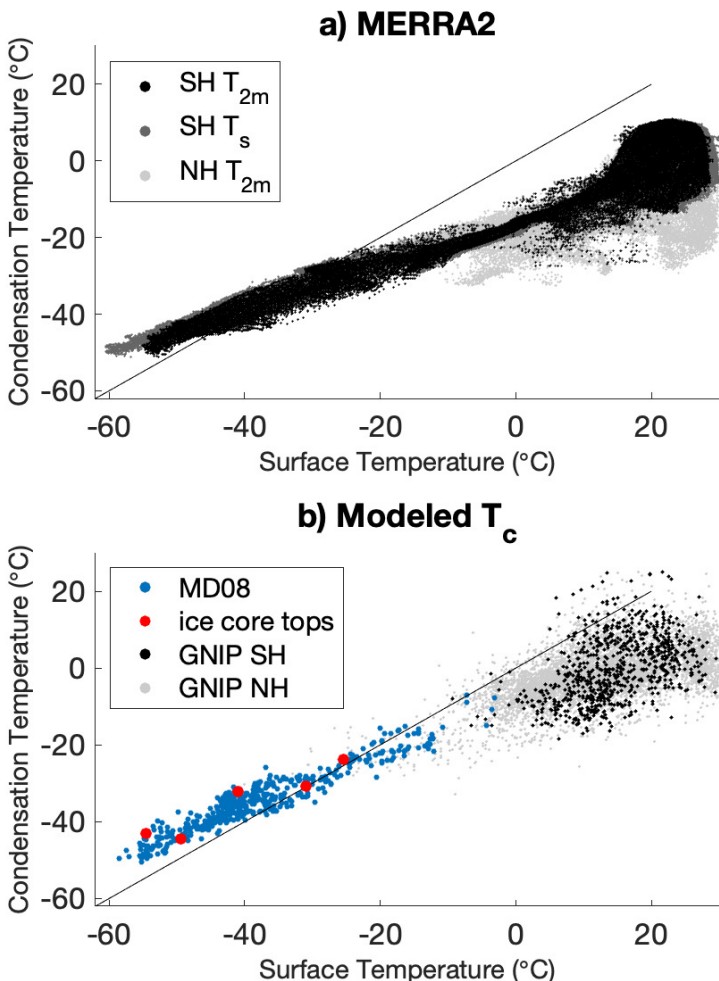

**Figure A10.** Weighted condensation temperature as a function of surface temperature. **a)** MERRA2 reanalysis data, 2008-2017. Seasonally and vertically weighted condensation temperature (accounting for reevaporation and condensation) for every grid point in the Southern Hemisphere against 2m air temperature (black) and surface temperature (dark grey). The same relationship with 2m air temperature is shown for the Northern Hemisphere in light grey dots. **b)** The reported surface temperature and the SWIM-modeled condensation temperature using pairs of $\delta^{18}O$ and $\delta D$ in the Masson-Delmotte et al. (2008) data base ('MD08', blue dots), from both Southern Hemisphere sites (black dots) and Northern Hemisphere sites (light grey dots) in the GNIP data base, and the average of the top 50 years of sites from several deep ice core sites (red dots). Solid black lines are 1:1.





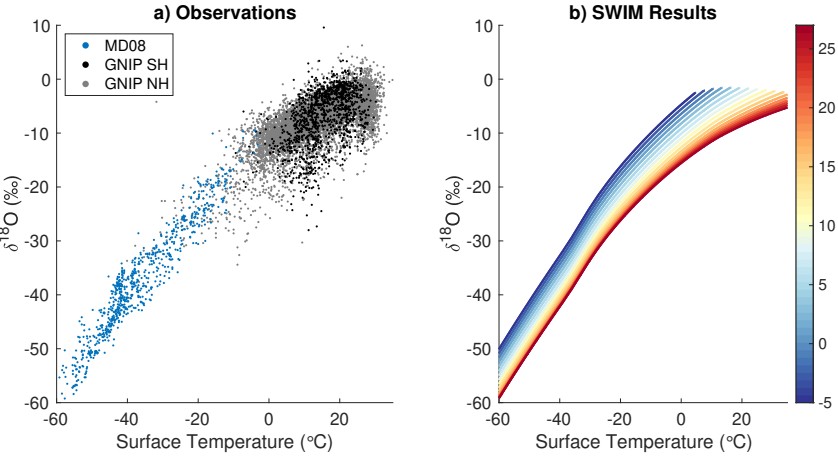

**Figure A11. a)** Observed relationship between $\delta^{18}O$ of precipitation and reported surface temperature in the Masson-Delmotte et al. (2008) data base ('MD08', blue dots), as well as the GNIP data base both from both Southern Hemisphere (black dots) and Northern Hemisphere (light grey dots). **b)** Modeled relationship between $\delta^{18}O$ of precipitation and surface temperature, using our linear scaling, colored by initial evaporation air temperature (in $^\circ$C).

### A3.3 Seasonality

Does seasonality in the hydrological cycle systematically bias climatological information in ice-core water-isotope records? While the difference between precipitation-weighted surface temperature and annual mean surface temperature is often discussed, this is not strictly the relevant comparison from the perspective of water isotope ratios of precipitation. As discussed

above, the critical comparison is between the annual mean surface temperature and the condensation weighted temperature (integrated over both time and altitude). Our analysis of the MERRA2 reanalysis data (Figure A10) takes seasonal variation in precipitation and the vertical temperature profile into account. Differences in seasonality of condensation at different sites contributes to the spread around the central relationship. It is nevertheless useful to investigate potential bias in the precipitation weighted surface temperature, since direct observations of Antarctic surface temperature are more common than full profiles

of the atmospheric column.

The potential for precipitation-weighting to bias annual average surface temperature depends on the phase angle between the seasonal cycles of precipitation and temperature. Only strong correlation or anti-correlation between the two cycles leads to persistent biasing. The potential for bias also depends on the ratio between stochastic and seasonal variability in both temperature and accumulation. If non-seasonal variance in accumulation is very large compared to the amplitude of the seasonal

cycle in accumulation, for example, then the potential for bias is small. We examine seasonality in monthly surface temperature and snowfall over Antarctica in the ERA Interim reanalysis as well as global precipitation in the MERRA2 reanalysis. While the the annual average Antarctic surface temperature and the precipitation-weighted surface temperature are often different in either reanalysis product, we find little systematic bias. Across the Antarctic the month-to-month and year-to-year variance



in snowfall is large compared to the climatological seasonal cycle. The stochastic sampling of the seasonal cycle in surface temperature overwhelms the potential bias introduced by the average seasonality of precipitation. Further, the timing of the climatological annual cycle in snowfall varies across the continent, whereas the annual temperature-cycle is quite coherent. The potential for seasonal bias thus varies dramatically between sites, even in the absence of the dominant stochastic sampling.

We compare the annual average surface temperature to the precipitation-weighted annual temperature at every grid point in the Southern Hemisphere. The mean bias for sites with typical Antarctic surface temperatures is less than 0.33°C, with the precipitation-weighted temperature being slightly colder on average. We find no systematic dependence of this bias on the surface temperature itself. While individual sites do show differences up to 4°C over the interval examined, our analysis does not suggest such differences are persistent at a site. None of these analyses of monthly data take into account potential biases

at the scale of individual precipitation events. The intermittency of Antarctic snow fall likely complicates the relationship between condensation-weighted and annual-mean temperature at the seasonal and annual scale. At the same time, however, precipitation intermittency reduces potential seasonal biasing at climatological timescales, by degrading any coherence in the seasonal cycles of accumulation and temperature.

Could seasonality in evaporation bias reconstructed source region $T_0$? We examine the seasonality of Southern Hemi-
sphere evaporation in the monthly MERRA2 reanalysis, comparing the annual average over-sea surface temperatures to the evaporation-weighted annual temperatures. Between 35°and 65°S, the bulk of Antarctic moisture sources, evaporation-weighted surface temperatures are slightly colder than mean annual surface temperatures (the mean difference is 0.123°C, with 95% of points between $-0.25$°C and 0.5°C). South of the climatological sea ice zone, mean evaporation temperatures are a couple degrees warmer than mean annual surface temperatures on average, though our moisture tagging analysis suggests
that these areas contribute at most a couple percent of the total moisture arriving at typical Antarctic sites.

## A4    Tuning the Simple Water-Isotope Model

We tune the Simple Water Isotope Model by adjusting the temperature dependence of the supersaturation of vapor over ice. Given insufficient observational and physical constraints, we parameterize the supersaturation as a linear function of temperature (Jouzel and Merlivat, 1984) as above, $S_i = a + b \times T$, set $a = 1$ and tune the slope, $b$. The supersaturation has a strong
influence on the kinetic fractionation (Equation A12) and thus the relationship between $\delta$D and $\delta^{18}$O in vapor and precipitation. We tune the model to yield the observed relationship between $\delta$D and $\delta^{18}$O in global precipitation, rather than the relationship between $\delta$ values and environmental variables such as surface temperature.

Our target observational data set includes water isotope measurements of precipitation from Antarctica and around the globe. The bulk of this compilation is that published by Masson-Delmotte et al. (2008). We include additional published surface snow
and precipitation measurements from the GNIP database (IAEA, 2001), from surface traverses at Dome A (Xiao et al., 2013; Pang et al., 2015), Dahe et al. (1994), and a $^{17}O_{xs}$ compilation from Schoenemann et al. (2014). We also include previously unpublished measurements from a transect of snow pits and shallow firn cores across the main divide of the West Antarctic Ice Sheet. Samples from five sites were collected spanning 80 km across the ice flow divide in the 2012/13 summer season. Samples were measured at IsoLab, University of Washington, Seattle WA, USA. Measurement techniques are described in





Markle et al. (2017). Measurements were made using laser spectroscopy (Picarro L2120-i analyzer). Data are reported relative to the VSMOW (Vienna Standard Mean Ocean Water) standard, and normalized to SLAP.

The global relationship between $\delta D$ and $\delta^{18}O$ has an approximate slope of 8, as codified in the historical definition of the deuterium excess parameter (Dansgaard, 1964). However the slope is not fundamental (Craig, 1961); as discussed in Section 1.2 the true observed relationship is nonlinear (Uemura et al., 2012), as is the theoretical relationship even in the absence of kinetic fractionation (Markle et al., 2017). Uemura et al. (2012) find an empirical fit between $\delta' D$ and $\delta'^{18}O$ in a global data set of precipitation. They use a 2nd order polynomial fit, which is the basis for the logarithmic deuterium excess parameter (Uemura et al., 2012; Markle et al., 2017) (Equation 4). From Equation A9, we can see that theoretical relationship between $\delta D$ and $\delta^{18}O$, given any amount of distillation, depends on the ratio of $\exp(^{D}\alpha_{tot})/\exp(^{18}\alpha_{tot})$, where each $\alpha_{tot}$ is itself a nonlinear function of temperature as outlined throughout Section A2.2. The ratio of exponential functions can be estimated to any arbitrary degree of accuracy with a polynomial function.

Our modern data set includes several new sets of measurements in addition to those used in Uemura et al. (2012). We find similar coefficients in a 2nd-order polynomial fit between $\delta'^{18}O$ and $\delta' D$ in our larger data set compared to those found by Uemura et al. (2012): $A = -29.2$ and $B = 8.45$ (see Equation 4). Because these coefficients are not significantly different than those previously published, and for consistency with that work, we use the coefficients found by Uemura et al. (2012) ($A = -28.5$ and $B = 8.47$) to define a logarithmic deuterium excess parameter, $d_{ln}$. We find no benefit nor justification for using higher order fits to this data set.

We run the SWIM model to produce an ensemble of temperature trajectories representing a wide range of possible evaporation and condensation temperatures ($T_0$ varies from 0 to 28°C; $T_c$ from 27 to -60°C). We then compare the resulting cloud of modeled $\delta'^{18}O$, $\delta' D$, and $d_{ln}$, finding a 2nd-order polynomial fit between the modeled $\delta' D$ and $\delta'^{18}O$ from the ensemble of temperature trajectories. We tune the model by iteratively adjusting the $b$ value in the supersaturation parameterization to minimize the difference between the modeled and observed relationship between $\delta' D$ and $\delta'^{18}O$ (Equation 4). This is easily visualized in a plot of $\delta'^{18}O$ and $d_{ln}$ (e.g. Figure A12.a), as the average $\delta'^{18}O$-to-$d_{ln}$ relationship is flat, in measured samples, by definition.

Using the local closure assumption we find an optimal tuning of $S_i = 1 - b \times T$, for $b = 0.00525$ °C$^{-1}$ as shown in Figure A12. The observational data allow large ranges of the value of $b$ to be rejected as shown in Figure A13b and c; the fit coefficients of the resultant $\delta' D$ and $\delta'^{18}O$ are clearly irreconcilable with observations. While it is possible to optimize $b$ as described above, there are limitations to this tuning procedure and the observational data may not allow discrimination within a small range of $b$ values. In principle we should not expect a 2nd-order polynomial fit between modeled $\delta' D$ and $\delta'^{18}O$ to be identical to the observed fit: the observational target data represent a variety of timescales from sub-seasonal to multi-year averages; the sites are neither evenly nor randomly distributed over the Antarctic continent; and the sites represented in the observational data set have specific moisture source distributions, mean latitudes of evaporation, and evaporation temperatures, which are not known *a priori*.

Because higher elevation, colder Antarctic sites likely have more equator-ward MSDs (Figure A8), we should expect more depleted $\delta'^{18}O$ in the target data to be associated with slightly warmer $T_0$ (that is *modeled* results from a single value of $T_0$





should not be strictly flat in the $\delta'^{18}O$-$d_{ln}$ space). If we take the MSDs determined from the GCM experiments described earlier as representative, and assuming the climatological meridional profile in surface air temperature, we should expect a 1-2°C increase in $T_0$ between $\delta'^{18}O$ values of -40‰ and -55‰ in the observational data set. This is a fairly small shift in mean $T_0$ compared to the range of $T_0$ that may contribute to a site, but should give some downward curvature to model results
of equal $T_0$ at the coldest $T_c$ values.

The appropriate weighting of model realizations with different $T_0$ could vary within our tuning cost function, in principle, depending on the site of the target data. However without knowing the true moisture source distribution and conditions for each sample in the target data *a priori*, assigning a single objective weighting scheme is difficult. We prefer values of $b$ in which the more depleted observational data transect model realizations of slightly warmer $T_0$, though this is not a strong constraint on
the tuning. While one can reasonably reject most possible values of $b$ (as in Figure A13), we cannot justifiably discern between others within a small range (e.g. $0.005 \leq b \leq 0.0055$). This gives a small, but inherent uncertainty to the model tuning, and in turn our temperature reconstructions. In spite of these limitations, the tuning procedure reproduces the observed isotope relationships well.

The model tuning is not especially sensitive to which reanalysis data set is used for correlations of the initial evaporation
conditions nor the season of evaporation. The model is however sensitive to which closure assumption is used.

While the linear, temperature-dependent parameterization of supersaturation is both simple and widely used, the physical processes determining supersaturation are complex. To understand the sensitivity of the model to this parameterization we also test a nonlinear parametrization of supersaturation, $S_i = a - b \times T - c \times T^2$. If the physical source of the high supersaturation at very cold conditions is related to the absence of condensation nuclei, the supersaturation may not linearly increase at very
cold temperatures, as there could be diminishing returns as the atmosphere becomes increasingly clean. A small but positive $c$ parameter that gradually decreases the slope in $S_i$ with decreasing temperature could be plausible. Only very small values of the second-order term, $c$, those of order $5 \times 10^{-6}$ °C$^{-2}$, are reconcilable with the observed $\delta'^{18}O$ to $d_{ln}$ relationship. The modern data cannot readily distinguish whether the added complexity of the nonlinear parameterization is a better fit than the simple linear parameterization. For this reason, the linear parameterization is the most justifiable choice, though the uncertainty
associated with this parameterization on temperature reconstructions will be examined below.

## A5 Air Parcel Mixing within SWIM

We have so far assumed that the moisture-weighted average of a set of independent pseudoadiabatic pathways, defined by a range of evaporation and condensation temperatures, can approximate the conditions experienced by precipitation falling at a polar site. We now aim to test the limits of this approximation and assess the influence of atmospheric mixing on the isotopic
composition of air masses within the simple model framework.

The influence of air mass mixing during evaporation is considered in the discussion of the closure assumption above. Here we consider mixing during transport of air masses with different initial evaporation conditions, different condensation histories, and different temperature, moisture content, and isotopic values at the time of mixing. The central question is whether the processes

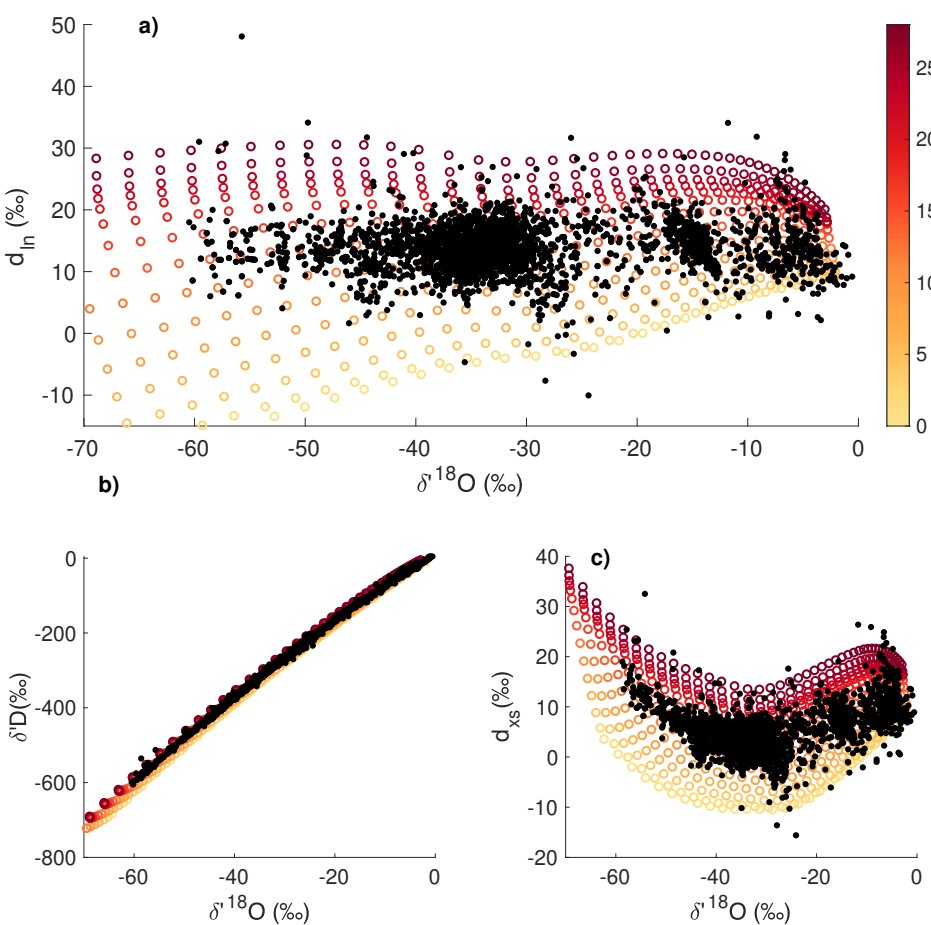

**Figure A12.** Tuning SWIM with linear supersaturation parameterization. **a)** The modeled $\delta^{18}$O and $d_{ln}$ of precipitation (colored circles), for a range of condensation and evaporation temperatures. Color shading shows source region evaporation temperature in $^\circ$C. Black dots are the target data set as described in the text. Modeled results are for the optimized supersaturation parameterization ($S_i = 1 - b \times T$, $b = 0.00525^\circ\text{C}^{-1}$) using the local closure assumption and the NCEP/NCAR reanalysis data set for source region correlations. **b)** Same as panel **a)** for the modeled and target $\delta^{18}$O and $\delta$D of precipitation. **c)** Same as panel **a)** for the modeled and target $\delta^{18}$O and $d_{xs}$ of precipitation.

of mixing can result in isotopic values of final precipitation that are significantly different than the moisture-weighted average of precipitation from un-mixed pathways. There are three processes associated with mixing to consider:

1. Non-uniqueness. Parcels that experienced different evaporation conditions can arrive at a condensation site of a given temperature with different isotopic values.



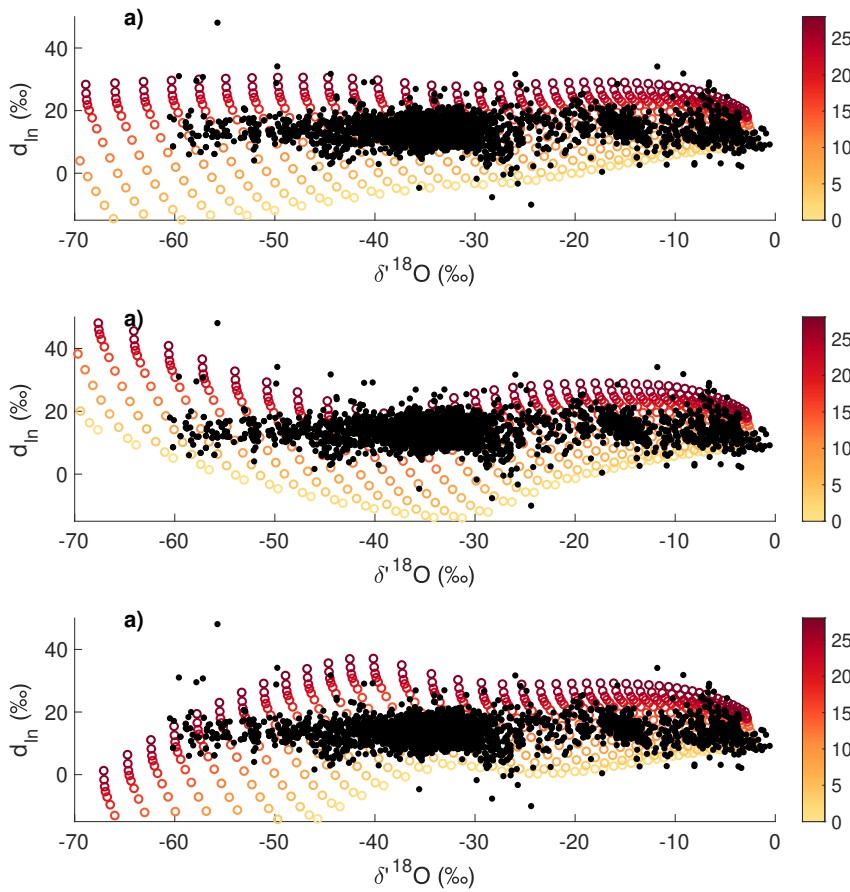

**Figure A13.** Comparison of tuned and rejected supersaturation parameterizations, $S_i = 1 - b \times T$. **a)** Tuned example: $b = 0.00525°\text{C}^{-1}$. Same as in Figure A12a. The modeled $\delta^{18}\text{O}$ and $d_{ln}$ of precipitation (colored circles) and target data set (black dots). Color shading shows source region evaporation temperature in $°\text{C}$. **b)** Same as panel **a)**, but for a rejected tuning: $b = 0.003°\text{C}^{-1}$. The modeled $d_{ln}$ curve upward strongly with $\delta^{18}\text{O}$, incongruently with the target data. **c)** Same as panel **a)**, but for another rejected tuning: $b = 0.007°\text{C}^{-1}$. The modeled $d_{ln}$ curve downward strongly with $\delta^{18}\text{O}$ and incongruently with the target data.

2. Mixing-induced condensation. Mixing two saturated or undersaturated air parcels of different temperatures may result in an oversaturated mixed parcel, due to nonlinearity in the Clausius-Clapeyron relationship. This process leads to additional condensation and fractionation (as well as warming due to latent heat release) and thus a more depleted isotopic value for a given temperature compared to the moisture weighted average of unmixed pathways.





3. Nonlinear mixing. For isobaric mixing of equal-massed air parcels, the final mixed temperature reflects their mass-weighted average (plus the effect of any latent heat release). However the relative abundances of water isotopes mix with the moisture content of the air parcels rather than their total mass. Thus while the temperatures have mixed linearly, the isotopic values of the resultant mixed parcel will be weighted nonlinearly toward those of the warmer and wetter

parcel. The result is that the mixed parcel has a less depleted isotopic value for a given final temperature compared to the moisture weighted average of the the unmixed-parcels.

Moisture-weighted differences between mixed and unmixed parcels only occur when air masses of different temperatures mix. Physically this may represent colliding fronts at synoptic scales. We consider two air parcels evaporated from identical starting conditions but which mix at different temperatures.

The influence of processes 2 and 3 are largest when the relative humidities of both parcels are at saturation, and the magnitude of the influences increase both as the difference in temperature between the two parcels increase and as the absolute temperature of the parcels increase. As temperature decreases the nonlinearity of the mixing of moisture approaches the linear (mass-weighted) mixing of temperature.

To assess the range of temperature differences associated with synoptic scales in the Southern Hemisphere, we examine

the difference in daily-mean two-meter air temperature from the ERA Interim reanalysis (Dee et al., 2011) over the Southern Hemisphere oceans. In summer, the day-to-day temperature differences have a standard deviation of less than 0.9°C and in winter they have a standard deviation of less than 1.5°C (other reasonable metrics of synoptic-scale variability such as lagged 2 or 5-day temperature differences, or grid point to grid point differences are similar). While there is surely the potential for strong mixing for any given synoptic event, for the purposes of paleoclimate reconstruction we are interested in the long term

average of many storm events, and thus the statistics of mixing generally.

We use the simple model to assess the range of final isotopic values of precipitation that can arise from two air parcels, which evaporated at the same initial conditions, then mixed at a range of different temperatures during transport. Air parcels are evaporated at specified initial air temperature ($T_0$ = 10°C) , cooled, and randomly mixed at any combination of temperatures whose difference does not exceed a threshold, then cooled the remainder of the temperature pathway to -30°C. We

assume no preferential temperature of mixing, and no preferential difference in temperature during mixing, though normally distributed probability of mixing with temperature differences up to 5°C (a high-end estimate based on the above analysis of daily temperature differences). This process is repeated 10,000 times to create a distribution of final $\delta^{18}$O (Figure A14.a), $\delta$D (Figure A14.b), and $d_{ln}$ (Figure A14.c) values of precipitation at -30°C, which is compared to the values from a parcel distilled along the same temperature pathway with no mixing (vertical red line in panels a-c, closed circle in panel d).

The resultant distributions are skewed and bimodal. The moisture-weighted means of the mixed distributions are shown in the vertical black lines, while the unmixed final values are shown in the vertical red lines. The means of the distribution are less depleted than the unmixed parcel, owing largely to process 3. Influence of process 2 can also be seen in the additional peak at more depleted values. These distributions vary as a function of both $T_0$ and $T_c$, though the differences in moisture-weighted-means between mixed and unmixed parcels are relatively small and consistent.





The idealized tests above show the influence of mixing (at different temperatures) of air parcels that evaporate from identical source conditions. Perhaps more realistically, air parcels from different sources can mix at different air temperatures, in which case all three mixing processes above are important. This can act to broaden the distribution associated with a given moisture-weighted mean isotopic value. In Figure A14.e-h we show the distribution of isotopic value of precipitation at -30°C from a simulation of 10,000 randomly mixed air parcels as described above, except that the two parcels have two different initial evaporation temperatures, 5°C and 15°C. We compare the distribution to the values associated with unmixed parcels originating at each evaporation temperature, as well as the moisture weighted mean of the unmixed parcels. Differences between the mean isotopic values of mixed and unmixed parcels are less than 0.2‰ for $\delta^{18}$O, 1.5‰ for $\delta$D, and 0.01‰ for $d_{ln}$. Interestingly, although the distributions are broader in this scenario compared to the scenario in which the moisture parcels come from the same evaporative conditions, the difference between the moisture-weighted means of mixed and unmixed parcels are actually smaller. This is presumably because the skewed influence of process 3 (which drives the persistent bias above) contributes less to the total distribution. The relationship between $\delta^{18}$O and condensation temperature and the relationship between $d_{ln}$ and evaporation temperature are similar whether or not mixing is present. Below we investigate the influence of mixing on our temperature reconstruction technique.

## A6    Optimal coordinates for reconstruction technique

Consider a water sample with mean values of $\delta^{18}$O and $\delta$D and normally distributed uncertainties $\sigma_{18}$ and $\sigma_D$. These uncertainties may arise from measurement uncertainty or in the mean $\delta$ value over some time or depth range represented by that sample. The $d_{xs}$ and $d_{ln}$ values of the sample have corresponding $\sigma_{xs}$ and $\sigma_{ln}$, respectively, resulting from the propagation of $\sigma_{18}$ and $\sigma_D$. How do these uncertainties influence the temperature reconstruction?

In Figure A15 we examine the estimation of $T_0$ using coordinates of $\delta^{18}$O and $\delta$D (Figure A15.a), $\delta^{18}$O and $d_{xs}$ (Figure A15.b), and $\delta^{18}$O and $d_{ln}$ (Figure A15.c). The uncertainty in the position of the measurement along both the $x$ axis ($\delta^{18}$O) and $y$ axis (either $\delta$D, $d_{xs}$, or $d_{ln}$) combine to give the total uncertainty in the position on the $T_0$ surface, shown as the targets in (Figure A15.a-c). The total combined uncertainty in the estimation of $T_0$ is shown as probability density functions (PDFs) for each method in Figure A15.d. All estimates yield the same mean value of reconstructed $T_0$, however the widths of the probability density functions are different for each method. The $\delta$D method yields the broadest PDF and thus most uncertain reconstruction. While the PDFs for the $d_{xs}$ and $d_{ln}$ reconstructions are similar, the $d_{ln}$ reconstruction has a narrower PDF and thus more confident reconstruction. This is because the $T_0$ isotherms are most separated along and most perpendicular to the $d_{ln}$ axis. The advantage of the separation of isotherms along the $d_{ln}$ axis is in part compensated by the broadening of $\sigma_{ln}$ compared to $\sigma_{xs}$, due to the propagation of uncertainties. However the angle of the isotherms to the $y$-axis is more important. Given a normal distribution of uncertainty along the $y$ axis, perpendicular isotherms of the variable we wish to reconstruct will result in the narrowest possible distribution of that uncertainty across isotherms. If the angle of the isotherms deviates from perpendicular, as in the case with $d_{xs}$ at more depleted $\delta^{18}$O values, that uncertainty will be spread across a wider range of isotherms. The axis of the influence of $T_0$ on $d_{xs}$ (and $\delta$D) is rotated with respect to its axis of variability in $d_{xs}$.

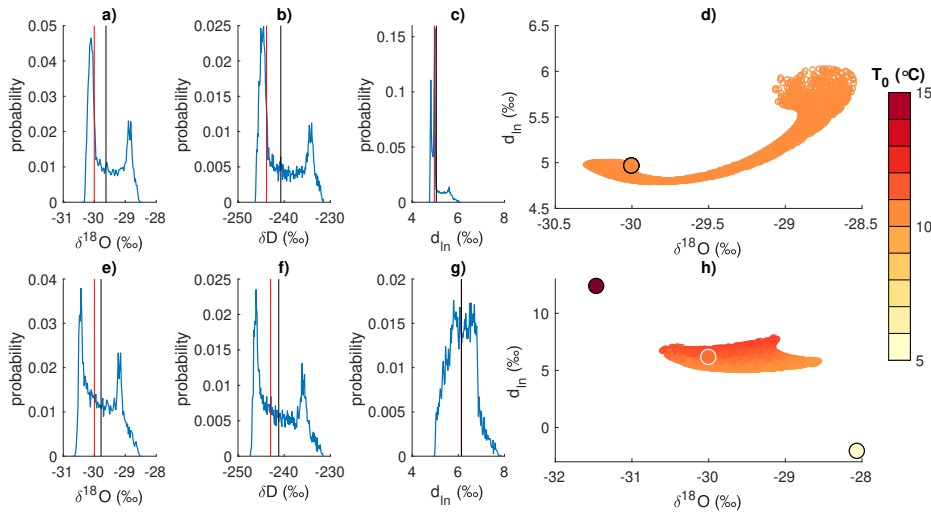

**Figure A14.** Influence of air-mass mixing during transport on water isotope ratios of precipitation. **a-d)** Air parcels from the *same* initial evaporation conditions (namely $T_0 = 10°C$) are distilled, stochastically mixed over a range different temperatures during transport, and distilled to a final precipitation temperature (-30°C). Histograms of the final $\delta^{18}O$ (panel **a**), $\delta D$ (panel **b**), and $d_{ln}$ (panel **c**), are shown and the moisture weighted means of those distributions are shown in the vertical black line. For comparison the final isotopic compositions from an unmixed pathway from 10°C to -30°C are shown in the vertical red lines. The distribution in $\delta^{18}O$-$d_{ln}$ space of the 10,000 mixed pathways is shown in **d)**. **e-h)** Same as for panels **a-d)** but for stochastically mixed air parcels arising from two different initial evaporation conditions ($T_0 = 5°C$ and $T_0 = 15°C$). Moisture weighted means of the mixed pathways are shown in vertical black lines while the moisture weighted means for equivalent unmixed pathways are in red. In **h)** model results are colored by the moisture-weighted $T_0$ resulting form the mixed pathways.

This result is of course ultimately tied to the same reasons that $d_{ln}$ provides a better qualitative proxy of source region changes than $d_{xs}$. The initial imprint of the source conditions are better preserved in $d_{ln}$ than $d_{xs}$. The infidelity of the historical definition of the parameter is the result of nonlinear biases from the linear slope of the definition, the nonlinear nature of equilibrium fractionation, and the cumulative influence of kinetic fractionation during transport (Markle et al., 2017).

5    In Figure A16 we show the SWIM results (under our base assumptions) overlain with every pair of $\delta^{18}O$ and $d_{ln}$ measurements (corrected for changes in seawater $\delta^{18}O$) form the eight Antarctic deep ice core records examined in this study.

**A7    Correlation of nonlinear and linear reconstruction techniques**

We compare the nonlinear temperature reconstructions of all eight ice core sites to linearized reconstructions using SWIM results for Holocene conditions as described in Section 4.3.1. We interpolate all records to even time spacing and compute

10    correlation matrices amongst cores for $T_0$ and $T_s$, using the linear and non-linear, technique (Figure A17). Reconstructed site surface temperatures, $T_s$, are extremely well correlated amongst all cores using either technique, though there is marginal im-



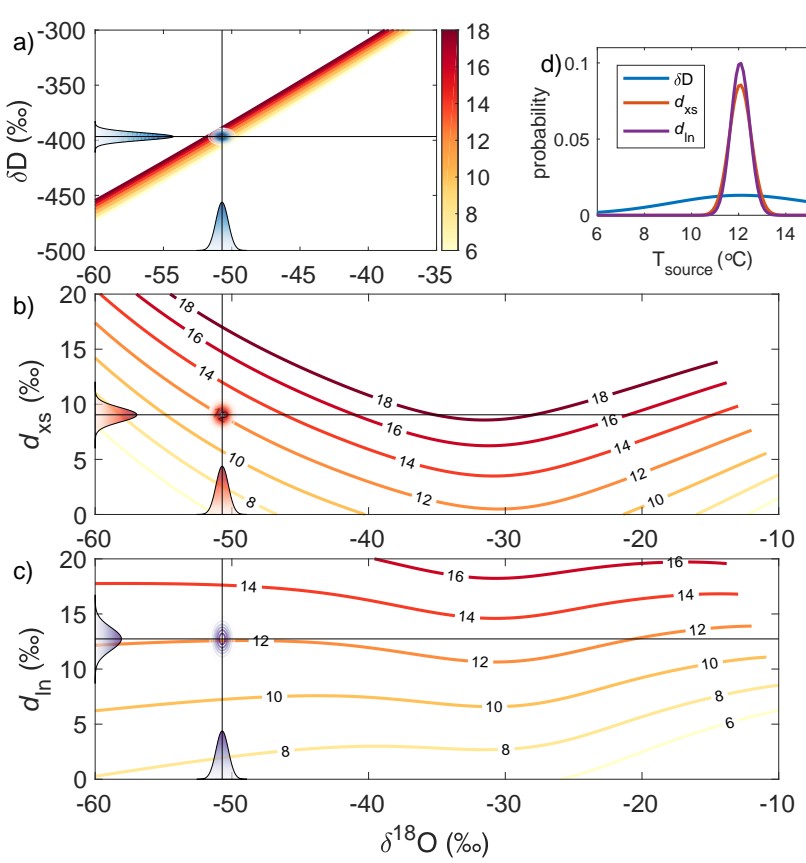

**Figure A15. a)** Evaporation source temperature, $T_0$, contours as a function of modeled $\delta^{18}O$ and $\delta D$ of precipitation. Uncertainty in $\delta^{18}O$ and $\delta D$ for an interval or sample are shown as PDFs of uncertainty along the respective axes. The intersection of these PDFs on the $T_0$ surface result in a 2-dimensional PDF in the reconstructed value of $T_0$, shown as a target. **b, c)** Same as panel **a)** but for the evaporation source temperature projected on to the $\delta^{18}O$ and $d_{xs}$ axes and the $\delta^{18}O$ and $d_{ln}$ axes, respectively. Uncertainties in $\delta^{18}O$ and $\delta D$ in panel **a)** are propagated into the PDF on the $d_{xs}$ and $d_{ln}$ axes, in **b, c)**. **d)** The uncertainty in the reconstructed evaporation source temperature ($T_{source} = T_0$), owing to the weighting of the combined 2-dimensional PDFs from panels **a)** (in blue), **b)** (in red), and **c)** (in purple).





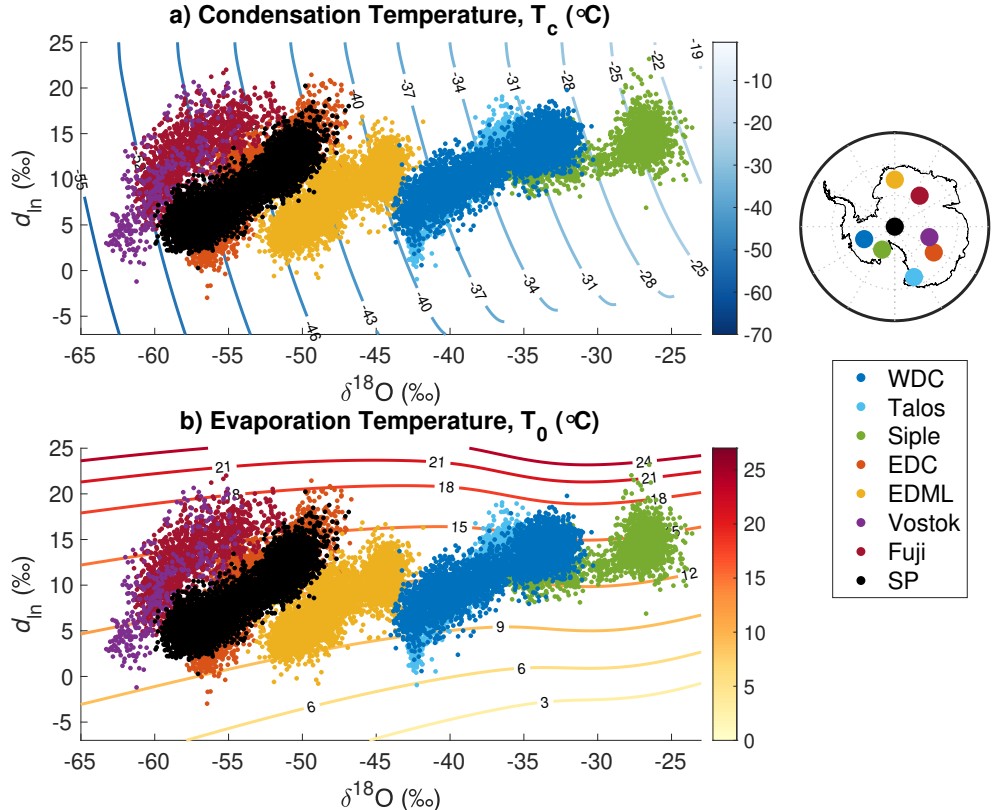

**Figure A16.** Inverted $T_0$ and $T_c$ surfaces as a function of modeled $d_{ln}$ and $\delta^{18}O$ of precipitation as in Figure 6. Overlain are pairs of $d_{ln}$ and $\delta^{18}O$ measurements for eight different deep ice core records, whose site locations are shown on the inset map. See Section 4.2 in the text for details on ice core records.

provement in correlation using the nonlinear technique. In the case of evaporation temperatures, $T_0$, there is dramatic improvement in coherence amongst the records when using the nonlinear technique. The increase in shared variance ($R^2$) explained using the non linear technique is shown in Figure A18. Note that the largest increase in shared variance is associated with the Siple record. This makes sense given the conditions of that site compared to the others and the patterns of partial slopes in

5   Figures 3 and 4.

By accounting for the fundamental nonlinearities in water isotope distillation we are able to reveal more coherent underlying climate signals in source region temperatures, which are otherwise obscured by linear temperature reconstruction techniques. For analogous reasons, Markle et al. (2017) argued that the logarithmic deuterium excess parameter $d_{ln}$ is a more faithful qualitative proxy of source region conditions than the linearly define $d_{xs}$. Compare the correlation matrices of the excess

10   parameters in Figure A19. The nonlinearly reconstructed $T_0$ and $d_{ln}$ parameter share the same correlation pattern amongst the ice cores, and show substantially more coherence than either linearly reconstructed $T_0$ or $d_{xs}$. The correlation pattern of

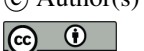



$d_{ln}$ and $d_{xs}$ between all core sites (Figure A20) reveal how nonlinear effects alter the traditionally defined $d_{xs}$ at the coldest Antarctic temperatures. The broad change from positive to negative correlation of $d_{ln}$ to $d_{xs}$ across sites is a reflection of the change in sign of $\partial d_{xs}/\partial T_c$ as a function of $T_c$.

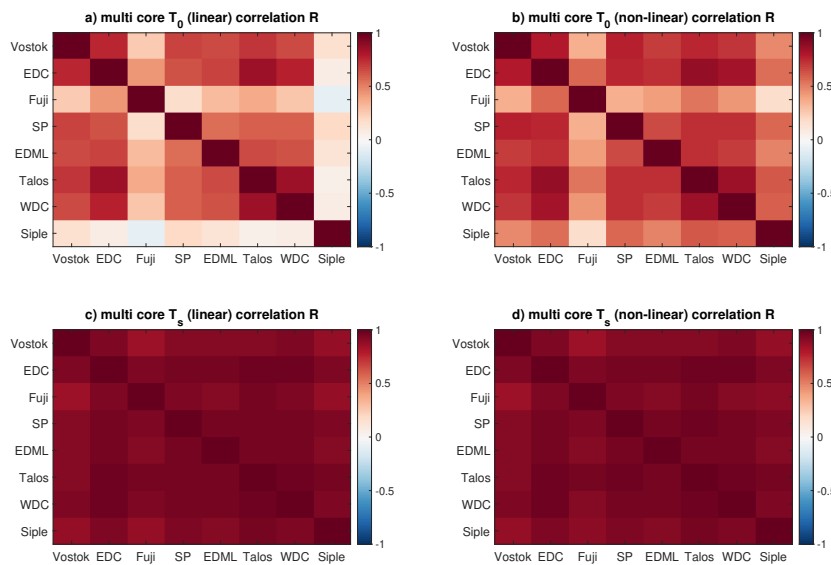

**Figure A17.** Correlation (R) matrices for temperature reconstructions of all core sites. **a)** Correlation among reconstructions of $T_0$ using a linearization of SWIM results for isotopic conditions of the Holocene. **b)** Correlation among reconstructions of $T_0$ using the full nonlinear SWIM results. **c)** Correlation among reconstructions of $T_s$ using a linearization of SWIM results for isotopic conditions of the Holocene. **d)** Correlation among reconstructions of $T_s$ using the full nonlinear SWIM results. All records are ordered by their approximate modern surface temperature.

## A8    Temperature reconstruction uncertainty

In this section we investigate uncertainty in our temperature reconstructions by examining the sensitivity of our results to assumptions and parameterizations in the model. We can compare reconstructed $T_c$ and $T_0$ from a set of $\delta^{18}$O and $d_{ln}$ measurements, using multiple iterations of the model in which the value of a parameter or an underlying assumption has been varied. In Figure A21 we show $T_c$ and $T_0$ reconstructions for the WDC record (Markle et al., 2017) arising from a number of model parameters and assumptions, discussed below. Because our reconstruction technique takes into account nonlinearities, differences in reconstructed temperatures may have mean offsets, and may have differences in variability that vary as a function of $\delta^{18}$O and $d_{ln}$. Thus uncertainty arising from a given parameter may vary between ice core sites. In general, varying a parameter in the model results in patterns of the partial slopes in $\delta^{18}$O and $d_{ln}$ with $T_c$ and $T_0$ that are similar, but shifted in



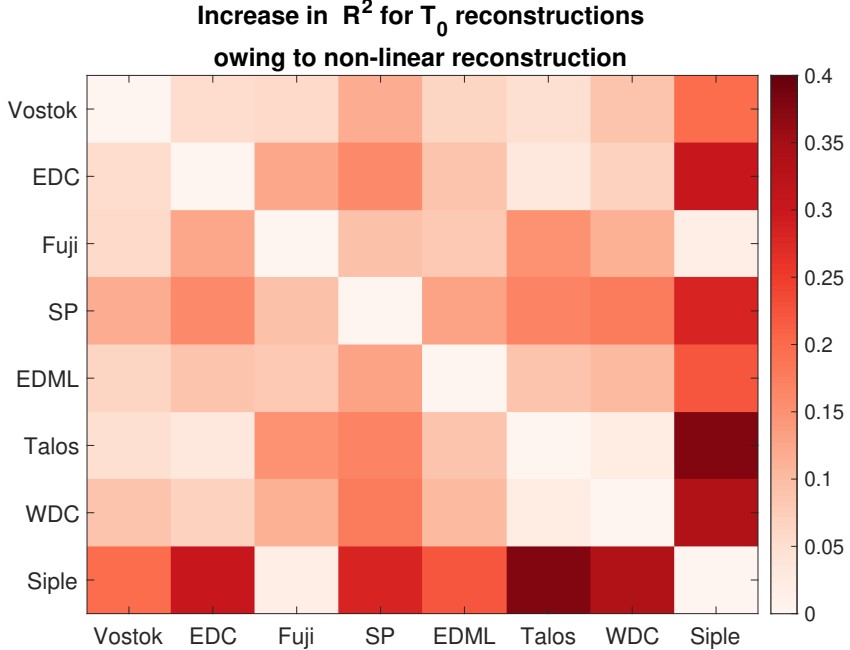

**Figure A18.** The increase in $R^2$ (shared variance) of the nonlinear reconstructions of $T_0$ for all sites over the the linear reconstructions (linearized around Holocene conditions). All records are ordered by their approximate modern surface temperature.

the $T_c$ and $T_0$ space. A consequence of this is that the uncertainty in absolute values of reconstructed $T_c$ and $T_0$ is generally larger than uncertainty in their relative variability.

It is useful to distinguish between uncertainty in the true value of a parameter in the modern climate and the possibility that the effective value may change as a function of climate. Further, not all sources of uncertainty are independent. Varying the

5 value of some parameters may require retuning the model before calculating the isotope state spaces. By ignoring this we risk conflating uncertainty in the reconstruction with bias in the reproduction of the modern mean state.

## A8.1 Sensitivity to model parameters

There is uncertainty in our reconstructions associated with the tuning procedure. While we can constrain the possible values of the $b$ parameter in the supersaturation function by comparison to modern data, variations within a small range should not

10 be ruled out given the imperfect constraint of modern observations. In Figure A21 we show the resultant uncertainty in the temperature reconstruction from uncertainty in the supersaturation parameterization ($b = 0.0051$ to $0.0054 °C^{-1}$). Uncertainty arising from other aspects of the distillation scheme, such as the value of the diffusive fractionation factors during transport, are encapsulated by the tuning uncertainty since adjusting those parameters require retuning the model.



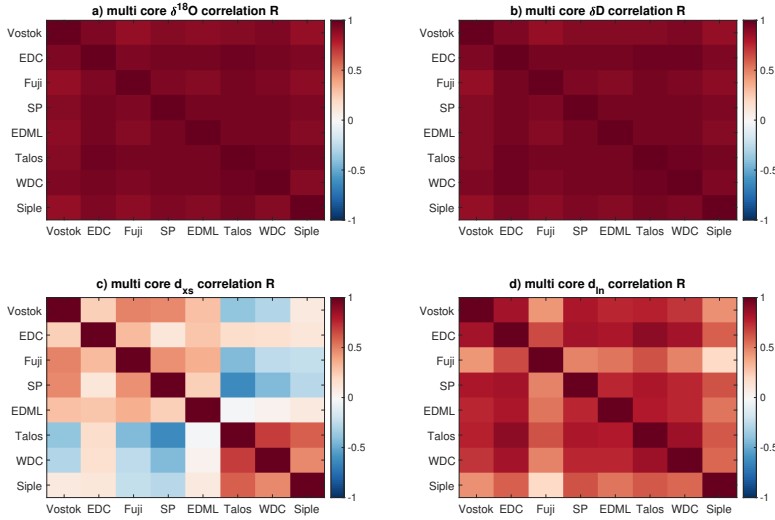

**Figure A19.** Correlation matrices for **a)** $\delta^{18}$O, **b)** $\delta$D, **c)** $d_{xs}$, and **d)** $d_{ln}$ among all core sites. All records are ordered by their approximate modern surface temperature.

Aspects of the initial evaporation scheme introduce uncertainty into our reconstructions. The value of $\alpha^{18}_{diff}$ during evaporation is important in setting the initial isotopic values of vapor. While we find the value 1.009 to give the best fit to modern observations, values within a small range may be defensible (Figure A6). The local closure assumption used in the evaporation scheme has known limitations (Risi et al., 2010), representing an end-member scenario for possible evaporative conditions.

While less applicable to past climate mean states, the global closure assumption provides an extreme test of the model's sensitivity. Using the global rather than local closure assumption can lead to differences in reconstructed absolute $T_0$ up to 1.5 °C for the WDC record, while differences in absolute $T_c$ are smaller ($\leq$ 1 °C). Relative variability in $T_0$ and $T_c$ is similar when using either closure assumption, and $\leq 0.3$°C.

We also examine the influence of the source relative humidity parameterization on our temperature reconstructions. In
our base model we use climatological correlations to determine an initial relative humidity given an initial air temperature; colder surface air temperatures over the ocean are associated with slightly higher relative humidity. We show the difference in reconstructions due to using either the NCEP or ERA Interim reanalysis. We can also ask how different our reconstructed $T_c$ and $T_0$ in WDC would be if we used fixed mean values of initial relative humidity, rather than values that depend on $T_0$. These differences are not true uncertainties in the reconstruction as variable surface relative humidity is a more physically
defensible choice than a fixed relative humidity, though these tests serve to demonstrate the robustness of the reconstruction to model assumptions. In a similar vein we can examine the sensitivity of the model to our precipitation parameterization, and the potential choices of that parameterization discussed in Section A1.2.





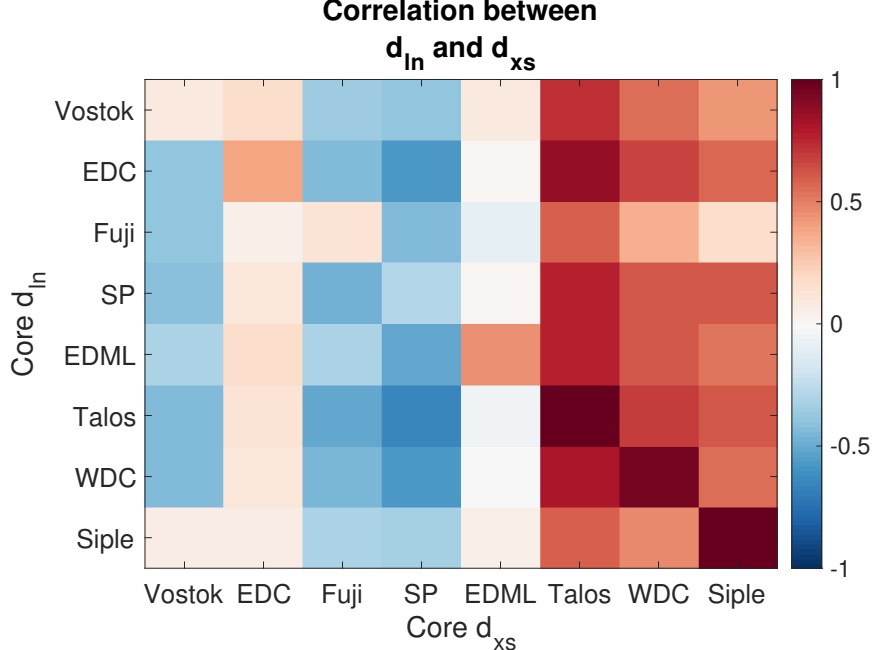

**Figure A20.** Correlation matrix between $d_{xs}$ and $d_{ln}$ between all core sites. All records are ordered by their approximate modern surface temperature.

### A8.2 Influence of mixing on temperature reconstruction

We assess the potential influence of atmospheric mixing on our temperature reconstruction framework by comparing the maps of $T_0$ and $T_c$ as functions of $\delta^{18}O$ and $d_{ln}$ to maps produced by a large ensemble of model runs that incorporate stochastic mixing. We consider a range of final condensation temperatures from -20°C to -50°C. We generate random pairs of pseudoa-
5 diabatic cooling pathways ending at every value of $T_c$ and random values of $T_0$ pulled from a normal distribution (with mean of 12°C and standard deviation of 4°C) similar to the modern Antarctic moisture source distributions. Air parcels cooled down these two paths are stochastically mixed at points along the path, and cooled to the final $T_c$. To mix, parcel temperatures must be above an absolute threshold temperature (-15°C) and have a relative difference within 5°C, as described above. This results in a conservative estimate of the influence of mixing: mixing at lower temperatures reduces the average difference between
10 mixed and unmixed pathways since the effects of mixing are larger when absolute humidity is higher. We take 50 random mixtures from each of $2 \times 50$ random cooling pathways, for each value of $T_c$ between -20°C and -50°C in increments of 0.1°C (a total of $1.5 \times 10^4$ unmixed and $7.5 \times 10^5$ mixed cooling paths). We then interpolate the results of both the mixed pathways and the moisture-weighted averages of the unmixed pathways to create maps of $T_0$ and $T_c$ as functions of $\delta^{18}O$ and $d_{ln}$ (at resolution of 0.1 ‰, Figure A22). Due to the stochastic mixing these maps are unevenly populated. The potential
influence of mixing on our reconstruction technique can be seen in the difference between the mean $T_c$ and $T_0$ maps resulting



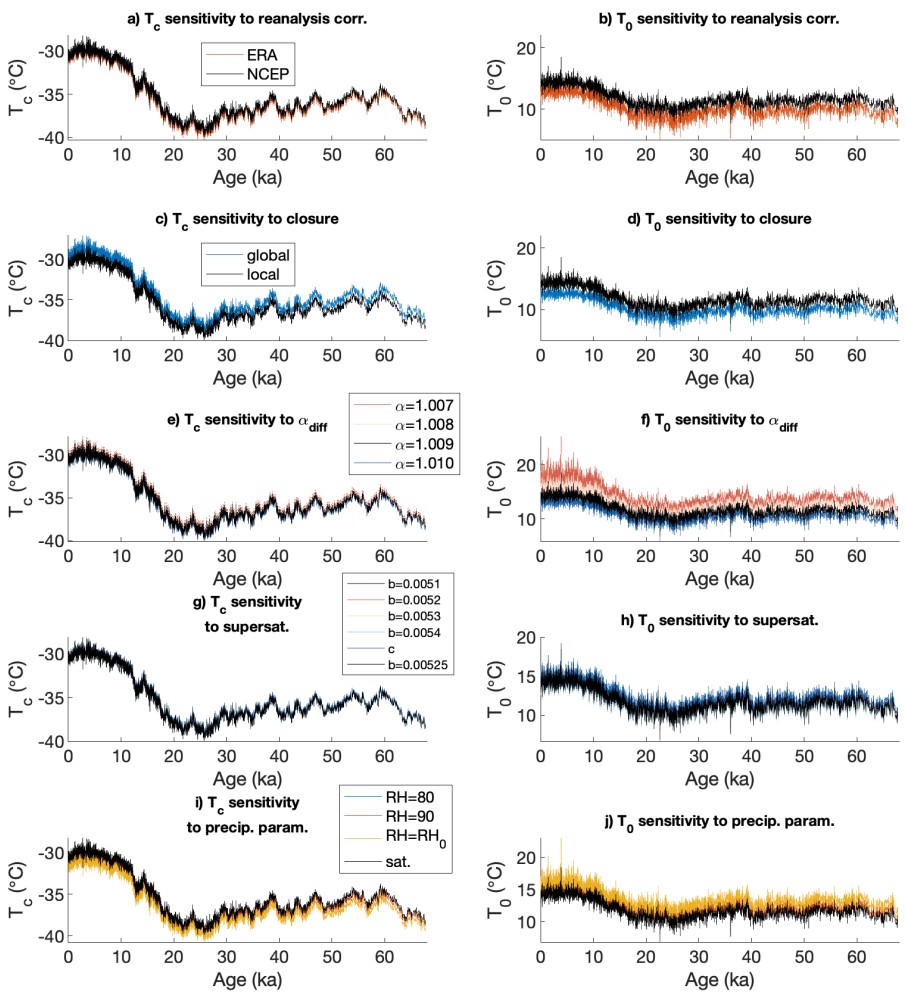

**Figure A21.** Temperature reconstruction sensitivity to model parameterizations. $T_c$ and $T_0$ reconstructions using WDC $\delta^{18}O$ and $d_{ln}$ (re-sampled to 10 year resolution) and SWIM model runs. Black lines show base model in all panels while colored lines are results in which model parameters and assumptions are varied: **a-b)** evaporation condition correlations based on NCEP (black) and ERA (red) reanalysis data; **c-d)** Global (blue) and local (black) closure assumption during evaporation. **e-f)** A range of values for $^{18}\alpha_{diff}$; **g-h)** a range of values for the $b$ parameter in the supersaturation parameterization, as well as a nonlinear parameterization ('c') as described in the text; **i-j)** several versions of the precipitation parameterization in which all moisture is removed above saturation ('sat'), all moisture is removed above initial RH ($RH = RH_0$, constant RH along path), and all moisture is removed above fixed 80% or 90% RH.





from the mixed (denoted with subscript $M$) and unmixed pathways (Figure A22). In Figure A22 we show the histograms of $\Delta T_c = T_{c,M} - T_c$ and $\Delta T_0 = T_{0,M} - T_0$. We test a range of mixing and threshold values in multiple Montecarlo simulations. In all cases the mean values of $T_{c,M} - T_c$ and $T_{0,M} - T_0$ are very near zero, $(< |\pm 0.06°C|$ for $T_c$, and $< |\pm 0.02°C|$ for $T_0$). The spread of the histograms in Figure A22, represent the inherent uncertainty in our reconstruction technique when mixing

is neglected. This uncertainty is less than $\pm 0.2°C$ for $T_c$ and $\pm 0.35°C$ for $T_0$ in all tests. We find similar results when using isobaric rather than pseudoadiabatic cooling pathways. Including moisture sources with $T_0 < 0°C$ in our mixing analysis has no significant influence on the mean difference between the weighted mean maps of either $T_c$ or $T_0$, though expanding the range of moisture sources to include $T_0 < 0°C$, does increase the range of $T_{0,M} - T_0$, by over a degree, in agreement with the analysis of unmixed pathways.

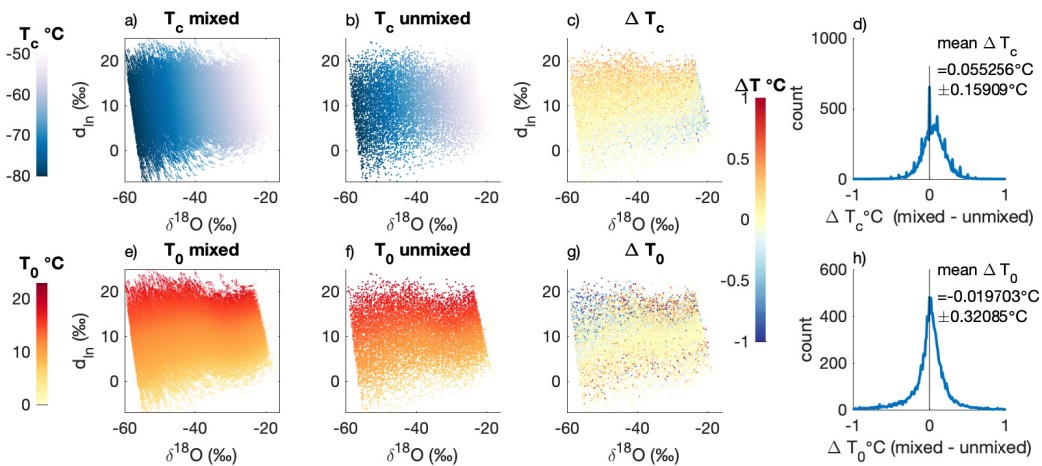

**Figure A22.** The influence of air parcel mixing on modeled isotope state spaces. **a & e)** The water isotope ratios of precipitation resulting from $7.5 \times 10^5$ stochastically mixed distillation pathways, colored by $T_c$ and $T_0$, respectively. **b & f)** The water isotope ratios of precipitation resulting from the corresponding $1.5 \times 10^4$ unmixed distillation pathways, colored by $T_c$ and $T_0$, respectively. **c & g)** The difference in $T_c$ and $T_0$, respectively, between the mixed and unmixed pathways as a function of the $\delta^{18}O$ and $d_{ln}$ space. **d & h)** Histograms of the differences, $\Delta T_c$ and $\Delta T_0$, from all points in the $\delta^{18}O$ and $d_{ln}$ space, resulting from mixing.

## A8.3 Combined Uncertainty estimates

To calculate the total uncertainty in our temperature reconstructions, we examine the combined influence of the major independent sources of uncertainty. These include tuning via the supersaturation function, the closure assumption, mixing in the atmosphere during transport, the precipitation parameterization, the diffusive fractionation factor during evaporation, and the relationship between initial air temperature and relative humidity. We calculate the absolute uncertainty for each $T_c$ and $T_0$

reconstruction interpolated at each pair of $\delta^{18}O$ and $d_{ln}$ measurements, as the absolute difference in reconstructions arising





form perturbations to parameter values or assumptions. To estimate the uncertainty in relative temperature changes we subtract the mean of each reconstruction before calculating differences due to parameter perturbations.

We estimate the uncertainty due to model tuning as the mean absolute difference from the base-scenario for reconstructions using values of $b = 0.0051$ to $0.0054$ in the supersaturation function. Likewise the impact of uncertainty in the value of the

diffusive fractionation factor is estimated as the mean absolute difference of reconstructions using $^{18}a_{diff} = 1.009 \pm 0.001$. We estimate the uncertainty introduced by the precipitation parameterization as the the mean absolute difference from the base-scenario of reconstructions using each of the alternate assumptions outlined in Section A1.2, applied symmetrically to the base scenario. We estimate the uncertainty arising from our assumed relationship between $T_0$ and $RH_0$ as the mean absolute difference in reconstructions using climatological fits from the NCEP/NCAR and ERA Interim reanalysis.

Based on the tests in Section A2.1, a conservative estimate of the uncertainty arising from mixing at the evaporation source is half the absolute mean difference in reconstructions employing the local and global closure assumptions, applied symmetrically about the base-scenario (local closure). Because of the stochastic nature of our atmospheric mixing simulations, our estimates of the $T_c$ and $T_0$ differences are nonuniform and unevenly populated, making interpolation in the $\delta^{18}$O-$d_{ln}$ space challenging (see Section A5). We thus take a conservative estimate of the absolute and relative uncertainty introduced by mixing during

transport as the mean and standard deviation of the differences in the mixed and unmixed reconstructions across the entires state space, respectively (see Figure A14).

We add the uncertainty from each independent source in quadrature as functions of $\delta^{18}$O and $d_{ln}$, symmetric around the base scenario. Finally, we include the additional uncertainty in our estimates of relative $T_s$ variability arising from the $T_c$ to $T_s$ relationship outlined in Section A3.2. We use the mean absolute difference of reconstructions using $T_c \propto 0.69 \pm 0.02 \,^{\circ}C/^{\circ}C \; T_s$

to estimate this uncertainty, which is added in quadrature to with the above uncertainties in $T_s$. An example of this spread of uncertainty on the WDC $T_s$ reconstruction is shown in Figure A23. Reconstructions of $T_s$, $T_s$, and $T_0$ for several major ice core records along with the combined relative uncertainty in those reconstructions is shown in Figure 8.

### A8.4 Uniqueness and source temperature

All Antarctic sites have mean initial evaporation air temperatures above $0^{\circ}$C, according to the the moisture source distributions

from water-tagged GCM experiments (Figure A24). In fact, 85-95% of all moisture that arrives at Antarctic sites in our modeling, initially evaporates from locations with annual average surface air temperatures above freezing. The relatively small but non-zero contribution of moisture from evaporation temperatures below freezing poses an interesting challenge to our temperature reconstruction method. While it is widely known that there is not a unique value of $\delta^{18}$O for every value of condensation temperature owing to the influence of evaporation temperature, there are not necessarily unique pairs of $\delta^{18}$O and $d_{ln}$ for every

pair of $T_c$ and $T_0$, if $T_0$ can be both above and below freezing. The $T_c$ and $T_0$ surfaces fold over on themselves in the $\delta^{18}$O and $d_{ln}$ space, for values of $T_0$ below $0^{\circ}$C. An example of such a folded surface is shown in Figure A25. Given a lack isotopic vapor measurements for evaporation air temperatures much below $0^{\circ}$C and that our evaporation scheme is not well calibrated for such conditions, these results are purely illustrative.





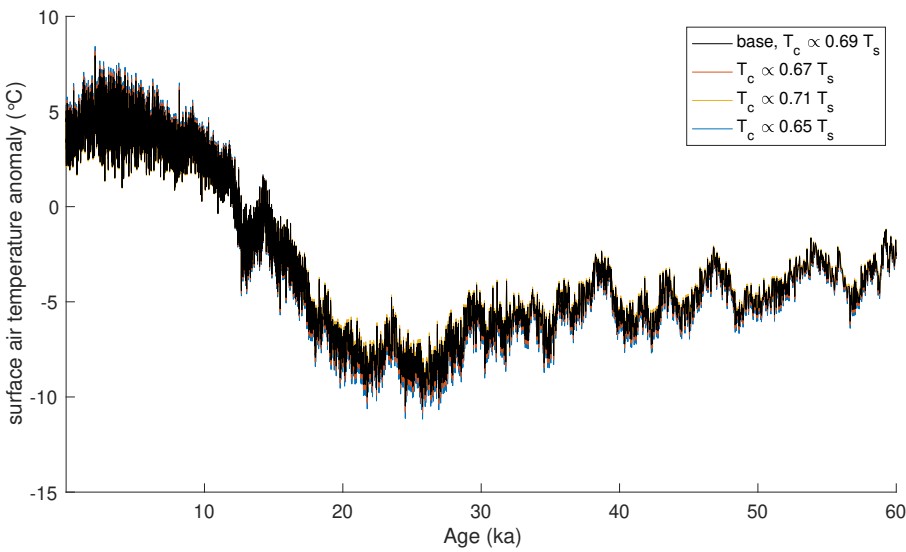

**Figure A23.** Sensitivity of the the $T_s$ reconstruction for WDC to the relationship between $T_c$ and $T_s$. Our base scenario is shown in black ($T_c \propto 0.69^\circ C/^\circ C\ T_s$), while the spread associated with a $\pm 0.02^\circ C/^\circ C$ uncertainty in the scaling factor is shown in red and gold. For comparison the results from a weaker slope of $0.65^\circ C/^\circ C$ are shown in blue. This range of scaling factors has little impact on the reconstructed temperature history.

Those caveats aside, we investigate the sensitivity of our temperature reconstructions to this potential non-uniqueness in the water-isotope state spaces. We run SWIM through a large field of $T_0$ values, from -28°C to 28°C. We can in principle resolve the non-uniqueness problem by combining reconstructions from either side of the folded surface, based on the contribution of total moisture represented by each pair of non-unique paths. Knowing that below-zero moisture sources contribute far less

to the total moisture reaching Antarctic sites (Figure A24), we examine two reasonable methods of moisture-weighting the reconstructions. In the first we simply weight each pair of reconstructed temperatures by the final mixing ratio ($r_{s(eff)}$) of each modeled distillation path. This approach has the advantage of allowing contributions to vary with temperature and thus mean climate, and leads to roughly 10-20% contributions from below 0°C moistures sources to modern Antarctic sites (using a local closure assumption). However, this approach ignores the influence of dynamics and topographic-energetic isolation

in determining Antarctic moisture source distributions, ultimately overestimating contributions from below 0°C moistures sources to higher-elevation, colder sites compared to our GCM-based MSD estimates. In this $r_{s(eff)}$-weighted scheme, higher Antarctic sites have a relatively greater contribution from colder moisture sources than warmer sites, owing to the curvature in the Clausius-Clapeyron relationship. Our moisture tagging analysis and previous studies (e.g Bailey et al. (2019)) suggest, however, that transport dynamics should lead to the opposite relationship. An alternate approach is to specify fixed contributions

from above and below 0°C moistures sources (e.g. 90% $T_0 > 0$°C, 10% $T_0 < 0$°C). While these average relative contributions are based on our moisture tagging analysis, we do not specify contributions as a function of site elevation or mean climate.



Reconstructions based on these approaches (both calculated and specified moisture weighting) are shown in Figure A26 for the WDC record. Considering the non-uniqueness leads to very small differences in reconstructed $T_c$ and $T_0$ *variability*: the standard deviation of differences in reconstructed $T_c$ is less than 0.07°C, using either method, and less than 0.19°C for $T_0$. Attempting to account for this non uniqueness does however lead to persistent mean offsets in absolute temperature, in particular we find colder absolute values of reconstructed $T_0$ for all ice core sites.

While these results are interesting, this attempt to account for non-uniqueness likely does not actually improve the absolute temperature reconstructions. Given the shape of the folded temperature surfaces in the modeled $\delta^{18}$O and $d_{ln}$ space, and the actual values of $\delta^{18}$O and $d_{ln}$ in ice core measurements, the model must extrapolate to extremely cold $T_0$ values for the below 0°C side of the folded surface. These values of $T_0$ are far colder (>10°C colder) than realistic evaporation temperatures likely to contribute moisture to high Antarctic sites given energetic constraints (Bailey et al., 2019) and our moisture tagging GCM experiments. Further, as stated above, our evaporation scheme is not well calibrated to such evaporation conditions. Near surface relative humidity in particular is not well constrained by our climatological correlations in these circumstances. Our model likely underestimates the depletion of the initial evaporate in these circumstances, meaning that the reconstruction solves for a very cold $T_0$ when a much warmer one (and perhaps a reduced $RH_0$) is actually correct. The net result of these considerations is that the analysis above should represent a quite conservative estimate of the influence of non-uniqueness on our temperature reconstructions and their relative variability; the real effect is likely much smaller though difficult to quantify precisely.

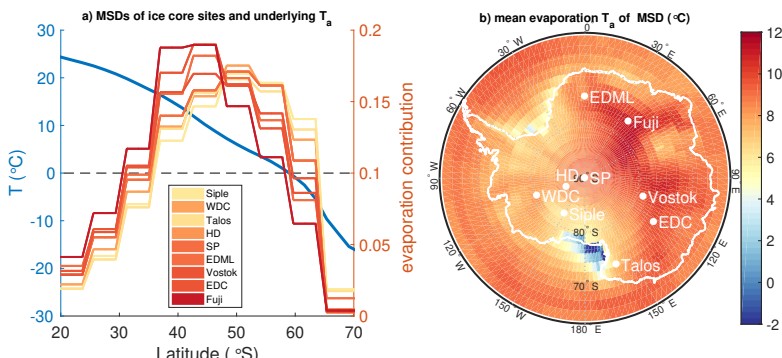

**Figure A24.** Annual mean temperature of moisture source regions from moisture tagged GCM experiments. **a)** Annual mean, zonal mean surface air temperature vs. latitude (blue) overlain with annual mean MSDs for several ice core sites, colored by MSD-weighted annual-mean air temperature. **b)** MSD-weighted annual-mean air temperature for every model grid point over the Antarctic. Note that this is not strictly the evaporation-weighted air temperature.

## A9 Comparison to previous reconstructions

We next reconstruct site and source temperatures for four East Antarctic ice-core records and compare to previously published linear reconstructions. We use records of $\delta^{18}$O and $\delta$D measurements (and calculate $d_{xs}$ and $d_{ln}$) from the Vostok (Jouzel





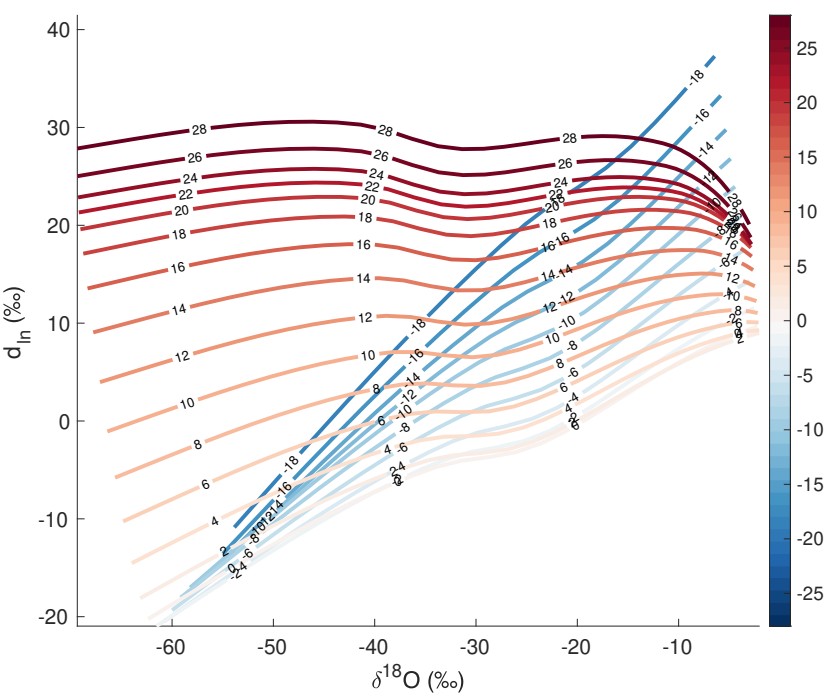

**Figure A25.** Isotope model results colored by initial evaporation air temperature $T_0$ (°C), using base model assumptions. For initial evaporation air temperatures below 0°C there are non-unique results in the $\delta^{18}$O and $d_{ln}$ space.

et al., 1997; Uemura et al., 2012), EPICA Dome Concordia (EDC) (Stenni et al., 2004, 2010), EPICA Dronning Maud Land (EDML) (Stenni et al., 2010), and Dome Fuji records (Uemura et al., 2012). After sea water correction, we use the ice core $\delta$D and $d_{xs}$ for the linear reconstruction, and $\delta^{18}$O and $d_{ln}$ for the nonlinear reconstruction. The linear reconstruction parameters from several studies are compiled by Uemura et al. (2012) (c.f. Tables 1 and 2 in Uemura et al. (2012)). Previous reconstruction

5 techniques solve for the source temperature, $T_{source}$, equivalent to our evaporation temperature, $T_0$, and for the site surface temperature, $T_{site}$. We convert our reconstructed condensation temperatures, $T_c$, to surface temperatures following the method in Section A3.2.

A comparison of relative changes in site and source temperatures are shown in Figure A27. The nonlinear reconstruction results of this study are shown in black, while published linear inversions for each core are shown in color. The difference

10 between the results of this study and the previous temperature reconstructions arise from differences between the linear and nonlinear reconstruction techniques as well as differences in the underlying water-isotope models used for the estimation of scaling relationships. In many cases, the previously-published linear inversions overestimate changes in both site and source temperature compared to the nonlinear reconstruction.



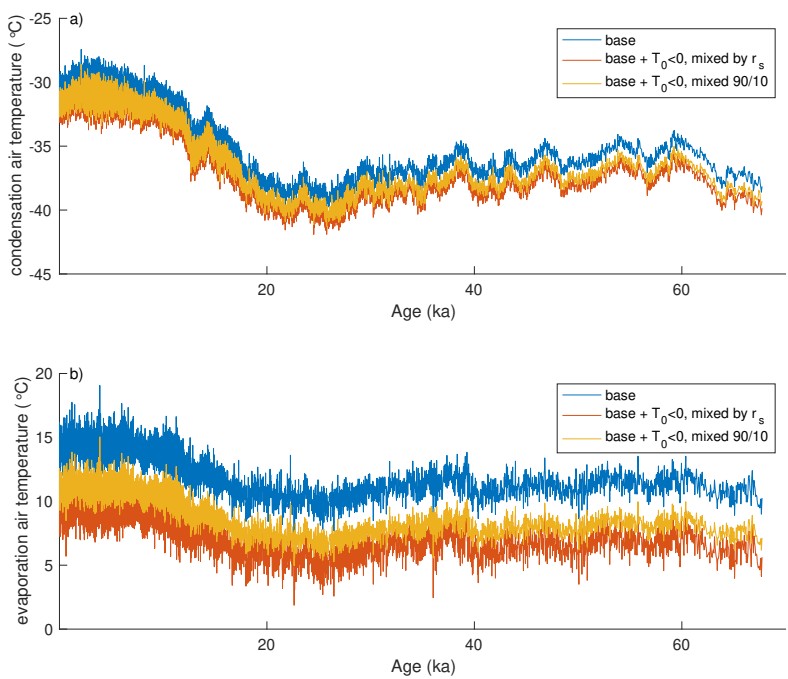

**Figure A26.** Temperature reconstructions for WDC accounting for below zero evaporation air temperatures. **a)** Reconstructions of $T_c$ under base conditions (blue; no contribution form $T_0 < 0°C$), with contribution from $T_0 < 0°C$ weighted by final $r_s$ (red), and with 90% contribution from $T_0 > 0°C$, 10% from $T_0 < 0°C$ (gold). **b)** Same as in **a)** but for reconstructions of $T_0$.

The over-estimation of reconstructed temperature change by the linear reconstruction makes physical sense. The largest source of nonlinearities in the water isotope to temperature relationships are in the deuterium excess parameter, $\partial d_{xs}/\partial T_c$ and $\partial d_{xs}/\partial T_0$. If one assumes these slopes are linear over a given range in $T_0$ and $T_c$, when in reality they are nonlinear, one will attribute a given change in $\Delta d_{xs}$ to a larger change in temperature than is actually required. This over estimate of the required

5    temperature change will be distributed across the reconstructed site and source temperatures in proportion to the values of the $\beta$ and $\gamma$ parameters. The same reasoning is true for nonlinearities in the relationships between $\delta D$ or $\delta^{18}O$ and the temperature boundary conditions, though the nonlinearities in these slopes are much smaller.

The residuals between relative temperature change in the nonlinear and previous linear reconstructions are shown in Figure A28. Residuals in the site temperature reconstructions are on the order of $\pm 2°C$ (Figure A28.a). The residuals are not

10    random but rather correlated to the reconstructions themselves, pointing to nonlinear biases.

The previous reconstructions use a different scaling between surface and condensation than that used in this study (see Section A3.2). However, the differences between the nonlinear reconstruction and the linear reconstructions do not arise solely because of this different surface-condensation temperature scaling. The residuals between reconstructed condensation temper-



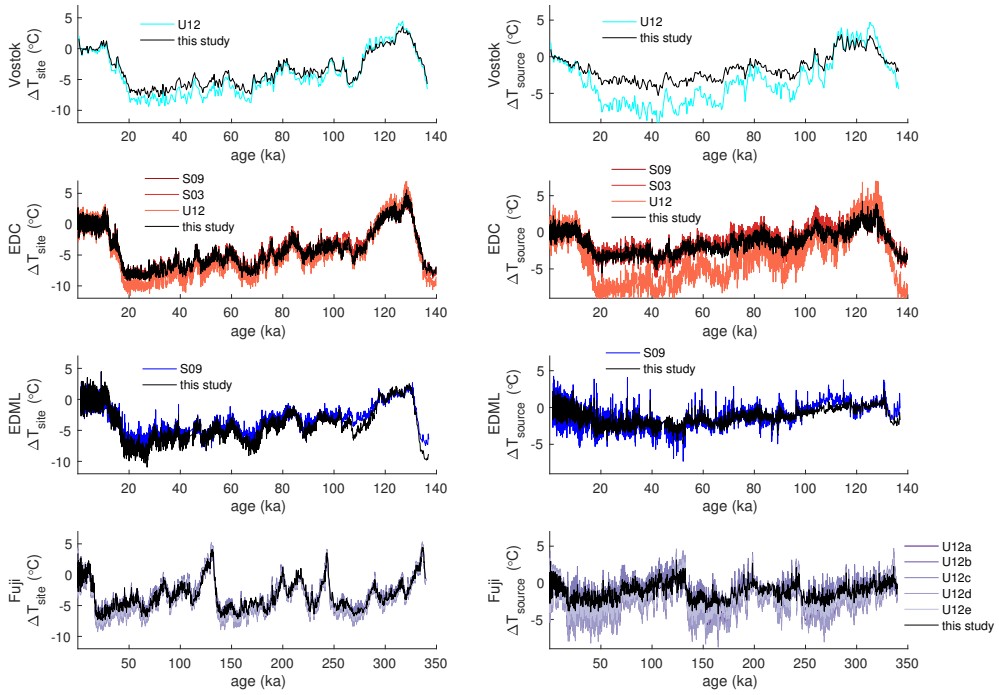

**Figure A27.** Reconstructions of relative change in Antarctic surface temperature ($\Delta T_{site}$, left panels) and source region evaporation temperature ($\Delta T_{source}$, right panels), for four East Antarctic ice-core site: Vostok, EDC, EDML, and Dome Fuji. The non linear reconstructions (this study) are shown in black while published linear reconstructions are shown for each site in color. The linear coefficients for the published reconstructions are compiled in (Uemura et al., 2012) (c.f. Tables 1 and 2). Linear methods labeled U12 for Vostok, EDC, and EDML were calculated by a simple Rayleigh-type model (Uemura et al., 2012). Reconstructions U12a-e for Dome Fuji represent a sensitivity study from Uemura et al. (2012). Reconstructions S03 and S09 are from Stenni et al. (2004) and Stenni et al. (2010).

atures are shown in Figure A28b. These differences are somewhat damped compared to those of the surface temperatures, owing to different assumed slopes in the condensation to surface temperature relationship, but are of similar magnitude and the time series of the residuals are again correlated to the reconstructions themselves.

The residuals between the reconstructed evaporation temperature anomalies (Figure A28c) have a large spread ranging from
5 about +3°C to -5°C. While the magnitudes of source temperature residuals are comparable to those of site temperature, they are far more significant, representing from 50% to over 200% of the total reconstructed variability in the source temperature.

The residuals between the reconstructed evaporation temperature anomalies (Figure A28c) have a large spread ranging from about +3°C to -5°C. As discussed above the largest source of potential biases are in the deuterium excess relationships to temperature, and should be greatest in the reconstruction of source temperatures. While the magnitude of source temperature

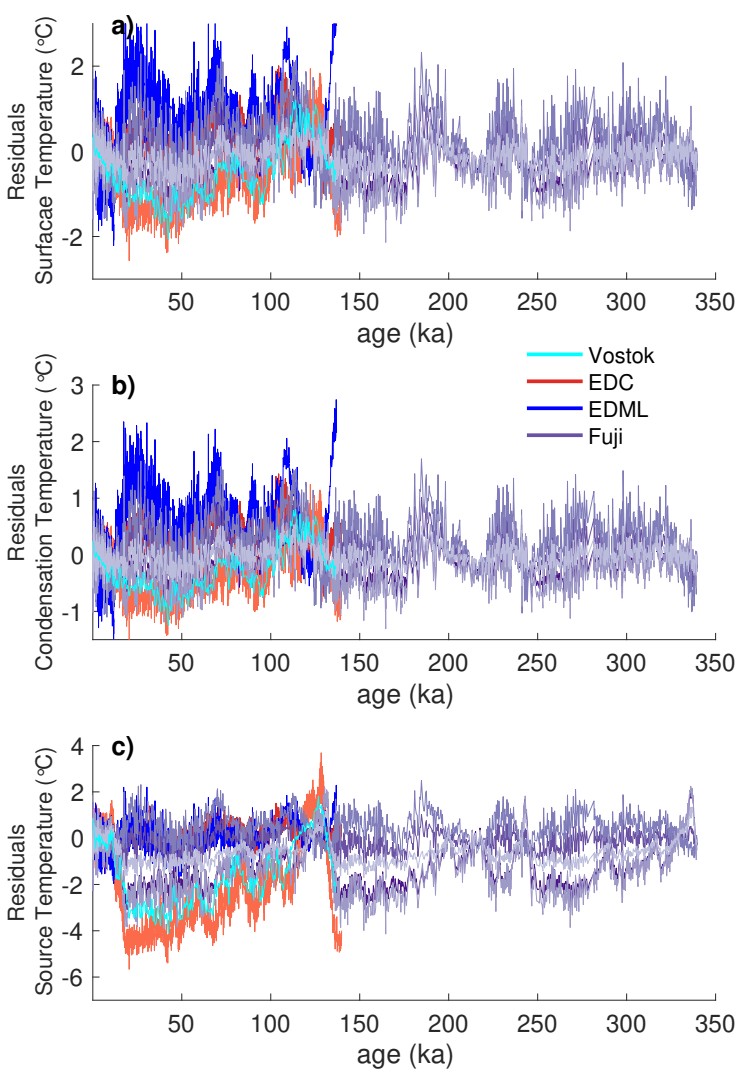

**Figure A28.** Differences between our full nonlinear reconstruction and multiple previously published linear reconstructions (Stenni et al. (2004), Stenni et al. (2010), Uemura et al. (2012)) of a) ice core site surface temperature, b) site condensation temperature, and c) evaporation source temperature, for multiple core sites. Colors correspond to reconstructions shown in Figure A27; Vostok is in cyan, EDC in reds, EDML in blue, and Dome Fuji in shades of purple).

residuals is comparable to those of site temperature, they are far more significant, representing between 50% to over 200% of the total reconstructed variability in the source temperature. This is related to the issues surrounding the qualitative interpretation



of source region changes from $d_{xs}$ versus $d_{ln}$ (Markle et al., 2017; Uemura et al., 2012) (see Section 1.2) and ultimately a consequence of the same distillation effects.

## A10    Three-parameter reconstructions

In the approach outlined above, we have considered the boundary conditions $T_c$ and $T_0$ to be the only independent input variables. In particular, we have assumed that the source region relative humidity, $RH_0$, is a dependent variable whose value is not fixed but determined by climatological correlation to $T_0$. Most previous linear reconstructions have calculated scaling factors based on fixed values of $RH$ or the average of variation in $RH$ over some range (Uemura et al., 2012; Winkler et al., 2012).

We can relax the assumption that $RH_0$ is dependent on $T_0$ and reconstruct three independent climate variables ($T_c$, $T_0$, and $RH_0$) if we have three independent constraints. While $\delta^{18}O$ and $d_{ln}$ alone are not sufficient, the $^{17}O_{xs} = \delta'^{17}O - 0.528\delta'^{18}O$ (Landais et al., 2008) can in principle provide the necessary additional information. We can allow $T_c$, $T_0$, and $RH_0$, to all vary as independent variables, defining a three-dimensional parameter-space, through which SWIM is run to produce three-dimensional isotope state-spaces.

While promising, this method currently has practical limitations. Our model does not reproduce the observed $^{17}O_{xs}$ to $\delta^{18}O$ relationship in Antarctic precipitation to sufficient precision to offer useful constraints. This may be a consequence of model limitations such as missing physical processes. Alternatively (or additionally) uncertainties in the absolute values of $^{17}O_{xs}$ in Antarctica precipitation may be too large to offer useful discrimination amongst variations in $T_c$, $T_0$, and $RH_0$ (Schoenemann et al., 2014).

The $^{17}O_{xs}$ of Antarctic precipitation in our model is sensitive to the supersaturation, diffusivities, and other parameters driving kinetic fractionation. Both small changes in the supersaturation parameterization, and uncertainties in the absolute value of $^{17}O_{xs}$, lead to large changes in the absolute value of reconstructed source-region conditions ($T_0$ and $RH_0$). It is worth noting that absolute values of $^{17}O_{xs}$ are three orders of magnitude smaller than the deuterium excess. Further, preliminary testing suggests that there may be significant non-uniqueness to address, that is a position in the three-parameter space, defined by $\delta^{18}O$, $d_{ln}$, and $^{17}O_{xs}$, does not necessarily lead to unique values of the boundary conditions.

This general approach is scalable. Additional quantities that are both influenced by the environmental pathway and measurable in an ice core, for example accumulation rate (Fudge et al., 2016), water isotope diffusion lengths (Johnsen et al., 2000), or the concentration of aerosols (Markle et al., 2018), may be added to the model. These additional proxies can allow for the reconstruction of additional independent variables and the relaxation of assumptions. Alternatively it may be possible to use the same approach to optimize model parameters like the supersaturation. We leave this task to future work.





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
