# Peer review of "Improving temperature reconstructions from ice-core water-isotope records"

_Climate of the Past, 2021_

## Author Comment (AC2)

- **RC1**: , Anonymous Referee #1, 13 Jul 2021

Markle and Steig are presenting quantitative temperature reconstructions obtained from Antarctic ice cores using an improved methodologies which considers the non-linearities in the water isotope/temperature relationships both at the sites (surface and condensation temperatures) and in the precipitation moisture source regions (SST and initial evaporation temperatures). They use a new and relatively "simple" (at least as stated by them) isotopic model, named SWIM, which relies on previously Rayleigh type models (for example the MCIM) and technically explained in a long Appendix, longer than the manuscript itself, although useful for understanding the whole concept. Furthermore, the authors provide a Southern Hemisphere (SH) temperature changes through time, providing Antarctic stacked records, as well as a spatial pattern of LGM-Holocene temperature change in the SH.

The paper calls into questions previously quantitative temperature reconstructions that used isotopic models as well but were not considering the non-linearities embedded in the isotope/temperature relationships.

The paper is interesting, innovative and the topic is appropriate for Climate of the Past.

Nevertheless, the authors should consider some comments reported below before resubmitting a revised version. I will group these comments below and then some detailed and minor/technical comments will follow.

Reply from authors in red below:

Thank you for this helpful review. We address the specific comments below.

**General comments:**

- The structure of the manuscript: as reported above, the appendix is very long, and I am not sure that from an editorial point of view this is a good point. However, what explained in the appendix is useful for understanding the SWIM isotopic model and the differences respect to the previous ones. So, I would leave the decision on what to do to the editor. However, I would suggest moving at least 1 or 2 figures from the appendix into the main text, in particular those reporting the differences between previous temperature reconstructions and the one reported in the present study (figure A 27) and the one reporting the main moisture sources for the different ice core sites (A24 or A8).

We agree that the appendix is both long and useful to understanding the proposed temperature reconstruction technique. We prefer to keep these details of the model associated with the manuscript describing the temperature reconstruction method rather than a separate manuscript describing the model alone. As the details of the model are relevant, we believe they do need to be published.

We have moved a figure describing moisture sources as well as one comparing temperature reconstructions to the main text.

- The figures: sometimes I found difficult to understand the different colours and, in some cases, for examples for EPICA Dome C the same colour is used for different reconstructions, as in the case of Stenni et al. 2010 and Uemura et al. 2012. Moreover, in some case it is not possible to understand the different ice core records. I believe that this information is needed. Indeed, for EPICA Dome C and probably also for EPICA Dronning Maud Land the old and new reconstructions are quite similar also considering the uncertainties in the reconstruction's methodologies.

We have remade the figures so that these colors are more easily distinguished. The reviewer makes an important point, some previous reconstructions (those of Stenni et al in particular) are similar to those in the present study, despite using both a different reconstruction technique and different underlying water isotope mode. We have made this more clear in the text.

- The impact of these new reconstructions seems to be more important for some sites. Something about this is already mentioned in the text but I would like to visualize better the major differences between previous and this study reconstructions. Perhaps also in the text. See the comments above on the figure colours.

Please see above responses. We have attempted to make these differences more visually clear. To the reviewer's comment that the impact of the new technique depends on the site: yes, this is true. Indeed, this is the point: because the temperature-isotope relationships are themselves nonlinear functions of temperature, the impact of accounting for this dependence in the reconstructions depends on the temperature of the site and the magnitude of temperature variability at the site. We have added additional figures both to the main text and supplement to make these differences more clear.

- Regarding the discussion about past elevation changes in West Antarctica from LGM and the Holocene: I would also refer to the Werner et al. (2018 Nature Comm)

paper regarding this. Regarding the EDML reconstruction, are the upstream effects considered?

Thank you for this comment. We have now included reference to this important paper. As noted in the text, upstream effects are not considered. The topic of this paper is the temperature variability at the site of precipitation at the surface, not the causes of those changes in temperature (which may happen, for a number of reasons, but including because the surface below the site of precipitation changes vertically or horizontally.) A study in preparation uses these naïve reconstructions to address the causes of temperature change across timescales, but is far beyond the scope of the already-long present study.

**Detailed comments:**

Page 10, Figure 5: It would be important to have a legend for the different ice core records. Moreover, some colours are very difficult to see. One record, but I do not know which is (the blue one...) has more positive dxs values at 20 kyr ... which is strange...

The colors in this figure show the mean modern temperature of the site. We have added site names and adjusted the colors to be easier to identify. Yes, several sites (South Pole, Fuji, EDC, and Vostok) all have relatively positive dxs at the LGM. This is owing to the temperature dependence of dxs on site temperature. Please see panel c) of the previous two figures. This is one of the central points of this manuscript.

Page 13, lines 14-19: I do not understand the difference between the absolute and relative uncertainties. Please, may you explain better? An this obviously refers also to Figure 7.

Thank you for this suggestion. We have made this clearer in the text, please see section A8. Recall that many ice-core water-isotope temperature reconstructions explicitly reconstruct relative variability in site and source temperature, that is changes in these variables over time. Our reconstruction technique requires us to reconstruct absolute temperature at the site and source. Of course we can calculate relative changes as well, by simply subtracting a reference temperature. The absolute and relative uncertainty presented in the old Figure 7, reflect the combined uncertainty associated with reconstructing absolute temperature in the past and relative variability in those temperatures, respectively. There is generally less uncertainty in reconstructing relative changes in temperature than absolute values. Section A8 describes these issues in detail. One can understand this in a couple ways. Consider Figure A22 in the original submission: differences in model

choices and parameters lead to different estimates of the absolute evaporation temperature for the site (for example), shown by the different colored lines. However, if we were to subtract a reference temperature from each estimate and only consider relative variability, the differences arising from the different model choices are much, much smaller. Alternatively consider the maps in Figures 2 and 6 of the original submission, which result from a single configuration of the model. To estimate uncertainty associated with a given parameter we calculate new maps after varying that parameter. Generally such maps will have a small absolute offset from the base maps (Figures 2 and 6 of the original submission) but very similar overall patterns of variability. This means differences in relative variability are generally (much) smaller between model configurations than differences in absolute values.

Page 17, lines 9-15: here you are referring to the difference in reconstructions techniques. I would suggest adding here the figure A27 and change the colour for EDC between Stenni et al (2010) and Uemura et al. (2012). See also my comments above.

Thank you for this suggestion, which we have adopted.

Page 17, lines 17-27: in this paragraph you are referring to other T reconstruction techniques. What about a recent paper by Buizert et al. 2021, of which one of the authors of the present study, is co-author? Also considering elevation changes effects and reporting quite different cooling during LGM than the ones reported here as well as previously by also other authors. Perhaps a comment on this would be the case, or here or also later in the discussion paragraph, also referring to Figure 11.

The paper mentioned here was published several months after the submission of our manuscript. We have added reference to this important paper in the main text, and comment briefly on the differences. There are many possible sources of difference between the temperature reconstruction of the Buizert et al paper and those presented here, including potentially large uncertainties in firn modeling and the fact that surface air temperature (the target of reconstruction of this paper) and surface ice temperature (the target of reconstruction of the Buizert et al paper) are necessarily different given atmospheric energy balance. Buizert et al. speculate, for example, that changes in the inversion strength can explain differences between theirs and previous results, but this is speculative and not easily addressed.

Given our estimates of changes in condensation temperature, it is at least possible that changes in the relationship between surface temperature and condensation temperature (related to the Antarctic inversion strength) could reconcile different estimates of surface temperature change at South Pole and Talos dome. For example, our results for South Pole could match those of Kahle et al. (2021) if the LGM relationship between surface and condensation temperature had a slope of about 0.85, or essentially 1, to reconcile with Buizert et al. (2021). While possible this would be a substantial change to the polar atmosphere. Given our estimates of condensation temperature changes at EDC, EDML, and Dome Fuji, it would be difficult to reconcile these with the very small Buizert et al. (2021) firn modeling estimates, through changes in the vertical structure of atmospheric temperature alone. This would likely require a negative lapse rate feedback, such as that in the tropical atmosphere rather than the positive lapse rate feedback seen in the high latitudes. A complete comparison of these techniques is far beyond the scope of the present study. We do briefly discuss the essential comparisons on Pg 21 of the updated manuscript.

Page 42, lines 6-7: this is in contract with observations in precipitation at Concordia station (Stenni et al., 2016) where mean annual precipitation weighted isotopic values are less negative than arithmetical means and temperatures are warmer ..... if I understood correctly.

As discussed in the text, it is possible that there are differences between precipitation-weighted and time-weighted mean temperatures at any individual site (over any short period of time). Indeed in our analysis of reanalysis data, we find that these differences at any single site can be large (a bias of up to 4 deg C in the time period we analyzed), but we find only a very small average bias across the continent (less than 0.33 deg C), nor do we find any dependence of this bias on site temperature. The lack of persistent bias arises, we argue, from variable seasonality of precipitation, and perhaps more importantly, intermittency of precipitation, which together tend to reduce the potential for bias, as described in the supplement. Further, as described in the text, comparison between time-weighted and precipitation-weighted mean temperature at the surface is not the relevant comparison for understanding biases in water isotope surface temperate reconstruction. The relevant comparison is between surface temperature and condensation temperature integrated over both altitude and time. Our analysis of the MERRA2 data take this into account, and we account for this in our scaling between surface and condensation temperature.

Page 42, lines 28-29: in Masson-Delmotte et al. most of the samples are from surface snow (or mean of firn shallow cores) rather than precipitation.

Excellent point. We have made this correction.

**Minor and technical comments:**

Page 1, line 16: Change "Barbante et al." in "EPICA Community Members"; change also in the References.

Thank you (our citation manager struggles with this reference).

Page 1, line 19: add "$*10^3$";

Multiplying (R_sample – R_std)/R_std by "10^3" is incorrect and not the definition of the delta value (though it is common, it is an error). Multiplying by "10^3 ‰" is a common convention (e.g. Dansgaard 1964) and perfectly correct, though redundant (10^3 ‰ = 100% = 1). We have updated the text just prior to the definition to note that delta values are commonly reported in per mil. Clarity about the units and magnitude of the delta values is important as confusion frequently leads to errors in, for example, the calculation of the nonlinear deuterium excess parameter.

Page 14, line 7: add "of" between "function" and "reconstructed".

Corrected, thank you.

Page 15, figure 8: the grey lines are not visible, and the same for light grey and thin dark grey ….

We have made these more visually distinct, though note that the uncertainty is small compared to sample-to-sample variability making it hard to see at this scale (particularly the relative uncertainty; this isn't a color issue but a magnitude issue). We have added a note about this in the figure caption and added additional figures to the supplement for better visualization).

Page 16: figure 9: it is not possible to see in a clear way, moreover no way to understand to which ice core records you are referring.

The point of this figure is not to show the difference for each individual record (which could not be accomplished in a compact figure) but to show the total range of differences across the continent and to emphasize that they vary in time (because they are a function of temperature). However, we have made an additional figure to make the distinction between cores easier to see (added to the supplement).

Page 18, lines 8-10: I would suggest to add here something more about elevation changes ..... see paper from Werner et al 2018.

We have made mention of potential causes of temperature variability, and make clearer our distinction between reconstructing temperature and disentangling its causes, both here and earlier in the text.

Page 19, figure 11: also here I had some difficulties with the colours in panel a).

We assume here you mean the different colors for the ice core sites. We've tried to use easily distinguishable qualitative colors here. With 8 sites, it is difficult to have completely dissimilar colors, though we've attempted to make them easy to see.

Page 21, lines 1..... again the comment above on elevation changes

Done. Thank you for this suggestion!

Page 21, lines 11-12: the linear definition of dxs is an unreliable .... At Dome C this doesn't seem the case..

This is precisely the point. The finding that linear dxs is reasonably linear source variability under some ranges of conditions and not others is why we chose the word "unreliable".

Page 26, line 22: I suppose that "complication" is compilation...

Yes, thank you for this.

Page 32, line 10: I suppose that "modification" is fractionation.

Yes, corrected.

Page 58, line 21:I suppose that one of "Ts" is Tc.

Yes, thank you for this catch.

Page 61, line 1: please, change "EPICA Dome Concordia" into EPICA Dome C.

Corrected, thank you.

Page 63, figure A27: see my comments above regarding EDC (please use different colours for Stenni et al and Uemura et al). Please check also EDML for upstream corrections.

These are different colors on our screen. However we have updated this figure to be much more clear.

As described in the text, we have not made corrections for flow nor elevation change.  Sufficient information is not available for all sites. More importantly,  this is an intentional choice for logical consistency. Ice sheet elevation changes and flow are sources of temperature variability from the perspective of the water isotope ratios as recorded in an ice core. Isolating those from other sources of temperature variability requires additional knowledge or assumptions. To maintain the broad utility of our reconstructions, we leave such considerations for future work. Note that correcting for flow requires assumptions about the surface lapse rate which likely changes with mean climate. Leaving such corrections out of this work requires us to accept that the location for which our reconstructions are representative may have changed slightly over time. This is completely consistent with our moisture source temperature reconstructions, which may have variability both because the temperatures at fixed locations change or because the locations of the moisture sources themselves change, as discussed in the text. We have clarified the discussion of these issues in the text. A complete accounting for the *sources* of temperature variability is the topic of a separate work in prep.

Page 63, lines7-9 and also page 64 lines 1-2: there are two sentences that are repeated.
Thank you, corrected!

**Citation**: https://doi.org/10.5194/cp-2021-37-RC1

---

## Author Comment (AC3)

- **RC2**: ['Comment on cp-2021-37'](), Anonymous Referee #2, 06 Sep 2021

General comments:

This manuscript describes the improved method for reconstructing Antarctic temperature based on ice core water isotope record. Although the manuscript is very long (70 pages), it is organized well and easy to read. The previous temperature reconstruction methods based on the Rayleigh-type model has been properly improved and many potential uncertainties and/or assumptions (closure assumption, inversion temperature, mixing of air mass etc.) are carefully evaluated. Overall, this manuscript is very interesting and suitable for publication in Climate of the Past. To improve the manuscript, I made some comments below.

Thank you for this helpful review!

Major comments

(1) Abstract "..However, there are important nonlinearities that significantly affect such reconstrugion..Here, we describe a temperature reconstruction method that account for these nonlinearities.."

I think the Abstract (and main text) overemphasizes only the difference between linear and non-linerar reconstructions. The difference between this and previous studies results from not only linear/non-linear technique but also different settings (evaporation, supersaturation, etc.) used for isotope modelling. I think the important contribution of this study is the careful examination of various factors one by one, which surely improve the understanding of uncertainty of several important assumptions. I think it is better to write this point in the abstract. In fact, the authors themselves noted that "The difference between the results of this study and the previous temperature reconstructions arise from differences between the linear and nonlinear reconstruction technique as well as differences in the underlying water-isotope models used for the estimation of scaling relationship".

Thank you for this comment. This is an important point. We have updated the abstract. We have attempted to address this issue directly by also conducting linear and non-linear reconstructions within just our model (Figure 9 in the original submission). Our new abstract reads:

"Oxygen and hydrogen isotope ratios in polar precipitation are widely used as proxies for local temperature.  In combination, oxygen and hydrogen isotope ratios also provide information on sea surface temperature at the oceanic

moisture source locations where polar precipitation originates.  Temperature reconstructions obtained from ice core records generally rely on linear approximations of the relationships among local temperature, source temperature and water-isotope values.  However, there are important nonlinearities that significantly affect such reconstructions, particularly for source-region temperatures. Here, we describe a relatively simple water isotope distillation model and a novel temperature reconstruction method that accounts for these nonlinearities. Further, we examine in detail many of the parameters, assumptions, and uncertainties that underly water isotope distillation models and their influence on these temperature reconstructions. We provide new reconstructions of absolute surface temperature, condensation temperature, and source-region evaporation temperature for all long Antarctic ice-core records for which the necessary data are available. These reconstructions differ from previous estimates due both to our new model and reconstruction technique, the influence of which is investigated directly.   We also provide thorough uncertainty estimates on all temperature histories.  Our reconstructions constrain the pattern and magnitude of polar amplification in the past and reveal asymmetries in the temperature histories of East and West Antarctica

(2) Evaporation from the ocean (Appendix A2.1).

P27 L15 ".. this "local" closure assumption.. "

> Since Merlivat and Jouzel (1979) assumed a global steady state of water cycle, the assumption (Rv=Re) has been commonly referred to as "global closure assumption". But, here, the authors termed this as "local" closure assumption. To avoid unnecessarily confusions, it is better to add short explanations about different terminology.

We believe the terminology used here is in line with the literature (e.g. Risi et al 2010, Pfahl and Sodemann 2014, and based on Merlivat and Jouzel 1979 and following Criss 1999. See also Landais et al 2008 and Barkan and Luz 2007). The global closure assumption, as we understand it, requires assuming a global steady state between evaporation and precipitation (whose average delta value is known), which then allows one to estimate R_e globally, from alpha_evap  = R_o/R_e. By contrast, the "local closure" assumption as used here, assumes that local evaporation is the only source of the vapor in question (R_v = R_e; similar to the assumption used in the experiments of Barkan and Luz 2007).  Equations 9 and 10 from Merlivat and Jouzel (1979) lead one to either of these closure solutions,

depending on what is assumed about R_e and whether one closes the water cycle for the global average, or assumes it is closed locally, hence our terminology. Our terminology is described in detail in Section A2.1.

Here is an excerpt from Pfahl and Sodemann 2014: "Merlivat and Jouzel (1979) introduced a global closure assumption in which the isotopic composition of the surrounding vapour $\delta^i_v$ was assumed to be equal to the isotopic composition of global precipitation, which in turn equals the isotopic composition of global evaporation." We do not believe that the assumption described here is the same as equating R_v to R_e (an assumption of local closure), but is properly termed the global closure assumption.

By the way, this section (A2.1) is interesting and includes important analyses. In fact, the different assumptions affect the T reconstructions (Fig A21). So, I think this section, at least some part, may be moved to main text.

Thank you. In order to keep the main text as easily-followable as possible. We prefer not to move this entire section to the main text. However, we appreciate the suggestion and have included mention of some of the most important findings in the main text.

(3) P20 L1-6 "We find smaller glacial-interglacial temperature change for East Antarctic sites compared to previous reconstructions .. The average warming at the two highest sites however, Dome Fuji and Vostok, is significantly less, just 6.9 degC or 59% of that at the lower sites."

This incomplete quotation of our text splices two very separate ideas: 1) comparison of our reconstructions to previous reconstructions, 2) comparison of temperature variability at different sites within our self-consistent temperature reconstructions.

> Previous T reconstructions at DF and Vostok are 7.5-7.8 deg C (e.g., Vimeux et al. EPSL 2002, Uemura et al., CP 2012) (depending on time intervals you choose). Thus, the difference between previous and reconstructions is only 0.6-0.9 degC, which is not significant. For objective comparison, it is better to describe the exact differences from past reconstructions and add short descriptions.

The pervious reconstructions show changes of site temperature up to 18% larger than our reconstructions for the same time intervals. We have included this description of the differences in the text. Moreover, we appreciate the suggestion

of describing exact difference and have updated and moved a figure into the main text (previously in the supplement) to make this point clearer.

The "significantly less" language in the manuscript the reviewer quoted is comparing the temperature change at DF and Vostok to the temperature change at Siple Dome and WAIS, within our own self consistent temperature reconstructions. These differences are indeed significant. This has been made more clear in the updated text.

(4) P53. L11-13 "such as the value of the diffusive fractionation factors during transport.."

Does your model include the eddy diffusive fractionation during transport, like described in Hendrick et al. (GBC, 2000)? If so, please describe it in the Appendix.

Hendricks et al., GBC, https://doi.org/10.1029/1999GB001198

This is a great question. The diffusive fractionation factors mentioned in that line are described in Section A2.2, and refer to moisture diffusion during precipitation formation. The eddy diffusion described in Hendricks et al 2000 (and the fractionation associated with it) relate to their formulation of global moisture transport which they break into (1 dimensional) eddy-driven diffusive mixing and large-scale advection (and whose scheme was later updated by Kavanaugh and Cuffey, 2002, 2003). Their model framework considers moisture transport (and fractionation) on a spatial latitude gid. Our model construction does not; our grid variable is temperature (whose mapping to latitude we do not assume). We assume moisture transport to be pseudo-adiabatic, giving a pressure dimension to our moisture transport (see Figure 2). We then consider the influence of non-adiabatic mixing in modifying the total fractionation, a relaxation of the pseudo-adiabatic assumption of large-scale moisture transport (Section A5). In this way our model construction addresses similar key concepts to those addressed by Hendricks et al (large scale transport and mixing), but using a different framework.

(5) Difference between delta-age based temperature.

A recent paper by Buizert et al. (2021) claimed that Antarctic temperature during LGM is less than those estimated with the water isotopes. And they suggest that the difference can be attributable to an altered Antarctic temperature inversion during the LGM. This finding is very closely related to the topic of this manuscript. Maybe this is beyond the scope of this paper. I would like to ask about some

comments about the differences between this and Buizert et al. (2021) (actually, the second author of this manuscript is a coauthor of Buizert et al. 2021).

Buizert et al. (2021) Antarctic surface temperature and elevation during the Last Glacial Maximum, Science, 10.1126/science.abd2897

The paper mentioned here was published several months after the submission of our manuscript. We have added reference to this important paper in the main text, but comment only briefly on the differences. There are many possible sources of difference between the temperature reconstruction of the Buizert et al paper and those presented here, including potentially large uncertainties in firn modeling and the fact that surface air temperature (the target of reconstruction of this paper) and surface ice temperature (the target of reconstruction of the Buizert et al paper) are necessarily different given atmospheric energy balance. Buizert et al. speculate, for example, that changes in the inversion strength can readily explain differences between our results, but this is speculative and not easily addressed. We discuss these comparisons on Pg 21 of the updated manuscript. A complete comparison of these techniques is beyond the scope of the present study.

Specific comments

Figure 5 left panels > I don not see each profile corresponds to which ice core. Please add appropriate legends in left panels.

We have updated this figure for clarity.

Figure 8 Legends > Please sort in order of condensation temperature, to make it easier to compare with color profiles in panel c and d.

Updated. Thank you for this suggestion.

Figure 9 panel a and b > I don not see each profile corresponds to which ice core.

Yes, this is true. The point of this figure is not to clearly these differences for each core, rather it is intended to show the range of differences for the reconstruction techniques. For compactness we have left this figure as is, however we have added an additional figure to the supplement that shows the difference for each core more clearly.

P8 L5-6 ".. and the changes in slopes across the parameter space are larger than these local changes.."

>I do not understand this sentence.

Thank you for this comment as we should clarify this point. For temperatures near $T_c$ = -30degC, there is a notable change in the partial derivatives of isotope parameters and $T_c$. However the total change in these partial derivates from $T_c$ = 0 to $T_c$ = -65 deg C, are even larger than those changes localized around -30 deg C. We have updated the text to make this more clear.

P17 L3. " .. colder temperature then previous studies.."

> than

Thank you! Corrected.

P21 L21-28 "There is a long-standing debate regarding the interpretation of "spatial" and "temporal .."

> Since this topic is not discussed in the main text, it seems strange to suddenly come up with a conclusion. How about moving this topic to the introduction?

Thank you for this suggestion. We have done so.

P36 L31. ".. that condensation occurs at aa range of temperature up to.."

> a

Corrected, thank you!

Figure A23 > Colored profiles cannot be recognized.

We have attempted to make this more clear. However, please note that the differences are very small, particularly compared to sample-to-sample variability, thus they may be difficult to see regardless of color profile. Indeed this is part of the point of the figure. We have made this more clear in the text.

**Citation**: https://doi.org/10.5194/cp-2021-37-RC2